# The patterns of elemental concentration (Ca, Na, Sr, Mg, Mn, Ba, Cu, Pb, V, Y, U and Cd) in shells of invertebrates representing different CaCO$_3$ polymorphs: a case study from the brackish Gulf of Gdańsk (the Baltic Sea)

Anna Piwoni-Piórewicz[1], Stanislav Strekopytov[2,a], Emma Humphreys-Williams[2], Piotr Kukliński[1,3]

[1] Institute of Oceanology, Sopot, 81-712, Poland
[2] Imaging and Analysis Centre, Natural History Museum, London, SW7 5BD, United Kingdom
[3] Department of Life Sciences, Natural History Museum, London, SW7 5BD, United Kingdom
[a] current address: National Measurement Laboratory, LGC Limited, Teddington, United Kingdom

Correspondence: Anna Piwoni-Piórewicz (apiwoni@iopan.pl)

**Abstract.** The shells of calcitic arthropod *Amphibalanus improvisus*, aragonitic bivalve *Cerastoderma glaucum*, *Limecola balthica*, *Mya arenaria* and bimineralic bivalve *Mytilus trossulus* were collected in the brackish waters of the southern Baltic Sea in order to study patterns of elemental concentration (Ca, Na, Sr, Mg, Ba, Mn, Cu, Pb, V, Y, U and Cd) in shells composed of different crystal lattices (calcite and aragonite). The factors controlling the elemental composition of shells were discussed in the context of crystal lattice properties, size classes of organisms and potential environmental differences between locations.

The study indicates that the elemental composition of studied species is a result of many biotic and abiotic factors acting on elemental incorporation mechanisms in a species-specific way. The incorporation of the main elements Ca, Na, Sr, Mg is mostly related to the crystal lattice properties of calcite and aragonite. Clams that precipitate fully aragonitic shells have a clear predominance of Sr over Mg in shells, contrary to dominant Mg accumulation over Sr in calcitic shells of barnacles. However, the barnacle calcite shell contains higher Sr concentration than bivalve aragonite. The elemental variability between size-grouped shells indicates that trace elements (Ba, Mn, Cu, Pb, V, Y, U and Cd) are were more variable than Na, Sr, Mg, but this is variable between studied species. Moreover, the elemental concentrations tend to be lower in larger than in smaller shells. Biological differences between and within species, such as growth and precipitation rates, feeding strategies including feeding rate and assimilation efficiency seem to be important factors determining the elemental accumulation in shells. We show that when species were obtained from different regions, the impact of local environmental factors associated with sampling location, such as sediment type, cannot be excluded.

## 1 Introduction

Marine invertebrates such as molluscs, brachiopods, corals, echinoderms, bryozoans and some groups of protozoa (foraminifera) are able to use naturally occurring elements to form their skeletons. The combination of inorganic CaCO$_3$ crystals with incorporated elemental impurities and organic compounds with a characteristic and ordered structure, mainly polymers create the composite strengthening material. As a result of long-term adaptation,

calcifiers are armed with hard protective internal and external body parts such as shells, tubes, walls or plates with various components, colours, shapes and functions. The biogenic $CaCO_3$ is deposited mainly in the form of two polymorphs, calcite and aragonite, which are commonly found co-existing in the same specimens and appear to

be precipitated in the same environment (Cusack and Freer, 2008; Morse et al., 2007; Taylor et al., 2008; Stanley et al., 2008). The bulk elemental composition of carbonate skeletons is determined by both $CaCO_3$ crystal lattice and proteins (C, O, H, N, S, P). Furthermore, various other elements, especially those forming divalent ions – mainly Mg and Sr, as well as trace elements e.g. Mn, Br, Cd, Cu, Pb, V – are present as impurities in calcium carbonate skeletons. Metals incorporated into skeletons originate in the environment, yet as some of them are the

components of enzymes or body fluids, the metal pathway to skeleton can be a multistage process (Cubadda et al., 2001; Luoma and Rainbow, 2008). The major sources of metals for organisms during the calcification process are aquatic soluble phases and the food base that may consist of suspended and/or sedimented particles (Newman and Unger, 2003; Rainbow, 1995; Rainbow and Phillips, 1993). However, the bioavailability of metals depends on a complex of environmental (e.g. salinity, pH, sediment type, oxygen conditions) and chemical (e.g. metal-particle

binding strength, the composition of particle, presence of other elements and compounds) factors (Blackmore and Wang, 2002).

Biological carbonates determine the largest component of the hydrosphere's carbon reservoir, and the biogenic production of $CaCO_3$ has a strong interdependence on the ocean composition, biogeochemistry and the carbon cycle (Cohen and McConnaughey, 2003). In the last few decades, increasing attention has been paid to the

relationship between the composition of shells and external environmental factors. It has been repeatedly observed that the mineral type and accumulated metal concentrations in $CaCO_3$ shells of marine invertebrates are integrated records of environmental conditions (Marchitto et al., 2000; Rodland et al., 2006). The elemental composition of the carbonate skeleton can provide records of seawater chemistry and has a significant potential for the fields of palaeoceanography and palaeoclimatology (Freitas et al., 2006; Gillikin et al., 2006; Khim et al., 2003;

Ponnurangam et al., 2016; Vander Putten et al., 2000). However, as recent studies have indicated (Dove, 2010), the influence of environmental parameters on shell precipitation could be very complex. The mineralogy and chemistry of shells are likely to be both linked to environmental conditions and controlled by the organism itself. Even a single population or closely related species within the same habitat may exhibit different accumulation strategies (Rainbow et al., 2000).

Calcite and aragonite differ in structure, symmetry and properties. The calcium ion has 9-fold coordination in aragonite and 6-fold coordination in calcite, and its coordination creates hexagonal and cubic packing (Putnis, 1992). Due to the spatial structure, aragonite shows preferential substitution with larger cations, such as Sr, while smaller cations, such as Mg, are energetically favoured in calcite. In natural systems, calcite commonly incorporates Mg (Morse et al., 2007; Reeder, 1983; Wang and Xu, 2001), and the solubility of Mg-

containing calcite is known to increase with increasing Mg substitution (Kuklinski and Taylor, 2009; Smith et al., 2006). At the temperatures and pressures of the Earth's surface, low-Mg calcite is the most stable form of $CaCO_3$ (de Boer, 1977). Nevertheless, in many marine organisms, aragonite and high-Mg calcite are the dominant phases precipitated from seawater (Dickson, 2004). The solution chemistry (Cusack and Freer, 2008), temperature (Balthasar and Cusack, 2015), pressure (Allison et al., 2001), $CaCO_3$ saturation state (Watson et al., 2012), $pCO_2$

(Lee and Morse, 2010) and phylogenesis (Kuklinski and Taylor, 2009; Smith et al., 1998; Smith and Girvan, 2010) are known to influence shell mineralogy. The main driving force controlling the mineralogy of precipitated $CaCO_3$

is the ratio of Mg to Ca ions in seawater (Cusack and Freer, 2008; Morse et al., 2007). A Mg/Ca ratio > 2 favours the precipitation of aragonite and high-Mg calcite. At high Mg/Ca ratios, such as those in modern seawater (Mg/Ca = 5.2), calcitic structures incorporate Mg, which is observed to inhibit calcite nucleation and growth, whereas aragonite nucleation is not affected by Mg in solution (De Choudens-Sanchez and Gonzalez, 2009; Morse et al., 2007). Most taxa producing low-Mg calcite are known to actively control the amount of incorporated Mg (Bentov and Erez, 2005; De Nooijer et al., 2014). Several studies of organisms that secrete calcareous skeletons have shown that lower seawater temperatures are correlated with the secretion of calcite skeletons with low Mg contents, rather than more soluble high-Mg calcite or aragonite skeletons (Taylor and Reid, 1990). The mineralogy of many calcareous structures changes with latitude, likely as a result of the temperature gradient from the poles to the equator (Kuklinski and Taylor, 2009; Loxton et al., 2014; Taylor et al., 2014). However, this is not an absolute rule (Cairns and Macintyre, 1992). Thermodynamics predicts that aragonite is the stable phase at pressures higher than 5000 hPa (roughly 40 m depth), and calcite is the stable phase at lower pressures. However, aragonite is still the major constituent of shells or pearls, indicating its metastable formation in shallow waters (Sunagawa et al., 2007). The incorporation of Sr was suggested to play a significant role in the biomineralogical precipitation of aragonite (Allison et al., 2001). Many studies have demonstrated a clear correlation between the concentration of Sr in the hard parts and precipitation of the aragonite layer (Iglikowska et al., 2016; Reeder, 1983). The ionic radius of Sr is larger than that of Ca; thus, Sr is more likely to form 9-fold coordination, which triggers metastable aragonite nucleation (Sunagawa et al., 2007).

Many studies have demonstrated that the biological control of shell composition is often more important than the environmental control (Carré et al., 2006; Freitas et al., 2005, 2006; Gillikin et al., 2005b). In biologically controlled mineralization, the organism drives the process of nucleation and growth of the minerals independently from the environmental conditions. Endogenous factors manifest themselves through co-regulation of all the structures and functions of the organism, including its sex, growth rate, metabolism, and feeding strategy (Lowenstam and Weiner, 1989). The main physiological processes involved in metal accumulation are ingestion, assimilation, elimination and growth (Wang and Fisher, 1997). Throughout the lifespan, the biological system experiences ontogenetic trends and seasonal variations in physiology, determining metabolic expenses based on life's needs. Biological effects have been repeatedly used to explain shifts of elemental concentrations in shells from a theoretical equilibrium (Davis et al., 2000; Roger et al., 2017; Watson et al., 1995). Ontogenetic fluctuations of the growth rate and metabolic activity affect the intensity of the metal uptake (Lee et al., 1998). Vander Putten et al. (2000) concluded that the seasonality of the accumulation of Mg, Sr and Pb in *Mytilus edulis* shells shows significant similarity across individuals, with a maximum during spring and early summer, and the elemental profiles cannot be explained by seasonal variations in the seawater composition. Carré et al. (2006) developed a model of ion transport in bivalve shells that shows that $Ca^{2+}$ channels are less ion-selective when $Ca^{2+}$ fluxes are higher. Other studies have found that the rate of trace metal uptake increases as mussel filtration rate increases (Janssen and Scholtz, 1979).

It is still difficult to separate the degree to which the organism itself controls the composition of the carbonate skeletons and how much this process is affected by the environment (Casella et al., 2017; Schöne and Krause Jr, 2016; Weiner et al., 2001). Although calcareous parts serve as a powerful tool for the interpretation of environmental conditions, a detailed insight into the different factors controlling the composition of $CaCO_3$ is critical. Tracking potential sources of variation within calcifying organisms of many species with diverse

mineralogy should provide valuable insight into the patterns driving the biomineralization process (Smith et al., 1998).

The aim of this study is to assess the patterns of metal variability (Ca, Na, Sr, Mg, Mn, Ba, Cu, Pb, V, Y, U and Cd) in shells of mussels and barnacles representing three possible mineralogical forms. The selected species were clams: aragonitic *Cerastoderma glaucum*, *Mya arenaria*, and *Limecola balthica*, bimineralic mussels *Mytilus trossulus*, and barnacle *Amphibalanus improvisus* with a fully calcitic shell, from the low-salinity environment of the southern Baltic Sea. A comparison of elemental concentration levels between size classes is also performed. Assuming the larger specimens are older than the smaller specimens of the same species it will allow us to demonstrate how different ontogenetical stages influence elemental accumulation. The potential influence of biological control and local environmental conditions on the observed metals concentrations in shells is briefly discussed. The study area offers the variety of fluctuating factors, giving the opportunity to increase our understanding of elemental variation patterns of skeletons formed in the area. Brackish waters affect the activity and speciation of metals, enhancing their bioavailability (Fritioff et al., 2005). The seasonal changes (e.g. surface temperature, primary production and freshwater inflow) determine the metal sources and drive the physiological processes of living organisms (Urey et al., 1951).

## 2 Material and methods

### 2.1 Study area

The study area is located in the Gulf of Gdańsk in the southern Baltic Sea, more precisely, in the outer Puck Bay and central Gulf of Gdańsk (Fig. 2). The gulf is partially sheltered from the northwest by the Hel Peninsula and from the west and south by the coastline (Kruk-Dowgiałło and Szaniawska, 2008; Rainbow et al., 2004). This location makes the seawater the most turbulent in January and the calmest in June, with weak bottom currents and minimal tidal amplitudes. The hydrophysical parameters of the gulf are mostly driven by the temperate climate and the following seasonal changes. Differences in air temperature and water mixing cause seasonal fluctuations of the surface water temperature, ranging from approximately 4 to 22°C (Uścinowicz, 2011). The Gulf of Gdańsk is a low-salinity system under the influence of brackish water from the open southern Baltic Sea and fresh waters from rivers, mainly the Vistula River; the Vistula is the largest river in Poland and has an average annual inflow in the estuary of 1080 m³ s⁻¹, which varies seasonally from 250 to 8000 m³ s⁻¹ and has a maximum in spring (Cyberski et al., 2006). Thus, the average water salinity in the gulf is 7, varying from approximately 5.5 in summer to 8.4 in winter (Bulnheim and Gosling, 1988; Szefer, 2002).

The Gulf of Gdańsk is an area highly influenced by human activities. This is due to intensive usage of its resources and to anthropogenic emissions originating from various coastal sources, river inflows and atmospheric deposition. The most significant input of industrial and municipal pollution into the gulf is derived from the Vistula River, which transports pollutants from a catchment area of 194,000 km² (Pruszak et al., 2005). Both the water discharge and sediment load into the gulf are strongly seasonally dependent. The maximum river flow typically occurs in spring, with the minimum in autumn and winter, mainly due to periods of snow and ice melting (Cyberski et al., 2006; Pruszak et al., 2005; Szefer et al., 1996). Winter storms cause re-deposition of sediments further into the Gulf of Gdańsk (Damrat et al., 2013). Because of the local conditions, mainly the limited water exchange,

river-borne contaminants remain in the ecosystem for decades, accumulating in the sediments and in living organisms (Glasby et al., 2004; Szumiło-Pilarska et al., 2016).

Most regions of the Baltic Sea have lower salinity and alkalinity (that is, lower and $Ca^{2+}$ and $CO_3^{2-}$ concentrations) than oceanic surface waters (Beldowski et al., 2010; Cai et al., 2010; Findlay et al., 2008). Due to the seasonality of temperature and biogeochemical cycle, the amplitude of $CaCO_3$ saturation state ($\Omega = [Ca^{2+}]$ $[CO_3^{2-}]$ $K_{sp}^{-1}$, $K_{sp}$ is a function of salinity, temperature and pressure, Kawahata et al., 2019) in the Baltic Sea is high in comparison to saline waters. It alternates between approximately 1 to 5 for calcite and 0.5 to 2.5 for aragonite (Findlay et al., 2008).

The community of calcifiers from the Gulf of Gdańsk is characterized by the dominance of benthic filter feeders and deposit feeders (Kruk-Dowgiałło and Dubrawski, 1998). These organisms exploit their growth potential during the productive seasons of the year when the availability of suspended matter, mainly phytoplankton is the highest (Pierscieniak et al., 2010; Staniszewska et al., 2016).

**2.2 Species**

The clam *Cerastoderma glaucum* (Mollusca, Bivalvia; Fig. 1.1), commonly known as the lagoon cockle, is a saltwater clam found along the coasts of Europe and North Africa, including in the Mediterranean and Black Seas, the Caspian Lake, and the low-salinity Baltic Sea. It is a euryhaline species living in salinities between 4 and 84. *C. glaucum* can tolerate habitats with wide range of temperatures, from periodically freezing to above 30°C. It is a filter feeder that burrows 2–4 cm below the surface in soft sediments at shallow depths of secluded bays. It actively lives near the sediment surface, acting as a biodiffuser (Urban-Malinga et al., 2013). The clam is surrounded by a ribbed aragonite shell, which is externally yellowish to greenish brown (Jelnes et al., 1971). In the brackish environment of the Gulf of Gdańsk, *C. glaucum* spawns in May-July and typically lives up to 4 years, achieving a height of 27 mm (Żmudziński, 1990).

The soft-shell clam *Mya arenaria* (Mollusca, Bivalvia; Fig. 1.2) is a marine invasive species introduced into European waters from the Atlantic coasts of North America (Behrends et al., 2005). It has a wide global distribution, mainly due to its adaptability to varying environments with salinities between 4 and 35 and temperatures between –2 and 28°C (Gofas, 2004; Strasser et al., 1999). *M. arenaria* burrows into the sediment up to 20–30 cm below the surface of the sea floor. *M. arenaria* is a filter feeder, filtering organic particles and microinvertabrates using long fused siphons, and a deposit feeder. In the Gulf of Gdańsk, *M. arenaria* is a common inhabitant of shallow waters down to a depth of 30 m. It spawns once or twice a year in spring or summer, at temperatures of 10 – 15°C. Individuals live 10 – 12 years. They have aragonitic shells and grow up to 70 mm (Żmudziński, 1990).

The clam *Limecola balthica* (Mollusca, Bivalvia, Fig. 1.3) lives in the northern parts of the Atlantic and Pacific oceans, in Subarctic and European waters from southern France to the White Sea and Pechora Sea, including the Baltic Sea (Strelkov et al., 2007). It is a euryhaline clam capable of living in a wide range of water salinities from 3 to 40 and at temperatures from –2°C to above 30°C (Sartori and Gofas, 2016). *L. balthica* is a filter feeder and deposit feeder that lives buried a few centimetres below the surface. It has a semi-sessile lifestyle, with the ability to undertake periodic migrations (Hiddink et al., 2002). In the Baltic Sea, *L. balthica* lives at depths of down to 40 m and grows to 24 mm. Adults reproduce in spring when the water temperature reaches 10°C and

live 12 years (Żmudziński, 1990). They have aragonitic shells varying in colour between individuals and locations, mainly exhibiting white, pink, yellow and orange (Sartori and Gofas, 2016).

The mussel *Mytilus trossulus* (Mollusca, Bivalvia; Fig. 1.4) is one of three closely related taxa in the *Mytilus edulis* complex of blue mussels, which, collectively, are widely distributed in the temperate and cold-water coasts of the Northern Hemisphere and are often dominant organisms on hard substrates of shallow nearshore habitats (Rainbow et al., 1999; Wenne et al., 2016). Generally, *M. trossulus* has a life span of approximately 12 years and grows to 100 mm, yet in the estuarine environment of the Gulf of Gdańsk, it reaches a maximum length of approximately 50 mm (Gofas, 2004; Żmudziński, 1990). Mussels are sessile filter feeders, mainly depending on phytoplankton. They reproduce from late spring to early autumn, depending on the temperature and food abundance (Larsson et al., 2017; Lauringson et al., 2014; Rainbow et al., 2004). The shell of *M. trossulus* is bimineralic and consists of two calcium carbonate layers: an outer calcite and inner aragonite layer in variable proportions between individuals (Dalbeck, 2008; Piwoni-Piórewicz et al., 2017).

The barnacle *Amphibalanus improvisus* (Arthopoda, Maxillopoda; Fig. 1.5), commonly named the bay barnacle, is a small sessile crustacean that typically exists in shallow coastal zones that are less than 10 m deep. It is widespread around the Atlantic and has been dispersed by shipping to many parts of the world, now having a worldwide distribution. It is a euryhaline and eurythermal species that is absent only from the Arctic and Antarctic seas (Kerckhof, 2002). *A. improvisus* is a filter feeder that inhabits hard substrates. In the Baltic Sea, the reproduction of barnacles starts in spring with temperatures over 10°C and ends in autumn. The species grows to approximately 10 mm in diameter with a maximum height of approximately 6 mm, and generally, it has longevity of one year (Żmudziński, 1990). It has a conical shell composed of six fused calcite plates (Weidema, 2000).

**2.3 Sample collection and preparation**

Samples of shells were gathered by a Van Veen grab sampler from four stations: GN, MA, M2 and MW, located in the Gulf of Gdańsk (Table 1, Fig. 2). Three species were found at MA (*Mya arenaria*), M2 (*Cerastoderma glaucum*) and MW (*Limecola balthica*), while two species were collected at station GN (*Mytilus trossulus*, *Amphibalanus improvisus*; Table 1). To ensure that the samples were not contaminated or modified by solutions of preservatives, the collected material was transported alive in tanks filled with seawater to the laboratory, where sample preparation took place.

By measuring the shell heights of clams and shell diameters of barnacles using a calliper with an accuracy of ±1 mm, the shells were classified into four size classes. The division into size classes was performed on the basis of the size reached by each species in the southern Baltic Sea environment (Table 2). Forty shells (ten in each class) were selected for *A. improvisus*, *C. glaucum*, *M. arenaria* and *L. balthica*. For *M. trossulus*, 20 shells were selected, with five in each class, while the results for the rest were obtained from Piwoni-Piorewicz et al. (2017). After the removal of soft tissues, each shell was viewed under a stereoscopic microscope to examine the epibiotic flora and fauna, which could contaminate the sample and bias the chemical analysis. If any organisms were present on the shell, they were carefully removed. To remove the biofilm, the periostracum was scraped with a scalpel, and pre-cleaned shells were placed in an ultrasonic bath (InterSonic IS-7S) in ultra-pure water for 30 minutes and then dried at 70°C for 24 hours. The shells were then crushed and ground into a fine powder with an agate mortar and pestle. Aliquots (2.8 − 849 mg, mean = 132 mg) of the powdered samples were weighed using a 5-digit analytical balance, placed into a 15 ml plastic tube (Sarstedt®) and dissolved in a mixture of 1.5 ml

concentrated nitric acid (HNO₃, Sigma Aldrich®, Trace SELECT for trace analysis), 1.5 ml ultra-pure water and 0.3 ml 30% hydrogen peroxide ($H_2O_2$, Merck® Suprapure grade). After 24 hours at 70°C, the liquid samples were diluted to c. 15 ml by weight with ultra-pure water.

**2.4 Elemental analysis**

Concentrations of chemical elements in the digested samples were determined at the Natural History Museum, London, using a Thermo iCap 6500 Duo inductively coupled plasma optical emission spectrometer (ICP-OES) for Ca, Na, Sr and Mg and an Agilent 7700x inductively coupled plasma mass spectrometer (ICP-MS) for Mn, Ba, Cu, Pb, V, Y, U and Cd. Calibration of the ICP-OES analysis was performed using solutions containing 1–50 mg $l^{-1}$ Ca, 0.01–5 mg $l^{-1}$ Na, and 0.001-0.5 mg $l^{-1}$ Sr and Mg in 0.8 M HNO₃. Na, Mg and Sr calibration solutions all

contained Ca at one hundred times the Mg concentration, to account for the effects on other wavelengths by high Ca contents in the samples and standards. Multiple wavelengths for each element were recorded, and line selection was performed by accounting for the suitability of the wavelength to the concentrations in the samples and accounting for any potential spectral interferences. The accuracy and reproducibility of the analyses were checked using two calcium-rich certified reference materials (CRMs), JLs-1 Limestone and JDo-1 Dolomite (both from

the Geological Survey of Japan). The reference materials were diluted to match the concentrations of Ca in the sample solutions. Ca, Mg and Sr concentrations were found to be within the uncertainty (1 SD) of the reported values (Imai et al. 1996).

The limits of quantification (LOQ) of the ICP-MS analysis were generally determined as the concentration corresponding to ten times the standard deviation of the signal obtained by analysing 0.8 M HNO₃ solution (6–7 times) in each individual run. ICP-MS was run in helium (He) mode (5 ml min⁻¹ He, 99.9995%

purity) for lighter elements (V, Mn, Cu, Y and Cd) to minimize the molecular interference from plasma and solution components and Ca from the samples.

The accuracy and reproducibility were checked by analyses of JLs-1 and JDo-1 before and after every batch of samples. The results obtained for all elements were within the uncertainty (2.5 SD) of the recommended

values (Imai et al. 1996). Accuracy of Pb determination cannot be checked using these CRMs because of the large spread of reference values probably due to insufficient homogeneity of Pb distribution in these samples. Based on the analyses of CRMs and matrix-matched solutions, the maximum analytical error for the typical range of concentrations in the shells can be estimated (in relative percentage) as 1.5% for Ca, Mg and Sr; 3% for Ba; 20% for Cu and U; and 4–10% for all other elements. More details on method validation were reported previously

(Piwoni-Piórewicz et al 2017).

**2.5 Statistical analyses**

To evaluate the effect of the shell size (ontogenesis stage) on the concentrations of trace elements in calcareous parts of *A. improvisus*, *C. glaucum*, *M. arenaria L. balthica* and *M. trossulus*, the concentrations of Ca, Na, Sr, Mg, Mn, Ba, Cu, Pb, V, Y, U and Cd were examined in the four size classes separately for each species. The data

were not normally distributed (Shapiro-Wilk test); therefore, significant differences between the mean concentrations of the selected elements in the size classes were identified by one-way Kruskal-Wallis nonparametric ANOVA (*p*-value = 0.05) and post-hoc Dunn's tests for multiple independent groups. Statistical computing and graphical visualizations were performed in RStudio.

## 3 Results

The concentrations of metals in studied individuals can be found in Table S1 in the Supplement. The mean concentrations in all studied organisms decreased in the following order Ca>Na>Sr>Mg>Mn>Ba>Cu>Pb>V>Y>U>Cd. However, the concentrations of given metals were different between shells of *Cerastoderma glaucum*, *Mya arenaria*, *Limecola balthica*, *Mytilus trossulus* and *Amphibalanus improvisus*, showing high variability (Table 3, Fig. 3).

There is a pattern of the highest concentration of Ca in all individuals, that ranges from $31,6000 \pm 26,000$ µg g$^{-1}$ (mean ± standard deviation) in *A. improvisus* to $36,3000 \pm 18,000$ µg g$^{-1}$ in *M. arenaria*. The rest of metals had mean concentrations in shells below 4,000 µg g$^{-1}$. In mussels most concentrated were Na>Sr>Mg, while in barnacles Mg>Na>Sr. The concentration of Na ranged, on average, from $2,000 \pm 300$ µg g$^{-1}$ in *M. trossulus* to $3,200 \pm 200$ µg g$^{-1}$ in *C. glaucum*. In *A. improvisus,* Mg ($3,940 \pm 320$ µg g$^{-1}$) was the dominant after Ca ($3,940 \pm 320$ µg g$^{-1}$), while mean concentration of Sr ($2,250 \pm 1,900$ µg g$^{-1}$) was lower than Na ($2,850 \pm 470$ µg g$^{-1}$). Clams *C. glaucum* (Mg: $80 \pm 30$ µg g$^{-1}$, Sr: $1,700 \pm 140$ µg g$^{-1}$, Na: $3,220 \pm 210$ µg g$^{-1}$), *M. arenaria* (Mg: $150 \pm 20$ µg g$^{-1}$, Sr: $2,160 \pm 280$ µg g$^{-1}$, Na: $3,080 \pm 210$ µg g$^{-1}$) and *L. balthica* (Mg: $80 \pm 30$ µg g$^{-1}$, Sr: $1,770 \pm 210$ µg g$^{-1}$, Na: $3,100 \pm 300$ µg g$^{-1}$) had concentrations of Sr over 15 times higher than those of Mg. Both metals were concentrated in shells of clams less than Na. Mussels *M. trossulus* were characterized by concentrations of Mg and Sr, reaching $1,060 \pm 170$ µg g$^{-1}$ and $1,170 \pm 100$ µg g$^{-1}$, respectively. This species also was characterized by concentration of Na ($2,010 \pm 310$ µg g$^{-1}$) higher than Mg and Sr (Table 3, Fig. 3).

The trace metals (Mn, Ba, Cu, Pb, V, Y, U, Cd) exhibit a general trend of highest concentrations in *A. improvisus* and *M. trossulus*, lower in *C. glaucum* and *L. balthica* and the lowest in *M. arenaria*. Mn was the most variable metal and increased from $1.4 \pm 1.0$ µg g$^{-1}$ in *M. arenaria* to $620 \pm 155$ µg g$^{-1}$ in *A. improvisus* (Table 3, Fig. 3).

The results of the Kruskal-Wallis nonparametric ANOVA test, which was used to compare the metal concentrations between the four size classes in each species, revealed the lack of variability within *M. arenaria*. The smallest variability was found in *L. balthica*. Only the concentration of Na ($H = 10.586$, $p = 0.014$) decreased with shell growth for this clam. The third clam species, *C. glaucum*, showed high variability of the four elements. Shells had increased concentrations of Sr ($H = 14.584$, $p = 0.002$), which was contrary to the decreasing concentration of Na ($H = 10.529$, $p = 0.015$), Mn ($H = 10.658$, $p = 0.014$) and Cd ($H = 11.655$, $p = 0.009$). Similarly, shells of *A. improvisus* also showed variability of four elements between size classes, namely, Mg ($H = 11.996$, $p = 0.007$), V ($H = 11.206$, $p = 0.011$), Cu ($H = 9.146$, $p = 0.027$) and Pb ($H = 13.308$, $p = 0.004$). However, in this case, the sequences of the changes were not straightforward but, rather, had a tendency to fluctuate between the smallest and the largest individuals. The highest variability was found within the bimineralic shells of *M. trossulus*. The size classes differed in terms of five elements. The incorporation of V ($H = 22.595$, $p < 0.001$), Cu ($H = 26.18$, $p < 0.001$), Y ($H = 10.819$, $p = 0.013$), Cd ($H = 15.353$, $p = 0.002$) and U ($H = 18.202$, $p < 0.001$) into shells decreased in larger mussels (Fig. 4).

Detailed analyses of the differences in the studied metals between the size classes based on post-hoc Dunn's tests for multiple independent groups indicated that the significant variations were not linear (Fig. 4). In *L. balthica*, Na concentration decreased in larger shells, showing differences between the size classes I and III and I and IV. In the shells of *C. glaucum*, Sr concentration increased gradually, reaching a peak in the size class IV,

while statistically significant differences were observed between the size classes I and IV and the III and IV. An inverse pattern was observed for Na, yet its concentration differed only between the smallest and largest clams. Shells were also characterized by a common trend of the Mn and Cd, which decreased from I size class and later reached a plateau. In shells of *A. improvisus,* the concentrations of Mg, V, Cu and Pb decreased in larger individuals. The levels of Mg in shells statically differed between the size classes III and IV; likewise, shells from the size class III had the highest concentrations. Metals V and Pb occurred at the highest concentrations in shells of the smallest individuals, and later, they oscillated around a similar mean value. Cu decreased in growing shells of barnacles, reaching the minimum in the size class III. There were no statistically significant differences between the classes III and IV, and the oldest half of the group maintained the downward trend. However, it is worth noting that Cu concentrations in many of the oldest shells are several times higher. Shells of *M. trossulus* were characterized by the highest variability of trace element concentrations between size classes. The trend of decreasing concentrations was clearly marked for V, Cu, Cd and U. The concentration of Y decreased in larger shells, but the oldest shells showed an upward trend; therefore, significant differences were found between the size classes I and III (Fig. 5).

The comparison of elemental concentrations in shells of mussels and barnacles from different regions, based on literature data is presented in Table 4.

## 4 Discussion

### 4.1 CaCO$_3$ polymorph type and elemental concentrations

The average Ca content of all species collected in the Baltic Sea was found to be 343,200 μg g$^{-1}$ (Table 3), which corresponds to c. 86% (w/w) of pure CaCO$_3$. For comparison, same species or congeners from different regions contain between 310,000 and 420,000 μg g$^{-1}$ of Ca in shell (Table 4), the results above 400,000 μg g$^{-1}$ probably being caused by analytical bias.

Among shell impurities, the most concentrated metals were Na, Sr, Mg in all studied CaCO$_3$ polymorphs (Fig. 3). These compositions are considered to be typical in calcareous skeletons of marine invertebrates, as those elements are energetically favoured in CaCO$_3$ crystal lattice, substituting for Ca (Allison et al., 2001; Iglikowska et al., 2016; Reeder, 1983; Sugawara and Kato, 2000; Wang and Xu, 2001, see also Table 4). The main elemental constituents of the Baltic shells showed generally uniform distribution in collected samples, and within the ranges typical for a given species. Clams, that precipitate fully aragonitic shells and bimineralic mussels were characterized by the following order of accumulated concentrations: Na>Sr>Mg, while in barnacles this order was different: Mg>Na>Sr (Fig. 3). The most evident observation was that the aragonitic shells of *C. glaucum*, *M. arenaria* and *L. balthica* contained over 15 times more Sr than Mg. In the calcitic *A. improvisus,* Mg was 1.8 times more concentrated than Sr, while the bimineralic shells of *M. trossulus* containing layers of calcite and aragonite, were distinguished by equalized concentrations of Mg and Sr in shells (Table 3, Fig. 3). Such pattern of Mg and Sr is observed in number of calcareous species including bivalves and barnacles (Table 4; Dalbeck, 2008; Iglikowska et al., 2016; Wang and Xu, 2001; Zhao et al., 2017). However, by comparing species, it is evident that the calcite of barnacles *A. improvisus* deviated from this general trend. Out of the species studied in this work, both Mg and Sr concentrations were higher in calcitic barnacles than in aragonitic clams (Table 3, Fig. 3). Kinetic and biological effects influence the partitioning of Sr between the shell and seawater (Urey et al., 1951), and Sr in

shells is known to significantly exceed its concentration expected at the thermodynamic equilibrium (Schöne et al., 2010). Furthermore, previous studies revealed that barnacles have significantly higher Sr shell concentration (up to 2,300 μg g$^{-1}$) than other marine invertebrates (Carpenter and Kyger, 1992; Ullmann et al., 2018). The concentration of Sr in shells of *A. improvisus* from this study (2,250 ± 190 μg g$^{-1}$) supports these findings, suggesting that barnacle precipitation process differs from other marine organisms. Carpenter and Kyger (1992) concluded that Sr is a sensitive indicator of precipitation rate this may explain the elevated Sr concentrations in barnacles. The shell increment in barnacles in rapid (15 – 30 mm per year, Milliman, 1974) due to their relatively short lifespan (approximately one year). Consequently, the observed Sr concentration of *A. improvisus* may be caused by higher precipitation rate of barnacle calcite in comparison with the precipitation rate of aragonite in long-lived bivalves (Tables 3 and 4, Fig. 3). Crystal lattice distortions in barnacle calcite induced by rapid incorporation of impurities (Pokroy et al., 2006) may allow for even higher Sr incorporation and support the high Sr concentration in *A. improvisus* shell. This indicates that Mg and Sr concentration in shell are controlled by CaCO$_3$ lattice properties, yet in a strongly species-specific way (Skinner and Elderfield, 2005).

The concentrations of Mn and Ba are several orders of magnitude higher in the calcitic shells of barnacle *A. improvisus* than in other species (Table 3, Fig. 3). It is expected that aragonite shells would incorporate Ba more intensively than would calcitic shells (Findlater et al., 2014; Gillikin et al., 2006), yet this trend is not observed in this study. Therefore, the pattern observed in this investigation is not related to the crystal lattice orientation (Markulin et al., 2019; Poulain et al., 2015). Barnacles appear to have a stronger capacity for Mn incorporation into shells than other benthic calcifiers (Pilkey and Harris, 1966), possibly due to higher growth rate or specific shell organic matter (Bourget, 1974). At one station (GN, Fig. 2), we observed the highest concentration of Mn and Ba in the shells, and this pattern might be determined by the environmental parameters at that location. Yet, shells of both species with the highest concentration of Mn (*A. improvisus, M. trossulus*) contained calcite, therefore we should not rule out that the polymorph type of CaCO$_3$ is regulating to some degree the level of shell Mn in low salinity environment.

It was observed that a great incorporation of Mg and Sr into shell can contribute to distortion of CaCO$_3$ crystal lattice and results in parallel increased incorporation of trace elements into the shells (Harriss, 1965; Davis et al., 2000; Finch and Allison, 2007; Dalbeck, 2008;).  In our study shell lattices of mussels and barnacles could be changed through the concentration of Mg and Sr higher than in clams. Therefore, the concentration of Cu, V, Cd, Y and U seems to be driven to some degree by crystal lattice properties having the highest values in species containing calcite in their shells (Fig. 3). It is also important to note that the smaller ionic radii of V, Cd, Y, U and Cu are energetically favoured in calcite, while larger Pb, in aragonite structure (Morse et al., 1997; Reeder, 1983; Wang and Xu, 2001). However, trace metals Cu, V, Cd, Y were present at the highest levels in bimineralic *M. trossulus*, while only U, as well as Pb, in calcitic *A. improvisus* (Table 3, Fig. 2), which suggests that additional factors, other than the crystal lattice effects, determines those concentrations. Furthermore, this study likewise revealed inconsistent variability of trace metal concentrations between aragonitic clams. Out of the three species of clams, *M. arenaria* was characterized by the lowest concentration of all trace metals (Table 3, Fig. 2). This also indicates that factors other than mineral properties co-regulate trace metals accumulation in shells.

This study considers metal concentrations in shells that were not subjected to chemical removal of organic matter prior to the dissolution of the carbonate matrix. Chemical cleaning of carbonate skeletons prior to chemical analysis and both improvement on the data quality and potential artefacts associated with this are widely discussed

in the literature (Barker et al., 2003; Holcomb et al., 2015; Loxton et al., 2017), but the plausible pre-treatment method for the removal of organics still needs to be found (Inoue et al., 2004). Mannella et al. (2020) showed that the suitability of chemical pre-treatments for organic matter removal from carbonate matrices should be evaluated on a case-by-case basis and, in case of relatively low organic content, should be avoided. In addition to $CaCO_3$ crystal lattice, shells of barnacles and bivalves contain up to 5.0 wt% of organic matter (Bourget, 1987; Marin and Luquet, 2004; Rueda and Smaal, 2004). Therefore, the specific features and composition of given organic matrix might be a partial cause of the inter-species and inter-individual variability of elemental concentration in shells (Fig. 3, Table 3; Takesue and van Geen, 2004). The organic fraction was found to be generally not associated with significant level of trace metals (Lingard et al., 1992; Takesue et al., 2008) and Sr, which is strongly incorporated into the crystal phase (Takesue and van Geen, 2004; Walls et al., 1977). In contrast, some authors found correlations indicating Mg and Mn as often associated with the shell organic matrix (Lorens et al., 1980; Rosenberg et al., 2001; Takesue and van Geen, 2004). Despite competition with $Ca^{2+}$ during biomineralization, they are biologically essential elements (Bellotto and Miekeley, 2007) that are incorporated into the organic matrix within precipitated $CaCO_3$, especially during rapid shell growth in *A. improvisus*. Therefore, high inter-species variability of Mg and Mn between barnacles, mussels and clams (Fig. 3) could be enhanced by specific properties of organic phases.

**4.2 Size classes and potential biological impact on elemental concentrations**

The recorded concentrations of metals in all populations exhibited marked variability among individuals (Table 3, Fig. 3), a feature previously recorded by several authors (Gillikin et al., 2005a; Vander Putten et al., 2000). In this study, individuals were collected over a wide range of sizes (Table 2), representing different ages and life span. Bivalves are long-living organisms, and these of the Gulf of Gdańsk have life expectancy of 4 – 12 years (Gofas, 2004; Żmudziński, 1990), contrary to barnacles with the relatively short lifespan of approximately one year (Bornhold and Milliman, 1973). The southern Baltic Sea is driven by cyclical environmental dynamics, which evoke physiological stress, determine the food base and drive biogeochemical cycle (Elder and Collins, 1991). Shells of *A. improvisus* were produced in the shortest period of time and experienced less abiotic variations than bivalves. Thus, life span to some extent may explain the lowest variability of elements in barnacles (Fig. 3).

Thébault et al. (2009) revealed low inter-individual elemental variability in bivalves and on this basis indicated the environment as a factor controlling their incorporation within shells. The heterogeneous distribution of the elemental concentrations in organisms from the brackish Gulf of Gdańsk may to some extent be caused by fluctuating biological factors that affect shell precipitation, and the chemistry of the shells could deviate from what is expected with purely environmental control. The biological influence on the shell chemistry in the southern Baltic Sea could be reinforced by unfavourable conditions for calcification. The low salinity (~7) and alkalinity, which is typical for the studied area of the Gulf of Gdańsk, cause a reduced $CaCO_3$ saturation state (Beldowski et al., 2010; Cai et al., 2010; Findlay et al., 2008). $Ca^{2+}$ and $CO_3^{2-}$ are essential components for the crystal formation, and when their concentrations in seawater are low, calcifying organisms exert selective $Ca^{2+}$ channels to enable an active ion capture from solution (Sather and Mccleskey, 2003). The required higher contribution of $Ca^{2+}$ active pumping results in greater degree of biological control over the calcification process (Sather and Mccleskey, 2003; Waldbusser et al., 2016) and shells are not produced in equilibrium with environmental conditions when it comes to an elemental concentration.

Within species, organisms from juveniles to adults experience morphological and functional changes related to sex, metabolic rate or reproductive stage, which complicate the biomineralization process (Carré et al., 2006; Freitas et al., 2006; Gillikin et al., 2005b; Schöne et al., 2010, 2011; Warter et al., 2018). The size-related elemental patterns in shells of *A. improvisus*, *C. glaucum*, *M. arenaria*, *L. balthica* and *M. trossulus* from the Gulf of Gdańsk indicate that if the significant variability exists, it is specifically expressed in trace metal concentrations.

The studied mussel *M. trossulus* and barnacle *A. improvisus* showed the greatest variability between size classes, while it was less pronounced in clams (Fig. 4). However, the varied elements were not uniform for these species, even though the organisms came from one location (Table 1). The size-related trend was observed for V, Cu, Y, Cd and U in molluscs, and Mg, V, Cu and Pb in barnacles (Fig. 3). Among clams, we found the lack of size-dependent changes within *M. arenaria*. In *L. balthica* only the concentration of Na decreased with shell growth,

while *C. glaucum* showed variability of Sr, Na, Mn and Cd (Fig. 4). Different patterns of metal accumulation for species in the same habitat were observed before (Rainbow, 2002, 1995). Rainbow et al. (2000) tested *A. improvisus* and *M. trossulus* from the Gulf of Gdańsk as environmental biomonitors by measuring the concentrations of Co, Zn, Fe, Cd, Pb, Mn and Ni in soft tissues. They found that mussels and barnacles occurring at the same location did not show the same variation in metal bioavailabilities, probably because barnacles were

particularly strong accumulators of trace metals (Rainbow, 2002, 1998). This shows that biological differences between species, such as feeding rate, assimilation efficiency (Luoma and Rainbow, 2005), route and degrees of metal uptake (Rainbow and Wang, 2001) are significant factors determining the elemental accumulation in shells.

        In this study, it was generally observed that, when statistical differences between size classes were recorded, the concentrations of metals decreased with the shell size. The reverse was found only for Sr in the shells

of *C. glaucum*, in which the concentration of Sr increased with size (Figs. 4 and 5). Large mussels pump less water per unit of the body weight, and their uptake of metals is lower than that in smaller individuals. When the concentrations of elements decrease with increasing shell size (Fig. 4), the incorporation might depend on the growth rate. The younger specimens could have a greater growth rate and shell precipitation rate, resulting in a greater uptake of trace elements (Dalbeck, 2008; Szefer et al., 2002). Rosenberg and Hughes (1991) suggested that

areas of higher shell curvature, such as the umbo, require greater metabolic expenditure, resulting in an increase of metal uptake. When the metabolic activity of organism decreases, the ionic flux likewise decreases, increasing the tendency of $Ca^{2+}$ to block other ions fluxes (Carré et al., 2006; Friel and Tsien, 1989). Therefore, in a low salinity environment of the Gulf of Gdańsk, metabolic fluctuations of organisms can have an exceptionally strong effect on elemental variability, which was high among studied individuals (Table 3, Fig. 3). The surface-to-volume

ratio decreases with size and affects the contribution of the adsorbed element content to the bulk concentration (Azizi et al., 2018). Therefore, the negative correlation between the bulk elemental concentration and shell size (Fig. 4), noted in some previous studies (Martincic et al., 1992; Piwoni-Piórewicz et al., 2017; Ritz et al., 1982), could have been caused by a greater potential of surface adsorption in smaller individuals. This is most pronounced for a few trace elements, the concentrations of which decreased across the four size classes in shells of *A.*

*improvisus* (V, Cu, Pb), *M. trossulus* (V, Cu, Y, Cd, U) and *C. glaucum* (Mn, Cd, Figs. 4 and 5). Catsiki et al. (1994) suggested that, apart from metabolic processes, an active detoxification mechanism in tissues is responsible for this trend, and its efficiency is higher in older and larger individuals.

        Nevertheless, many of the elements studied herein showed a lack of statistically significant relationships between the shell sizes. Few trace elements in the shells of *A. improvisus* (Y, U and Cd), *M. trussulus* (Pb), *C.*

*glaucum* (V, Cu, Y, Pb and U) and all trace metals in the shells of *L. balthica* and *M. arenaria* (V, Cu, Y, Pb, U and Cd) showed no significant variability related to the size of organisms (Figs. 4 and 5). This is not an unusual pattern for marine invertebrates, which has been shown by a number of studies. Saavedra et al. (2004) observed no differences between Cd, Pb, Cr, Ni, As, Cu and Zn concentrations for different shell lengths of the raft *Mytilus galloprovincialis* separated into four size classes. Protasowicki et al. (2008) similarly found that the concentrations

of Hg, Pb, Cd, Cu, Zn, Cr, Ni, Fe, Mn, V, Li and Al in the shells of the mussel *Mytilus edulis* from the Polish coast of the Baltic Sea did not vary between shell sizes. This inconsistent relationship shows that trace metals concentration in shells of different sizes might be under influence of a number of factors including species-specific biological mechanisms (e.g. metabolic rate).

**4.3 Environmental factors and elemental concentrations**

Organisms derive elements in dissolved and particulate forms primarily from surrounding water, sediments and their food base (Freitas et al., 2006; Gillikin and Dehairs, 2005; Poulain et al., 2015). The concentration of basic macro elements Na, Mg and Sr in surrounding seawater is generally proportional to salinity (Beldowski et al., 2010; Cai et al., 2010; Findlater et al., 2014; Wit et al., 2013), therefore the environmental level of these metals

should be rather homogenous in the study area (salinity range 6.9 – 7.3, Table 1). And indeed, our analysis has shown low variability of the concentrations of Na ($2{,}010 \pm 310$ µg g$^{-1}$ – $3{,}220 \pm 210$ µg g$^{-1}$) and Sr ($1{,}170 \pm 100$ µg g$^{-1}$ – $2{,}250 \pm 190$ µg g$^{-1}$, Table 3) in the shells of studied species, which might reflect environmental stability. However, the concentration of Mg in shells as this element was highly variable ($3{,}940 \pm 320$ µg g$^{-1}$ – $80 \pm 30$ µg g$^{-1}$, Table 3, Fig. 3). Although the salinity was indicated as the factor controlling the concentration of Na, Sr and

Mg in environment and skeletons built there (Elderfield and Ganssen, 2000; Rosenheim et al., 2004), the range of surface salinity in the study region was unlikely to explain the observed range of Mg concentrations. Therefore, the concentration of Mg seems to not be dependent on the environment, but likely to have be influenced by physiological or kinetic factors. Vander Putten at al. (2000) observed that the skeletal Mg/Ca variations in *M. edulis* could not be interpreted solely based on variations in the seawater Mg. Furthermore, Dodd and Crisp (1982)

showed that the Mg/Ca ratios of most estuarine waters only differ significantly from the open-ocean ratios at salinities below 10, and variations in the shell Mg/Ca ratio must be caused by factors not linked directly with the environment.

Among trace elements (Mn, Ba, Cu, Pb, V, Y, U and Cd), Mn and Ba were definitely the most concentrated in shells (Fig. 3), as previously reported for bivalves (Lazareth et al., 2003; Vander Putten et al.,

2000). Such high concentrations of Mn and Ba are usually associated with freshwater inputs to estuarine systems, which likewise causes phytoplankton blooms (Gillikin et al., 2006; Vander Putten et al., 2000; Thébault et al., 2009). Consequently, Ba and Mn may be taken up by calcifiers through ingestion of phytoplankton (Stecher et al., 1996) or decaying algal flocs (Brannon and Rao, 1979; Stecher et al., 1996). The elemental incorporation into shell has been shown to correlate at species level with changes in primary production and phytoplankton blooms

(Freitas et al., 2006; Lazareth et al., 2003; Vander Putten et al., 2000). Therefore, the food base and elemental transport in the trophic chain might cause the spatial and species-specific elemental variability in shells (Fig. 3). This pattern is generally confirmed in different regions by the contents of Mn and Ba relative to other trace elements (Fig. 4).

Seawater trace element concentrations in the study area depend on seasonality and human activity, but elemental concentrations in sediments represent long-term chemical background. In coastal regions, trace elements discharged from various sources, such as atmosphere, rivers, and plankton blooms, can be rapidly transported from the water column to the bottom sediments (Bendell-Young et al., 2002; Szefer et al., 1995). In this study, samples were collected in two regions of the Gulf of Gdańsk: the outer part of the Puck Bay (stations MA, M2 and MW) and central part of the Gulf of Gdańsk (station GN, Fig. 1). The shells of clams collected in the outer Puck Bay, with sandy sediment (Szefer et al., 1998; Szefer and Grembecka, 2009; Uścinowicz, 2011), were depleted in trace elements. On the other hand, mussels and barnacles from the central Gulf of Gdańsk, with sandy mud, were characterized by higher elemental concentrations than clams (Fig. 3). On this basis we can assume that the concentrations of trace metals Mn, Ba, Cu, Pb, V, Y, U and Cd in shells might be related to sediment granulometry and increase from sandy to silty type (Góral et al., 2009; Uścinowicz, 2011). The maximum concentration of trace elements within a given region is commonly associated with the finest sediment fraction (<2 μm) as compared to the sandy fraction (Kim et al., 2004; Szefer et al., 1998). In addition, the formation of Mn-Fe oxyhydroxides in the surface sediments (Glasby and Szefer, 1998; Szefer et al., 2002) has a particular role in absorbing trace metals in the fine sediments (Pruysers et al., 1991; Szefer et al., 1995; Tessier et al., 1979). It has been previously shown that for mussel *M. trossulus*, clam *L. balthica* and barnacle *A. improvisus* from the central part of the Gulf of Gdańsk (near the GN station), elemental concentrations in shells (Szefer and Szefer, 1985) and tissues (Rainbow et al., 2000, 2004; Sokolowski et al., 2001) were similar to that in the Vistula River plume and higher than metal concentrations in macrozoobenthos from the outer Puck Bay.

However, the studied metals exhibited variations in accumulated concentrations between species, both within the Puck Bay (M2: *C. glaucum*, MA: *M. arenaria*, MW: *L. balthica*) and within the central Gulf of Gdańsk (GN station: *M. trossulus*. and *A. improvisus*, Fig. 3). The highest concentration of trace metals in *C. glaucum* (M2: 10 m) and *L. balthica* (MW: 31 m) within clams from the sandy Puck Bay could be driven by elevated amounts of trace elements in oxygenated zones where Fe-Mn oxyhydroxides accumulate. At the Mn(II)/Mn(IV) redox interface manganese oxides may predominantly precipitate on the periostracum of molluscs comparing to inorganic surfaces (Strekopytov et al., 2005), which may, in turn, influence the incorporation of trace elements into shells. *M. arenaria* likewise was collected in shallow zone (MA: 10 m), but had, nevertheless, the lowest concentrations of all studied trace elements (Fig. 3). This bivalve mainly spends life buried 20 – 30 cm in a sediment (Żmudziński, 1990). Szefer et al. (1998) reported that in the Gulf of Gdańsk the enrichment factors for Cu, Zn, Ag, Cd and Pb are highest in the <2 μm fraction and decrease with increasing both fraction size and depth of the sediment. This may by a reason why *C. glaucum* and *L. balthica* have similar patterns of elemental accumulation, while *M. arenaria* was characterized by the lowest values (Fig. 3). Such dependence indicates that sediment could be a factor controlling concentrations of trace metals in the shells.

Mussels *M. trossulus* and barnacles *A. improvisus* from the same location (Fig. 1, Table 1) showed greater chemical differentiation than clams. Much lower concentrations of Mn and Ba were found in the shells of molluscs (Mn: $54.0 \pm 15.0$ μg g$^{-1}$, Ba: $17.0 \pm 6.70$ μg g$^{-1}$) than in barnacles (Mn: $625 \pm 160$ μg g$^{-1}$, Ba: $73.0 \pm 20.1$ μg g$^{-1}$, Table 3). A similar relationship was found in the soft tissues of *M. trossulus* and *A. improvisus* collected from different locations in the Gulf of Gdańsk in May 1998 (Rainbow et al., 2000). The range of Mn in the soft tissues of *M. trossulus* varied from 19.0 to 41.0 μg g$^{-1}$, while that in *A. improvisus* ranged from 187 to 307 μg g$^{-1}$ and was interpreted as species-specific accumulation efficiency. This trend is not noticeable to some trace elements (Cu,

V, Y and Cd), which were not the most concentrated in barnacles (Fig. 3). This was the most likely caused by differences in bioaccumulation strategy between mussels and barnacles and the trace elements deposited over time in the shells are determined by environmental and biological conditions. As this study was conducted in a highly dynamic costal system, notable elemental variability between mussels and barnacles is associated with fluctuating environmental conditions. The *A. improvisus* shell is a record of annual variability only compared to the *M. trossulus* shell, which records many years of fluctuations of environmental parameters.

Some studies have investigated trace element concentrations in shells (but many more have concentrated on the soft tissues) of marine invertebrates as a tool to assess trace metal contamination of the aquatic environment. The concentrations of trace elements Mn, Ba, Cu, Pb, Cd and U in the shells of a studied organism in Gulf of Gdańsk and in other regions was found to be highly variable, even within a single taxon (Table 4). Therefore, the environmental conditions prevailing during biomineralization are largely reflected in the trace element concentrations of the shells; nevertheless, their interpretation requires consideration of biological factors specific to the species.

## 5. Conclusions

The shells of calcitic *Amphibalanus improvisus*, aragonitic *Cerastoderma glaucum*, *Limecola balthica*, *Mya arenaria* and bimineralic *Mytilus trossulus* from the Gulf of Gdańsk are accumulators of a wide spectrum of elements from the surrounding environment. The elemental concentration levels in studied species are not only determined by their bioavailability in the environment. Many biotic and abiotic factors are acting on the shell incorporation mechanism and their effect is likely to be species-specific.

By determining Ca, Na, Sr, Mg, Mn, Ba, Cu, Pb, V, Y, U and Cd in the shells of given species we found some patterns of metal accumulation. At a local scale of the Gulf of Gdańsk, the main elements Na, Sr and Mg are mostly dependent on crystal lattice properties of calcite and aragonite. Clams that precipitate fully aragonitic shells have a clear preference for accumulating Sr over Mg in shells, contrary to dominant Mg content over Sr in barnacle shell calcite. It is energetically more favourable for larger cations such as Na and Sr to enter the aragonite lattice with smaller cations (e.g. Mg) favouring calcite. However, this relationship breaks down when comparing shells of different species or genera. For example, the barnacle calcite contains higher Sr concentration than the bivalve aragonite. The level of main elements, especially Sr and Mg, seems to be determined by specific biological features, such as growth and precipitation rate.

In case of trace metals Mn, Ba, Cu, Pb, V, Y, U and Cd, factors other than given crystal lattice presence, seem to determine their concentrations. The elemental variability between size-grouped shells indicates that trace elements were more variable than Na, Sr, Mg, but this varies between species. Moreover, there is a trend of the elemental concentrations being lower in larger than in smaller shells. Biological differences between and within species, such as feeding including its rate and assimilation efficiency related to age of organisms (size of the shell), are potentially important factors determining the elemental accumulation in shells.

Given that the specimens were obtained from two regions of the Gulf of Gdańsk: the outer part of the Puck Bay and central part of the Gulf of Gdańsk, an impact of local environmental factors, such as the sediment type and the food base cannot be excluded as factors controlling concentration of trace elements in shells.

On this basis, it can be emphasized that by analysing the species-specific biological factors in parallel with the elemental variability of shells within widely distributed species, it might be possible to separate the environmental and biological signals in the biomineralization process.

**Author contribution.** APP and PK designed and led the study. SS and EHW conducted elemental analysis. APP prepared the original draft, which was reviewed and edited by all co-authors.

**Competing interests.** The authors declare that they have no conflict of interest.

**Data availability.** The underlying research data can be accessed at Institute of Oceanology, Polish Academy of Sciences, Powstańców Warszawy 55, 81-712 Sopot, Poland.

**Supplement.** The supplement related to this article will be available online.

**Acknowledgements.** We would like to thank Dr Halina Kendzierska from the University of Gdańsk for cooperation during sampling, and Radosław Brzana and Jerzy Abramowicz for photos of studied species presented in Figure 1.

**Financial support.** The research leading to these results received funding from the Polish National Science Centre
in the frame of project contracts LOGGER/2017/25/N/ST10/02305 and PANIC/2016/23/B/ST10/01936.

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

**Tables and Figures**

**Table 1.** Details of the sampling locations. Temperature (T) and salinity were measured near the bottom during sample collection.

| Station | Coordinates | Collected species | Depth (m) | T (°C) | Salinity | Sediment type[a] | Sampling date |
|---|---|---|---|---|---|---|---|
| | | | | measured in situ | | | |
| GN | 54°32'N 18°48'E | *Amphibalanus improvisus* | 36 | 3.1 | 7.3 | Sandy mud, silty sand | May 2013 |
| GN | 54°32'N 18°48'E | *Mytilus trossulus* | 36 | 3.1 | 7.3 | Sandy mud, silty sand | May 2013 |
| M2 | 54°38'N 18°33'E | *Cerastoderma glaucum* | 10 | 16.9 | 6.9 | Sand | June 2014 |
| MA | 54°37'N 18°32'E | *Mya arenaria* | 10 | 19.8 | 6.9 | Sand | June 2014 |
| MW | 54°37'N 18°37'E | *Limecola balthica* | 31 | 4.7 | 7.0 | Fine sand, sandy mud, silty sand | June 2014 |

[a] Based on Uścinowicz (2011)

**Table 2.** The range of shell sizes for each studied species within a given size class.

| Species | Mineral type | Size classes (mm) | | | | Maximum size in the Gulf of Gdańsk (mm)[a] |
|---|---|---|---|---|---|---|
| | | I | II | III | IV | |
| *Amphibalanus improvisus* | Calcite | 3–4 | 5–6 | 7–8 | 9–10 | 10 |
| *Mytilus trossulus* | Bimineralic | 6–5 | 16–25 | 26–35 | 36–44 | 50 |
| *Cerastoderma glaucum* | Aragonite | 4–8 | 9–12 | 12–16 | 16–20 | 27 |
| *Mya arenaria* | Aragonite | 10–20 | 20–30 | 30–40 | 40–49 | 70 |
| *Limecola balhtica* | Aragonite | 4–7 | 8–11 | 12–15 | 15–18 | 24 |

[a] Based on Żmudziński (1990)

 **Table 3.** Elemental concentrations in shells of studied organisms, presented in decreasing order of means for each species (SD = standard deviation, SE = standard error).

**Barnacle *Amphibalanus improvisus*: CALCITE**

| Element (El) | N | Mean conc. [µg g$^{-1}$] | ±1SD | ±1SE | Weight ratio El/Ca [arbitrary] | Molar ratio El/Ca [mmol/mol] |
|---|---|---|---|---|---|---|
| Ca | 37 | 316,000 | 26,000 | 4,300 | 1 | 1 |
| Mg | 37 | 3,940 | 320 | 100 | 0.012 | 20.5 |
| Na | 37 | 2,850 | 470 | 100 | 0.009 | 15.7 |
| Sr | 37 | 2,250 | 190 | 30 | 0.007 | 3.3 |
| Mn | 37 | 625 | 160 | 26.31 | 1.978 | 1438.4 |
| Ba | 37 | 73.6 | 20.1 | 3.30 | 0.231 | 67.3 |
| Cu | 37 | 3.90 | 2.2 | 0.36 | 0.012 | 7.7 |
| Pb | 37 | 1.34 | 0.79 | 0.13 | 0.0042 | 0.82 |
| V | 36 | 0.47 | 0.156 | 0.03 | 0.0015 | 1.16 |
| Y | 37 | 0.21 | 0.12 | 0.02 | 0.0007 | 0.30 |
| U | 37 | 0.11 | 0.043 | 0.01 | 0.0003 | 0.06 |
| Cd | 37 | 0.06 | 0.02 | 0.003 | 0.0002 | 0.07 |

**Clam Cerastoderma glaucum: ARAGONITE**

| Element (El) | N | Mean conc. [µg g$^{-1}$] | ±1SD | ±1SE | Weight ratio El/Ca [arbitrary] | Molar ratio El/Ca [mmol/mol] |
|---|---|---|---|---|---|---|
| Ca | 40 | 348,000 | 16,000 | 2,500 | 1 | 1 |
| Na | 40 | 3,220 | 210 | 30 | 0.009 | 16.1 |
| Sr | 40 | 1,700 | 140 | 20 | 0.005 | 2.2 |
| Mg | 40 | 80 | 30 | 5 | 0.0002 | 0.4 |
| Ba | 40 | 15.0 | 10 | 1.61 | 0.043 | 12.6 |
| Mn | 40 | 11.9 | 10.5 | 1.66 | 0.034 | 24.9 |
| Cu | 40 | 4.08 | 6.29 | 0.99 | 0.012 | 7.3 |
| Pb | 34 | 0.14 | 0.1 | 0.02 | 0.0004 | 0.08 |
| U | 40 | 0.10 | 0.084 | 0.01 | 0.0003 | 0.05 |
| Y | 40 | 0.07 | 0.074 | 0.01 | 0.0002 | 0.09 |
| V | 37 | 0.07 | 0.061 | 0.01 | 0.0002 | 0.16 |
| Cd | 37 | 0.01 | 0.009 | 0.002 | 0.00003 | 0.01 |

**Clam *Limecola balthica*: ARAGONITE**

| Element (El) | N | Mean conc. [µg g$^{-1}$] | ±1SD | ±1SE | Weight ratio El/Ca [arbitrary] | Molar ratio El/Ca [mmol/mol] |
|---|---|---|---|---|---|---|
| Ca | 40 | 345,000 | 17,000 | 2,780 | 1 | 1 |
| Na | 40 | 3,100 | 300 | 50 | 0.009 | 15.6 |
| Sr | 40 | 1,770 | 210 | 30 | 0.005 | 2.3 |
| Mg | 40 | 80 | 30 | 4 | 0.0002 | 0.4 |
| Ba | 40 | 13 | 10 | 1.686 | 0.038 | 11.0 |

| | | | | | | |
|---|---|---|---|---|---|---|
| Mn | 40 | 7.9 | 7.1 | 1.024 | 0.023 | 16.7 |
| Cu | 40 | 3.0 | 3.4 | 0.547 | 0.009 | 5.43 |
| Pb | 39 | 0.14 | 0.16 | 0.026 | 0.0004 | 0.08 |
| Y | 40 | 0.08 | 0.11 | 0.017 | 0.0002 | 0.10 |
| V | 35 | 0.04 | 0.031 | 0.005 | 0.0001 | 0.08 |
| U | 40 | 0.03 | 0.024 | 0.004 | 0.0001 | 0.02 |
| Cd | 37 | 0.01 | 0.008 | 0.001 | 0.00002 | 0.01 |

**Clam *Mya arenaria*: ARAGONITE**

| Element (El) | N | Mean conc. [µg g$^{-1}$] | ±1SD | ±1SE | Weight ratio El/Ca [arbitrary] | Molar ratio El/Ca [mmol/mol] |
|---|---|---|---|---|---|---|
| Ca | 40 | 363,000 | 18,159 | 2,870 | 1 | 1 |
| Na | 40 | 3,080 | 207 | 33 | 0.0085 | 14.8 |
| Sr | 40 | 2,160 | 280 | 44 | 0.0060 | 2.72 |
| Mg | 40 | 150 | 25 | 4 | 0.0004 | 0.68 |
| Ba | 40 | 2.60 | 1.6 | 0.249 | 0.00716 | 2.1 |
| Mn | 40 | 1.40 | 1 | 0.160 | 0.00386 | 2.8 |
| Cu | 40 | 0.031 | 0.024 | 0.004 | 0.00009 | 0.053 |
| Y | 40 | 0.020 | 0.018 | 0.003 | 0.00006 | 0.025 |
| V | 40 | 0.016 | 0.014 | 0.002 | 0.00004 | 0.035 |
| Pb | 40 | 0.015 | 0.014 | 0.002 | 0.00004 | 0.008 |
| U | 40 | 0.006 | 0.005 | 0.0008 | 0.00002 | 0.003 |
| Cd | 37 | 0.002 | 0.003 | 0.0005 | 0.00001 | 0.002 |

**Mussel *Mytilus trossulus*: CALCITE and ARAGONITE**

| Element (El) | N | Mean conc. [µg g$^{-1}$] | ±1SD | ±1SE | Weight ratio El/Ca [arbitrary] | Molar ratio El/Ca [mmol/mol] |
|---|---|---|---|---|---|---|
| Ca | 40 | 344,000 | 28,000 | 4,440 | 1 | 1 |
| Na | 21 | 2,010 | 310 | 70 | 0.006 | 10.2 |
| Sr | 41 | 1,170 | 100 | 10 | 0.003 | 1.6 |
| Mg | 40 | 1,060 | 170 | 30 | 0.003 | 5.1 |
| Mn | 35 | 54 | 15 | 2.511 | 0.157 | 114.16 |
| Ba | 38 | 17.0 | 6.7 | 1.084 | 0.049 | 14.40 |
| Cu | 40 | 14.0 | 10 | 1.592 | 0.041 | 25.44 |
| Pb | 32 | 1.00 | 1.4 | 0.240 | 0.003 | 0.56 |
| V | 40 | 0.78 | 0.52 | 0.082 | 0.002 | 1.78 |
| Y | 40 | 0.70 | 1.2 | 0.187 | 0.002 | 0.91 |
| Cd | 40 | 0.09 | 0.11 | 0.017 | 0.0003 | 0.09 |
| U | 40 | 0.05 | 0.033 | 0.005 | 0.0002 | 0.03 |

**Table 4.** Elemental concentrations in shells from different regions based on this study and literature data.

| Element concentration in shell | Ca | Na | Sr | Mg | Mn | Ba | Cu | Pb | Cd | U |
|---|---|---|---|---|---|---|---|---|---|---|
| | [µg g$^{-1}$] | | | | [µg g$^{-1}$] | | | | | |

| Species | | | | | | | | | | |
|---|---|---|---|---|---|---|---|---|---|---|
| *Amphibalanus improvisus* [a] | 316000 | 2850 | 2250 | 3940 | 625 | 73 | 3.90 | 1.34 | 0.11 | 0.06 |
| *Balanus sp.* [o] | | 1600 – 5000 | | | 80 – 3800 | | | | | |
| *Balanus sp.* [z] | | | | | 39 – 313 | | | <18.2 | 0 – 1.48 | |
| *Balanus balanoides* [d] | | | | | 77 | | | | | |
| *Amphibalanus reticulatus* [h] | | | | 1000 – 2000 | | | | | | |
| *Cerastoderma glaucum* [a] | 348000 | 3220 | 1700 | 80 | 11.9 | 15.0 | 4.08 | 0.140 | 0.011 | 0.096 |
| *Cerastoderma glaucum* [g] | 400000 | | | 280 | 26 | | 0.16 | 17 | 0.015 | |
| *Cerastoderma edule* [r] | | | | | | | | 1.11 – 4.42 | 0.03 – 4.94 | |
| *Limecola balthica* [a] | 345000 | 3100 | 1770 | 80 | 7.9 | 13.0 | 3.0 | 0.14 | 0.007 | 0.034 |
| *Macoma (=Limecola) balthica* [s] | | | | | | | | 2.8 | | |
| *Mya arenaria* [a] | 363000 | 3080 | 2160 | 150 | 1.40 | 2.60 | 0.031 | 0.015 | 0.002 | 0.006 |
| *Mya arenaria* [i] | | | | | | | 9.5 | | 0.05 | |
| *Mytilus trossulus* [a] | 344000 | 2010 | 1170 | 1060 | 54 | 17 | 14 | 1 | 0.091 | 0.053 |
| *Mytilus trossulus* [j] | | | 1185 | | | | | | | |
| *Mytilus trossulus* [g] | 310000 | | | 1400 | 120 | | 0.16 | 24 | 0.029 | |
| *Mytilus trossulus* [g] | 420000 | | | 1700 | 90 | | <0.02 | 5.2 | 0.009 | |
| *Mytilus edulis* [e] | | | 970 | 970 | 24.2 | 3.31 | <dl | 0.321 | | 0.007 |
| *Mytilus edulis* [x] | | | 862 | 856 | 0.13 | 0.26 | <dl | 0.385 | | 0.0009 |
| *Mytilus edulis* [n] | 398000 | | | | | | | | | |
| *Mytilus galloprovincialis* [v] | | | | | | | 9.54 | 1.19 | | |
| *Mytilus galloprovincialis* [i] | | | | | | | 10 | 7.3 | | |
| *Mytilus galloprovincialis* [b] | 360000 | 2500 – 4000 | 500 – 800 | 900 – 1400 | <160 | 0 – 3.8 | 0 – 5.5 | 0 – 5.0 | 0 – 3.0 | |

| *Shell weight ratio element/Ca* | Na/Ca | Sr/Ca | Mg/Ca | Mn/Ca | Ba/Ca | Cu/Ca | Pb/Ca | Cd/Ca | U/Ca |
|---|---|---|---|---|---|---|---|---|---|
| | [arbitrary] | | | [mg/g] | | | | | |
| *Amphibalanus improvisus* [a] | 0.012 | 0.009 | 0.007 | 1.978 | 0.231 | 0.012 | 0.0042 | 0.0003 | 0.0002 |
| *Amphibalanus amphitrite* [t] | | 0.01 | | | | | | | |
| *Balanus balanoides* [u] | 0.009 | 0.009 | 0.01 | 0.0005 | | | | | |
| *Balanus balanoides* [u] | 0.010 | 0.007 | 0.01 | 0.001 | | | | | |
| *Balanus balanoides* [u] | 0.017 | 0.008 | 0.01 | 0.0003 | | | | | |

| *Shell molar ratio element/Ca* | Na/Ca | Sr/Ca | Mg/Ca | Mn/Ca | Ba/Ca | Cu/Ca | Pb/Ca | Cd/Ca | U/Ca |
|---|---|---|---|---|---|---|---|---|---|
| | [mmol/mol] | | | [µmol/mol] | | | | | |
| *Amphibalanus improvisus* [a] | 15.7 | 3.3 | 20.5 | 1438 | 67.3 | 7.71 | 0.82 | 0.070 | 0.057 |
| *Amphibalanus improvisus* [p] | | 4.20 | 20.55 | 0.71 | | | | | |
| *Balanus Balanus* [c] | 20.49 | 3.85 | 14.83 | 0.077 | 0.055 | | | | |
| *Mya arenaria* [a] | 14.8 | 2.72 | 0.68 | 2.80 | 2.09 | 0.053 | 0.008 | 0.002 | 0.003 |
| *Mya arenaria* [m] | | 2.1 – 2.45 | 1.0 – 1.9 | 0.06 – 0.14 | 1.7 – 3.7 | | 0.002 – 0.005 | | |
| *Mytilus trossulus* [a] | 10.16 | 1.55 | 5.07 | 114.16 | 14.40 | 25.44 | 0.56 | 0.094 | 0.026 |

| | | | | | | | | |
|---|---|---|---|---|---|---|---|---|
| *Mytilus edulis* [w] | | | | 60 – 190 | | | | |
| *Mytilus edulis* [l] | 1.1 – 1.6 | 2.5 – 6.0 | | | | | | |
| *Mytilus edulis* [f] | 1.92 | 8.67 | | | | | | |
| *Mytilus edulis* [k] | 1.34 – 1.5 | 5.1 – 7.86 | | 13 – 35 | | | | |
| *Mytilus edulis* [n] | 0.7 – 1.5 | 2.5 – 12.0 | | 100 – 450 | 20 – 70 | | 0.2 –1.3 | | |
| ***Cerastoderma glaucum*** [a] | 16.1 | 2.23 | 0.378 | 24.9 | 12.6 | 7.3 | 0.078 | 0.011 | 0.046 |
| *Cerastoderma edule* [y] | 1.6 | 1.5 | | 20 – 60 | 15 | | 0.01 | | |

<dl – below the detection limit

[a] Baltic Sea, Gulf of Gdańsk (this study), [b] Adriatic Sea, Croatia (Rončević et al., 2010), [c] Arctic (Iglikowska et al., 2018), [d] Atlantic, USA (Blanchard and Chasteen, 1976), [e] North Sea, Germany (Ponnurangam et al., 2018), [f] Baltic Sea, Germany (Heinemann et al., 2008), [g] Baltic Sea, Gulf of Gdańsk (Szefer and Szefer, 1985), [h] Bay of Bengal, India (Raman and Kumar, 2011), [i] Black Sea (Mititelu et al., 2014), [j] Canada (Klein et al., 1996), [k] Cultured (Freitas et al., 2009), [l] Cultured (Heinemann et al., 2011), [m] Cultured (Strasser et al., 2008), [n] Cultured (Vander Putten et al., 2000), [o] Cultured (Gordon et al., 1970), [p] Denmark (Ullmann et al., 2018), [r] England (Price and Pearce, 1997), [s] Canada, estuary (Thomas and Bendell-Young, 1998), [t] Hong Kong (Zhang et al., 2015), [u] Irish Sea, UK (Bourget, 1974), [v] Spain (Puente et al., 1996), [w] North Sea, UK (Freitas et al., 2016), [x] Portugal (Ponnurangam et al., 2018), [y] Portugal (Ricardo et al., 2015), [z] S-W coast of India (Ashraf et al., 2007)

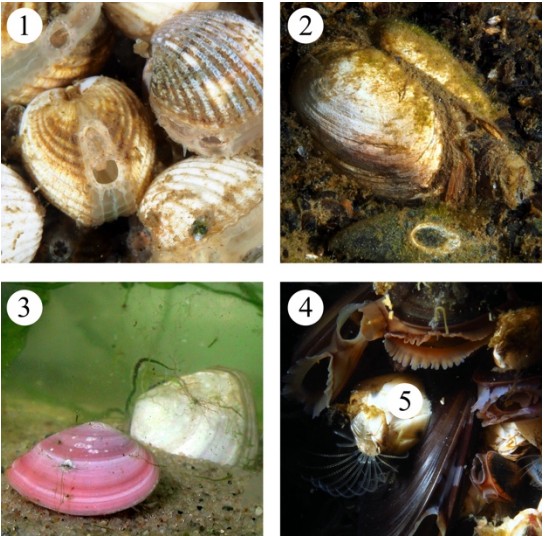

**Figure 1.** The investigated species photographed *in situ*, clams: (1) *Cerastoderma glaucum*, (2) *Mya arenaria*, and (3) *Limecola balthica*, mussels (4) *Mytilus trossulus* and barnacle (5) *Amphibalanus improvises*.

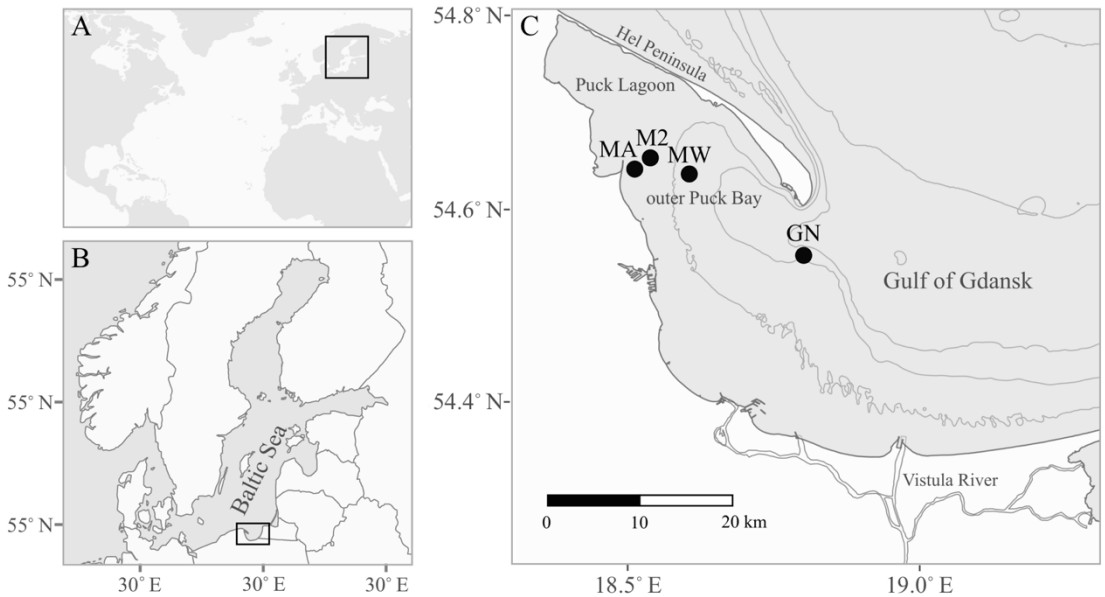

**Figure 2.** The location of the study area: the Baltic Sea (A) and the Gulf of Gdańsk (B) marked by black rectangles. Figure C shows the sampling stations as black circles (see Table 1 for station details). The grey lines indicate 20 m isobaths.

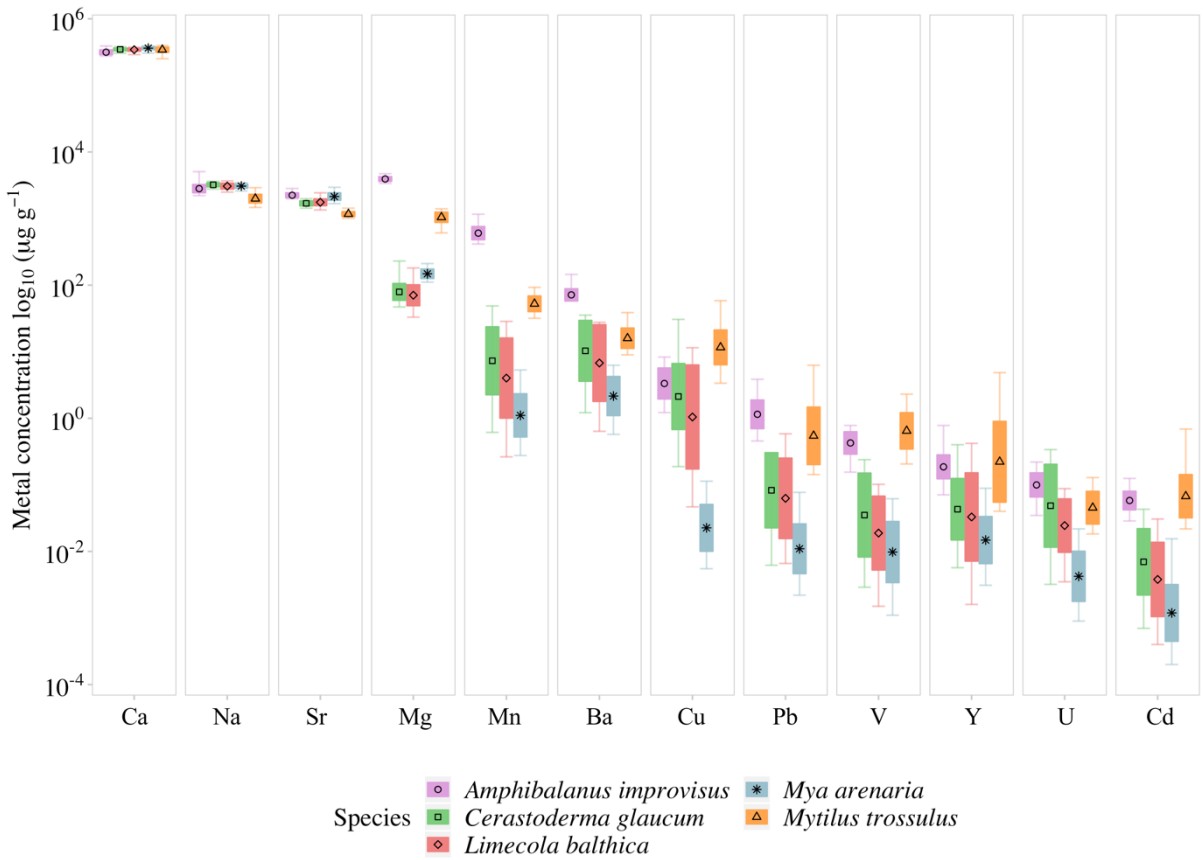

**Figure 3.** The distribution of logarithmically transformed mean elemental concentrations in shells of studied species (grouped by colour). The boxplots represent standard deviations (±1SD) around means, and whiskers indicate the minimum and maximum concentrations for each studied species.

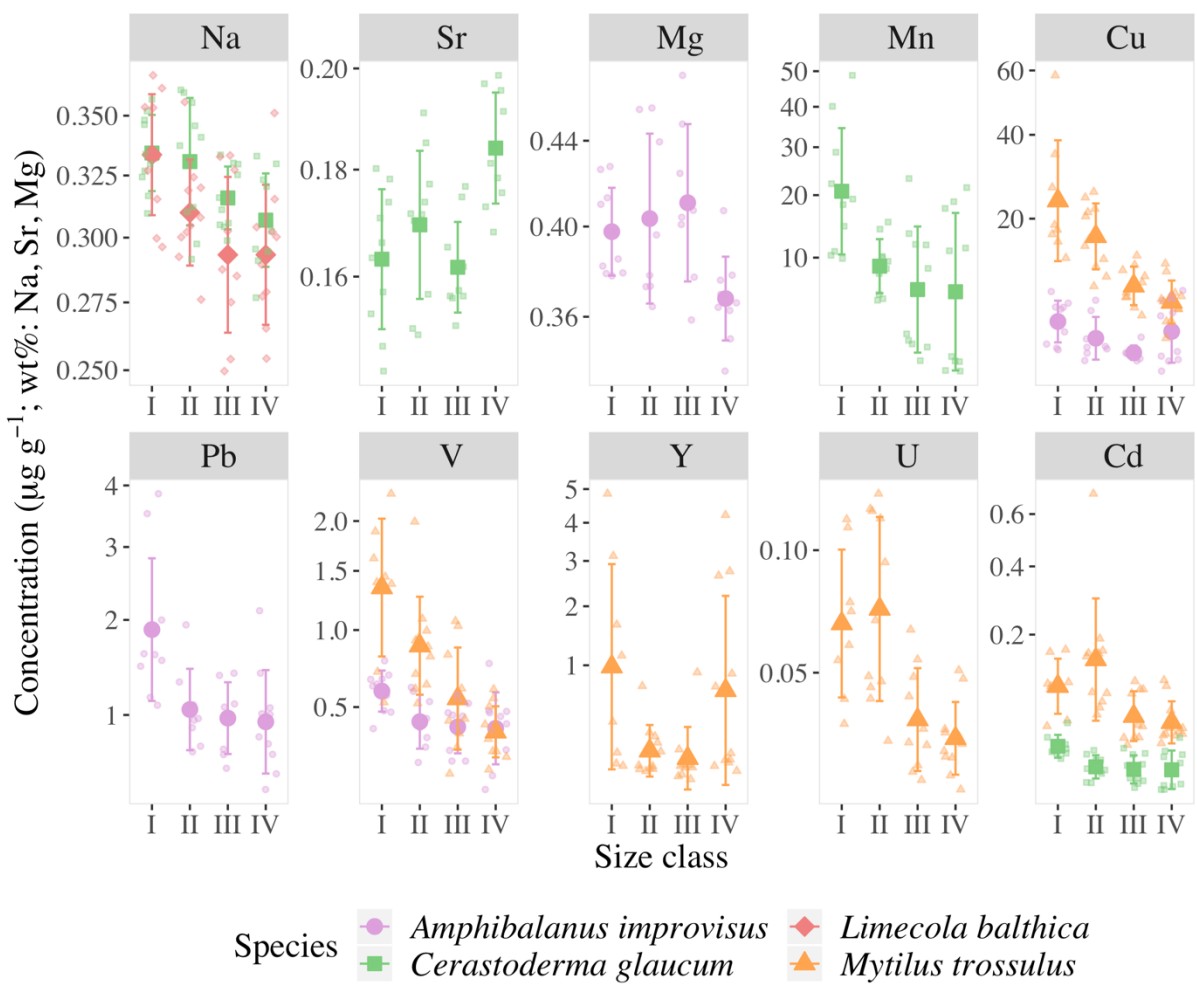

**Figure 4.** The mean concentrations of metals with statistically significant differences between the four size classes (for size class details see Table 2) in the shells of studies species, error bars indicate standard deviations (±1SD). Concentrations of Mn, Cu, Pb, Cd, V and Y are presented in square root scale.

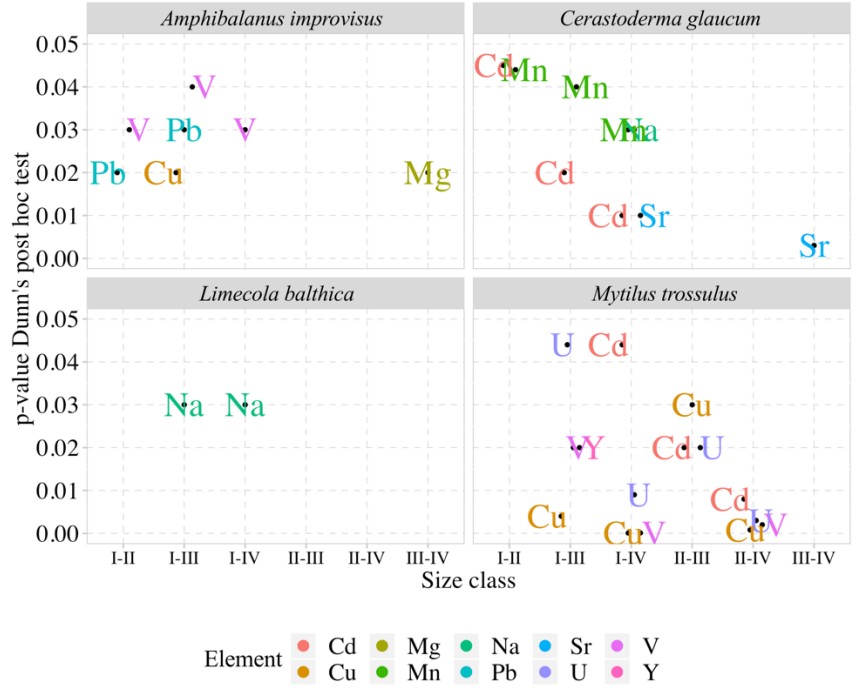

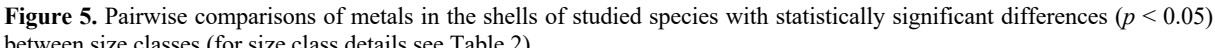

**Figure 5.** Pairwise comparisons of metals in the shells of studied species with statistically significant differences ($p < 0.05$) between size classes (for size class details see Table 2).