# Peer review of "The patterns of elemental concentration (Ca, Na, Sr, Mg, Mn, Ba, Cu, Pb, V, Y, U and Cd) in shells of invertebrates representing different $\text{CaCO}_3$ polymorphs: a case study from the brackish Gulf of Gdańsk (the Baltic Sea)"

_Biogeosciences, 2019_

## Referee Comment (RC1) · Inge van Dijk (Referee) · 14 Oct 2019

In the manuscript "Elemental composition of invertebrates shells composed of different CaCO3 polymorphs at different ontogenetic stages: a case study from the brackish Gulf of Gdansk (the Baltic Sea)" by Piwoni-Piórewicz et al., the authors compare chemical composition of shells of bivalves and arthopods, which create their shell with (a mixture of) different polymorphs of CaCO3. This is a interesting study which investigates the effect of CaCO3 characteristics on element incorporation, and also tries to disentangle the biological signal by studying different shell sizes and by comparison with data derived from inorganic CaCO3 experiments. Although I am specialized in foraminiferal carbonate chemistry (and cannot truly judge the ecological side of the manuscript), I read the manuscript with interest. All in all, I am excited by studies which include the measurement of multiple elements on biogenic carbonates and I think it is an interesting study. However, I have some questions about the dataset and I am missing certain angles in the current discussion.

Major comments:

I was surprised to read about the cleaning methods used in this study. When investigate chemical composition of foraminiferal shells, we use a much more intense cleaning method with oxidation and reduction steps (especially when specimens are collected in the field), to remove any organics as well as diagenetic coating (like Mn‐Fe‐oxide coatings). I wonder if e.g. only mechanical removal of any organisms present on the shell is enough to remove any (organic or chemical) trace completely, and if coatings are present on the shell after the described cleaning protocol. Are there studies comparing different cleaning techniques?

Furthermore, the amount of organic inside the CaCO3 might have a huge effect on the elemental composition of the shell, e.g. on Na, and the contribution of organics might differ between species and CaCO3 polymorphs, as also acknowledge by the authors (in l.54: 'species-specific organic matrix'). With the method described in this study, the organic matrix will also be analysed. I would like to see this issue discussed in the revised version of the manuscript. Has there been any study on the chemical composition of the organics in different species of bivalves, albeit on the compounds of the matrix or by microscale-analysis of the shell with e.g. nanoSIMS on cross-sections? This should be at least mentioned in the discussion as a potential reason for the offset between species, if not discussed in full detail.

The authors are comparing small and big adults and use the obtained data to look at ontogenetic effects. However, the authors also state this is a very variable environment, with big seasonal (and maybe yearly) changes. Overprinted on the size effect, there is a time effect: the bigger specimens have recorded events that the smaller specimens did not experience. I'm missing the longevity of the different species in the discussion of the results. For example: The size effect observed for A. improvisus (life span = 1 year) might simply be a seasonal signal in food supply (and thus maybe growth rate) or physio- or chemical parameters like seawater temperature. If the samples are taken in end of summer (sample date not mentioned in the manuscript, should be added), it would explain why the Mg of the larger specimens is lower: lower temperatures lead to lower incorporation of Mg and these larger specimens likely experienced the winter period, while the smaller specimens maybe spawned in spring. As for the species with longer lifespans (of 10-12 years), could the decrease in element incorporation with size (thus, for older specimens) be due to increased heavy metal output of the Vistula river over the last years? Is there any (historical) data on this?

The authors see some difference between size classes, they conclude that in general smaller specimens have increased trace metal incorporation. Can this also be due to absorption of elements or diagenetic precipitation on the outside of the shell, compared to the more pristine $CaCO_3$ below the surface, leading to a surface/volume effect? E.g. larger specimens have thicker shells, and thus lower surface over area ratio?

I would also like to see habitat depth included in the discussion. The authors are assuming the shell chemistry of the different organisms tested are all reflecting either food or ambient seawater chemistry, but some organisms are living on hard substrates, while other live a few cm in the sediment, and the most extreme one (Mya arenaria) can live 20-30 cm in the sediment. The latter species would have a totally different "ambient seawater conditions", as it is exposed to interstitial water that is very likely to have totally different chemical signature then the overlying water. It would probably be in contact with e.g. much higher Mn concentrations. This is not reflected in the shell

chemistry, so maybe this species does not take up elements from the seawater, but more from food intake?

For your purpose it would be best to have also some kind of idea of the (evolution) of trace metal concentrations in seawater over time. Is there any chemical data on the seawater available from this area? E.g. about the metal concentrations close to the Vistula river (l. 478)? Is there data showing that station GN is indeed increased in heavy metals compared to the other stations? These other stations are located in a bay area, making it possible the residence time of the water is higher, and there might be an actual increase in the metal concentration here.

In retrospect, for the main goal of the study, it maybe would have been better to not analyze full shells, but make small aliquots/subsamples by e.g. drilling the shell. Is there any data (in literature) on small-scale variation in the shells of (some of these) species?

In my opinion, at the moment, you have a combination of too many variables: CaCO3 polymorph, different stations environmental variables (incl. unknown chemical compositions of the seawater), size effect and the vital effect (calcification pathway) of the organisms. It becomes very difficult to disentangle different drivers of shell chemistry, which means you have to be more careful in your conclusions, or at least convince readers which variables are minor/neglectable. I think some variables, like the different sampling stations, can be convinced as being minor, by showing chemical variability or the hydrological situations between the stations.

The authors often point out the strong seasonality in this region, section 2.1 and through the manuscript, e g. l. 481-483. Maybe it is possible to add a (supplementary) figure to section 2.1, if needed compiled from literature data, about the environmental variability in this area, to show differences in physio-chemical parameters. This way, readers, like myself, that are not familiar with the study area can have a good overview of the (yearly) environmental variability in this area.

[Figure]

Minor comments:

(Since I have a lot of major points for the discussion section, I give minimal textual changes, since I believe the manuscript, especially the discussion, will probably greatly change after revision.)

Throughout the manuscript:

-Change 'Mg/Ca ratio' to 'Mg/Ca'.

-Check manuscript for (double) bracketing issues, for instance in l.207: '(Darwin, 1854) (Arthopoda, Maxillopoda)' should be e.g. '(Arthopoda, Maxillopoda; Darwin, 1854). Also lines 217, 228, etc. For l.131 and l. 515: reference should not have brackets.

Abstract:

-The abstract as it reads a bit stiff. Please consider rewriting this section. For example, l. 28-29 on sample location can be merged with the first sentences, while line 29-30 is an explanation of the method, which should be either removed, or shortened, in my opinion.

-l. 26-27: 'The potential impact of environmental factors on the observed elemental concentrations in the studied shells is discussed': Is this really the case? Since there is no data on the environmental parameters presented, it is difficult to discuss the data in this framework.

Introduction:

-l.64-65: 'crystal layers are precipitated successively at regular periodicities,' is not true for all marine calcifyers, like Foraminifera. make it clear when you switch from all marine calcifyers to marine invertebrates.

-l.88-90: maybe add Stanley, 2008 , it is a nice overview paper. Stanley, S. M. (2008). "Effects of global seawater chemistry on biomineralization: past, present, and future." Chemical reviews 108(11): 4483-4498.

-l.144: remove . after shells

Method section:

-l. 263: When were the samples taken?

-l. 295 What was the Ca concentrations in the sample solutions? 100ppm or varying?

-l.297: What were the accuracy and precision of the measured elements?

Discussion section:

-I would advise to divide the discussion session in smaller paragraphs to increase readability.

-I would like to see variables as life span, habitat depth and organic material in the shell (see above) included in the discussion.

-l. 372: You obtained specimens with the same polymorph from two contrasting temperatures: i.e. aragonite Cerastoderma glaucum (16.9°C), Limecola balthica (4.6°C), Mya arenaria (16.9°C), I would like to see a discussion on the (absence of) temperature effect Sr incorporation, which is currently lacking in the manuscript, while it is being discussed for Na and salinity.

-l. 478: ref? Are there any studies on this?

-l. 483: suggest to change 'animal' into 'organism'

-Fig 2 and 4: indicate which polymorph of CaCO3 is used, like Fig. 3.

-Fig. 3: where possible, please use the same scaling for the y-axis for comparability, e.g. y axis of Mg for aragonitic species.

---

## Referee Comment (RC2) · Anonymous Referee #2 · 20 Nov 2019

Review of manuscript bg-2019-367

"Elemental composition of invertebrates shells composed of different CaCO3 polymorphs at different ontogenetic stages: a case study from the brackish Gulf of Gdansk (the Baltic Sea)" by Anna Piwoni-Piorewicz, Stanislav Strekopytov, Emma Humphreys-Williams, and Piotr Kuklinski

This manuscript describes the elemental composition of aragonitic and bimeralic bi-

valves, as well as a calcitic barnacle collected from the Gulf of Gdansk. The role of ontogeny is assessed by comparing elemental composition across four different size classes for each species. The authors conclude that differences in the elemental composition are dependent on the polymorph type i.e. calcitic vs. aragonitic, and that both environmental as well as ontogenetic controls strongly affect the elemental variabilities.

This is an interesting study, and in my opinion, could be a welcomed contribution for the community working on geochemistry of carbonates, biomineralisation and their potential application as recorders of past environmental and climatic reconstructions. I also welcome the authors approach by focusing the study on different calcifying groups and carbonate polymorphs from one region, which could potentially provide further insights on the underlying controls determining the elemental composition of the different biominerals. However, I have some serious issues when it comes to the analytical procedures, the data quality, the study design and interpretation of the data, as well as the manuscript focus and structure that, in my opinion, need to be carefully addressed before it may be considered for publication in Biogeosciences. I find the manuscript generally well written in terms of language, but given the diverse dataset with multiple variables, I strongly recommend to make the text as well as the figures more accessible to the readers and make the manuscript more clearly and systematically structured. I am missing critical information in the Methods part, and the Discussion section lacks several important aspects that need addressing and overall needs to be sharpened. The written and visual presentation of Results could also benefit from improvement, and, personally, I would advise the authors to better extract the principal findings of the study in the Abstract. All data from this manuscript should be provided in a Supplement, or other appropriate and accessible online data repository. In the following, I try to summarise my main concerns (as in my opinion the manuscript will need substantial modifications, I have not provided minor editorial comments at this stage; and neither have checked the referenced).

Fundamental information regarding the measurements and concentration calculations

is missing or unclear, and the analytical uncertainties and repeatability based on appropriate reference materials are also lacking. In addition, in my opinion, the sample preparation and pre-cleaning procedures are questionable. The authors must prove the validity and explain their analytical procedures, and take into account analytical uncertainties when presenting or interpreting their data.

Furthermore, as detailed below, I have problems understanding how whole shell bulk measurements may be used to assess the role of ontogeny or even environmental variations. By using entire shells, the authors 'average' the composition of the growth lines precipitated during earlier and later stages of life, as well as the composition of growth lines built during different seasons or under different environmental conditions. Thus, I am not sure that by comparing bulk values from smaller vs. bigger (younger vs. older) individuals it is possible to determine whether environmental or ontogenetic controls drive the composition of the shell. Simply, the differences between the mean bulk values would depend on the elemental variability encompassed in the shell, which would depend on individuals' growth and environmental conditions experienced. ÂňThe mean bulk values from older individuals integrate large intra-shell variabilities, while in younger individuals smaller intra-shell variabilities, but I think that with this design it is difficult to disentangle the underlying controls on the elemental composition of the carbonates. In my opinion, the authors need take into account these problems, before any interpretations can be made. While a great deal of information is unfortunately lost by using average values, and I think the authors really have to reconsider the interpretations that can be made from this data and discuss their limits, I do acknowledge the authors' efforts for measuring numerous individuals, which I do not think is often done, and perhaps a point to that could be better taken advantage of. Just as a suggestion, maybe, this could be of use for defining the 'typical range' for each element for each species in the Gulf of Gdansk, which could be then compared to literature values from same / similar species in other parts of the world with very different settings. If possible, I think it would be interesting to see how the general elemental concentrations and variability compares between regions or not, and could be of use when constraining environmental influences on the biomineral composition. In addition, I would also like to see a comparison between the different sampling sites within the Gulf of Gdansk. While on one hand it could be perhaps assumed that the differences between the sites are negligible, this is a very dynamic environment, and it might be that spatio-temporal variations account, at least partially, for some of the observed variabilities.

Moreover, the problem of using bulk also limits the interpretation of the data when it comes to the different polymorphs, and particularly this is the case for the bimineralic bivalve. Here, the authors may only conclude whether the composition of the mixture is different to other species building pure calcite vs. aragonite. However, it does not answer the question whether the composition of the calcitic part or aragonitic part within fundamentally differs and how much, which I think is the relevant question here. When discussing the composition of the bimeralic bivalves the authors could, at least, attempt to estimate the contribution from each polymorph to the mixture, and discuss the implications.

One thing that has surprised me the most about this study is that, despite the careful organism sampling strategy, the authors did not consider collecting and measuring water samples. In my opinion, this should come first in this kind of studies, and something I was expecting to see, and thus a real shame it was not done, especially since the authors had the opportunity to do so (and elemental analyses on water samples are relatively more straightforward that on carbonates). Data on seawater chemistry is critical for the calculation of partitioning coefficients, which could ease the interpretation of the results from different sites (in the case that the chemistry at the different sites strongly varies). While it may be a tall task to ask for the measurements at this stage, the authors should, at least, compile the available information on local concentrations of elements in seawater (including additional physico-chemical characteristics), and estimate the partitioning coefficients for each element for the different species.

Specific comments:

Line 1-2: I would suggest to reconsider the title language – 'composition' and 'composed', as well as 'different' twice in the same sentence, this is not orderly.

Line 32: 'Mg > Sr > Na' this needs a written definition first.

Line 195-197: Here, it would be particularly useful to provide concrete numbers on the local carbonate chemistry (other than $\Omega$). Ideally, this should have been measured upon the collection of the specimens from in situ water samples, however, if this is not available the authors could at least summarize the information from the literature. An overview table with the physico-chemical characteristics of the local waters (including temperature and salinity trends etc., carbonate chemistry as well as the elemental composition), would be particularly useful.

Line 267: Why no water samples were collected?

Line 282-286: I have difficulties following this protocol and serious doubts on its effectivity and validity. Previously, the authors state that the periostracum was first physically removed. This is good and indeed important as it constitutes a large amount of organic material, which is difficult to treat chemically without having an impact on the carbonate. However, organic rests might still be present on the inside of the shell for example from the mantle, and foremostly in the pore spaces. Thus, physical cleaning is insufficient, and at least at a powder stage it is a generally established routine to apply a cleaning protocol step, consisting of oxidation of organics by buffered hydrogen peroxide (Barker et al., 2003 G3 4, 8407). As far as I am aware, this protocol or close adaptations are commonly applied to a wide range of calcifiers from forams to corals, bivalves and even brachiopods. In this sentence the authors indeed mention the use of H2O2, but only after the dissolution of the sample, which logic I cannot follow. All in all, I do not think that this is the correct way to treat carbonates samples, and would strongly recommend to first demonstrate the validity of this protocol (if the authors insist on using it, or follow a more broadly used protocol such as that of Barker et al., 2003).

Moreover, I am wondering why the entire shells were crushed? Surely, >100 mg is not

required for the analyses, as concentration measurements are typically done on <mg level. Why did the authors decide to measure the entire shell instead of e.g. a profile across the shell or different growth bands? Such approach, I believe, would be much better for defining an ontogenetic trend, and could also provide some insights into the intra-shell variability. The intra-shell variability, in particular, would be very meaningful to assess before any mean bulk values are used for interpretation of ontogenetic or environmental signals – i.e. how heterogenous are the shells, what is the driver, is it random or not, how big is the variation and what it reflects? I think it is really a shame this was not considered beforehand as a great amount of information from the shells is lost when measuring the whole shells rather than specific parts. Furthermore, I am not convinced that comparison of bulk large vs. small individuals, in this case, answers the question whether ontogenetic trend drives the elemental variability. For numerous calcifiers group, the partioning of elements between seawater and the carbonate is within a certain range band 'baseline' which is principally determined by their calcification mechanisms and mineralogy, and then this variability of the 'baseline' may be driven by environmental factors. In such case, simply by a probability, larger individuals would have lived longer vs. smaller individuals and thus likely witnessed during their life time more environmental fluctuations (e.g. temperature, nutrients, pH, O2, etc.). Thus, when using an average of an entire shell, it is reasonable to assume that the mean of the shell integrates larger intra-shell and therefore elemental variations in the older individuals in contrast to the younger, simply because they experienced more changes over their life. I believe that this is also quite apparent in Fig. 3. How may one, therefore, discriminate between ontogeny vs. environmental variability?

Also, when it comes to ontogenetic trends, let's take for example bivalves and specifically Mytilus, as far as I am aware, broadly speaking their shell growth follows von Bertalanffy growth curve (see e.g. ig. 3; Steffani & Branch, 2003; Mar Ecol Prog Ser 246, 197-209), which is common for many calcifiers. This means that during the very early shell formation the carbonate precipitation is relatively faster, which for the incorporation of numerous elements translates into kinetic effects. It is thus the geochemical

composition of the umbo and the first growth lines vs. the latter growth lines (the ones at the growth 'plateau') that form the greater part of the valve that is commonly attributed to being driven by ontogeny. Potentially, in the case of the very small and thus very young individuals, their geochemical composition may reflect one environmental condition e.g. certain season and one ontogenetic stage i.e. the one dominated by kinetic factors, but I am not sure this can be directly compared to older individuals which mean elemental composition then reflects different ontogenetic stages (with potentially different contribution of each to the bulk), and broad range of seasons. Or am I missing something?

Line 294: What type of solutions? What do you mean by matrix-matched – one solution for each carbonate polymorph? Please provide more details.

Line 300: Why were the standards not treated the same way as samples? First, I do not think it is acceptable that the authors do not process the standards and the samples in the same way, and second, I do not think that the standards are representative and should be compared to these samples. The authors need to provide the measured absolute values (as well the relative standard deviation over the analysis period at least) of comparable biogenic standards such as JCp-1 or JCt-1, or similar internationally accepted alternatives.

Also, regarding the methodology, I am wondering how were the obtained counts converted into concentrations; e.g. did the authors use a calibration line for this or standard-bracketing? Did you normalise all measurements to a stable concentration of a selected element, e.g. Ca? What was the precision of the individual analyses, and the long-term reproducibility? How many times was each sample measured? Line 305 'most trace elements' – which elements were measured in He mode and which not? The authors must provide these details with rigour. Line 307 'periodic analyses' do you mean the standards were not measured along with the samples in a sequence? I have serious doubts on these analytical protocols, and especially do not consider it a good practice to not include standards along with samples in a run.

Line 311: I would really welcome some visual representation for this – i.e. pictures of the different species, maybe with the different ontogenetic stages for each. It is really shame this is not provided; the authors study various interesting species, which offers an opportunity to include visually appealing picture figures, which is not used. Perhaps this is too much to ask, but given that the species build very different carbonate types and I assume microstructures, scanning electron microscope images could also be very relevant and interesting here.

Line 321: Throughout the Results section the figures are referred to very sporadically only, and there are several instances that a value is given and a statement is made, however the figure is not referred to afterwards. Foremostly, all individual panels of the figures need sub-categories (e.g. a, b, c, etc. please check the Biogeosciences format style), and need to be mentioned where the individuals results are being discussed.

Line 322: I am not sure what the authors mean here, please rephrase.

Line 327: I would say it is more appropriate to use $\mu$g/g rather than mg/kg.

Line 328: When concluding that some elements were 'generally present at higher concentration' or lower please also provide the concrete numbers in the text, here, but also in further parts of this section it is missing.

Line 334: What do you mean by 'lack of ontogenetic trend'?

Line 371: The entire Discussion section needs major revision, and foremostly substantial reorganisation in order to make it more suitable to the readers and a wider audience. I am aware that dealing with many different variables like several elements, size classes, species and carbonate polymorphs is not easy, but the authors really need to find a better way for presenting their findings and extracting their 'main message points' to the audience. At the moment I find the Discussion very broad and, to me, it does not provide clear answers to the research questions. I am afraid that often problems are addressed that cannot be resolved by the present dataset. I would say

that it is better if one or two key points are discussed in-depth rather than touching on the surface many (these may still be mentioned, but the in a more concise form, with focus on the key points).

The structuring is also relevant for the other parts of the manuscript and especially the Results section. I would start with ensuring that were possible, the geochemical data is presented in a more systematic manner. The Discussion could benefit from being divided into different subsections, where different aspects are being discussed. The data quality and limitations need discussing, as well each of the different factors controlling the incorporation of the elements into the carbonate (preferably in different subsections), a comparison to other studies, and the implications of the presented findings (for e.g. biomineralisation, application as recorders of environmental conditions). At this stage, it is difficult for me to make a concrete suggestion on how to subdivide this, the authors need to see what works best when structuring the Discussion and the message they would like to convey. I would also suggest to separate the Results section, perhaps by species could work well for this part. Al

Line 372: There are numerous studies on Mg and Sr in carbonate, which uses and incorporation mechanisms, potential proxy-applications etc. need a better summary. Same for all other elements, the discussion of each element should be opened by the factors that control its incorporation into the carbonate. Also, as these are often not similar for calcite and aragonite, and especially since this study is focused on the incorporation of elements into different polymorphs, these two should be treated separately.

Line 375: The statistics should be provided in brackets. Also, please be specific, how much?

Line 378: 'Mg was the dominant impurity', please rephrase, what do you mean?

Line 383: Please be specific, what species?

Line 397: What is the origin of the high Sr in barnacles?

Line 405: The concentrations are sometimes given in mg/kg and sometimes in wt%, which is confusing. Please be consistent throughout the manuscript in figures, and this should be preferably $\mu$g/g.

Line 414: Please explain, what do you mean?

Line 478: I wonder how would the data look if the metal concentrations are plotted as a function of the distance to the Vistula River mouth? Can you conclude that it is the contamination that controls the trace metal composition? A comparison to the species from non-contaminated water might help.

Line 481: Yes, and it is really necessary to add that the whole shells were measured. Therefore the mean values integrate these variations.

Line 486-489: Please rephrase. Also, of course, they varied but it is difficult to determine why.

Line 497: 'chemical profiles' please rephrase, as far as I am aware no chemical profiles were made.

Line 479-509: This sections contains many redundant parts, and the discussion could be sharpened.

Line 510: In addition to relative increase or decrease in concentrations, also the variability in the elemental concentration for a size class should be considered (although I am not sure if the differences between size classes will be significant).

Line 520: Please be specific, which trace elements (please provide in brackets; similar cases can also be found in other parts of manuscript).

Line 527: Yes, but as mentioned I doubt this has anything to do with the size / age.

Figure 1: Please provide the full site names in the figure caption to abbreviations. What are the grey lines in the big panel (bathymetry?), please specify in caption as well.

Figure 2: This figure needs error bars. The analytical uncertainty should be shown here, as well as the variation of the mean i.e. the 2SD of the mean for each group and the respective n should be provided too. Also, what size classes were used for this? Is this the mean of a certain size class or the mean of all individuals, this needs definition in the caption. It may be more appropriate, too, instead of the mean of all individuals to depict the mean and the variation of each size class. I would also include information on the different polymorphs of each species. In general, I have no problems with the figures being black-and white only, but personally, I would try to improve the visual representation. In this case, maybe increasing the figure size to double and placing the legend within the top right corner could help separate a bit more out the different elements. Also, this is a detail, but to make it more intuitive, the grey filled symbols could be the aragonitic species, empty symbols the calcitic and half-filled for example bimineralic.

Figure 3: What is the x-axis? Please make the y-axis similar where possible, this is really difficult to read for me. Also, the information on the differences between size classes should be removed as at the moment there is too much information in this figure. The individual panels are missing sub-headings that should be also referred to in the manuscript text.

Figure 4: Please appropriately label all panels as 'a,b,c, etc.' What do you mean by 'raw data as black dots'? (I see blue dots.) Please include polymorphs, analytical uncertainty, indicate the sizes for each category. Maybe better to put each species in a separate row. Why some size classes have values in between the size class number categories?

Figure 5: I find this figure difficult to follow, maybe there is a better way to illustrate the message? Should be 'dashed line' instead of 'broken line'. Why are some panels darker? Please specify in the caption.

---

## Author Comment (AC1) · 15 Jan 2020

We are grateful for this review, that will help us to improve the manuscript. We carefully read the comments and tried to answer all questions in a clear and concise manner.

Major comments:

Comment: I was surprised to read about the cleaning methods used in this study.

[Figure]

When investigate chemical composition of foraminiferal shells, we use a much more intense cleaning method with oxidation and reduction steps (especially when specimens are collected in the field), to remove any organics as well as diagenetic coating (like MnâARFeâARoxide coatings). I wonder if e.g. only mechanical removal of any organisms present on the shell is enough to remove any (organic or chemical) trace completely, and if coatings are present on the shell after the described cleaning protocol. Are there studies comparing different cleaning techniques?

Response: Concerning any surface contamination, the difference between the foraminifera and mollusk shells is the specific surface area, which is low in case of relatively large shells. The argument against chemical cleaning protocols was based on the observed preferential leaching of Mg (and, therefore, potentially many trace elements) out of carbonate skeletons during chemical cleaning observed by Loxton et al. (2017).

Comment: Furthermore, the amount of organic inside the CaCO3 might have a huge effect on the elemental composition of the shell, e.g. on Na, and the contribution of organics might differ between species and CaCO3 polymorphs, as also acknowledge by the authors (in l.54: 'species-specific organic matrix'). With the method described in this study, the organic matrix will also be analysed. I would like to see this issue discussed in the revised version of the manuscript. Has there been any study on the chemical composition of the organics in different species of bivalves, albeit on the compounds of the matrix or by microscale-analysis of the shell with e.g. nanoSIMS on cross-sections? This should be at least mentioned in the discussion as a potential reason for the offset between species, if not discussed in full detail.

Response: Removal of organics without mobilisation of any trace elements associated with CaCO3 is not a task that is easy to achieve. There are good studies on this subject, e.g., Barker et al (2003), Holcomb et al (2015), see also Loxton et al (2017) for further discussion of this issue. However, we do not believe there is a single accepted protocol for bivalve shells that is tested and validated for a large range of trace ele-

ments. We, therefore, have opted for the analysis of the bulk composition instead of trying to analyse selectively the CaCO3 phases. We will make it more obvious and discuss further in the revised version of the manuscript. In the revised version of the manuscript, we will emphasise that the variation observed could also be due to presence of organic material within the carbonate structure. This contribution should be minor relative to the major influence of the carbonate shell, while bivalve and barnacle shells contain in general up to 5% organic matter (Bourget; 2004; Rueda and Smaal, 2004), yet some patterns were found eg. for Mg and Sr (Walls et al., 1977; Lorens and Bender, 1980; Takesue and van Geen, 2004).

Comment: The authors are comparing small and big adults and use the obtained data to look at ontogenetic effects. However, the authors also state this is a very variable environment, with big seasonal (and maybe yearly) changes. Overprinted on the size effect, there is a time effect: the bigger specimens have recorded events that the smaller specimens did not experience. I'm missing the longevity of the different species in the discussion of the results. For example: The size effect observed for A. improvisus (life span = 1 year) might simply be a seasonal signal in food supply (and thus maybe growth rate) or physio- or chemical parameters like seawater temperature. If the samples are taken in end of summer (sample date not mentioned in the manuscript, should be added), it would explain why the Mg of the larger specimens is lower: lower temperatures lead to lower incorporation of Mg and these larger specimens likely experienced the winter period, while the smaller specimens maybe spawned in spring. As for the species with longer lifespans (of 10-12 years), could the decrease in element incorporation with size (thus, for older specimens) be due to increased heavy metal output of the Vistula river over the last years? Is there any (historical) data on this?

Response: In this study, individuals were collected in a wide range of sizes from each station, representing different ages and various periods of time, living under the influence of seasonal changes. The idea was to find any patterns related with the biological effect of organisms. This part of the discussion should include a more detailed environmental background, which we will introduce (based on literature data) to draw more certain conclusions about the biological effect. We will put more detailed sampling information.

Comment: The authors see some difference between size classes, they conclude that in general smaller specimens have increased trace metal incorporation. Can this also be due to absorption of elements or diagenetic precipitation on the outside of the shell, compared to the more pristine CaCO3 below the surface, leading to a surface/volume effect? E.g. larger specimens have thicker shells, and thus lower surface over area ratio?

Response: This may be one of the reasons for the variability of metal concentration in shells of different sizes and we will include this aspect in the discussion.

Comment: I would also like to see habitat depth included in the discussion. The authors are assuming the shell chemistry of the different organisms tested are all reflecting either food or ambient seawater chemistry, but some organisms are living on hard substrates, while other live a few cm in the sediment, and the most extreme one (Mya arenaria) can live 20-30 cm in the sediment. The latter species would have a totally different "ambient seawater conditions", as it is exposed to interstitial water that is very likely to have totally different chemical signature then the overlying water. It would probably be in contact with e.g. much higher Mn concentrations. This is not reflected in the shell chemistry, so maybe this species does not take up elements from the seawater, but more from food intake? For your purpose it would be best to have also some kind of idea of the (evolution) of trace metal concentrations in seawater over time. Is there any chemical data on the seawater available from this area? E.g. about the metal concentrations close to the Vistula river (l. 478)? Is there data showing that station GN is indeed increased in heavy metals compared to the other stations? These other stations are located in a bay area, making it possible the residence time of the water is higher, and there might be an actual increase in the metal concentration here.

Response: The sediment type, feeding strategy and environmental sources of metals are important factors affecting the concentration of metals in shells and should be discussed. In the revised version of the manuscript, this subject will be improve. We will add information about sediment type in study area in the context of metal bioavailability. In sandy sediments, elemental concentrations are even several orders of magnitude lower than in silty sediments (Kim et al., 2004). We will also put more emphasis on discussing feeding strategy on the concentration of metals in shells. As previously mentioned, we will introduce a more detailed environmental background (based on literature data).

Comment: In retrospect, for the main goal of the study, it maybe would have been better to not analyze full shells, but make small aliquots/subsamples by e.g. drilling the shell. Is there any data (in literature) on small-scale variation in the shells of (some of these) species?

Response: There are such datasets e.g. for Mya arenaria (Strasser et al., 2008) and we will discuss them in the manuscript. Our strategy of investigating the whole shells was a deliberate choice in order to achieve better detection limits that would be possible with the spatially resolved analysis (e.g. LA-ICP-MS) and to concentrate on analyzing many individuals of different species. We presented data on the level of 12 metals in shells of mussels and barnacles from the Baltic populations, which have not been previously reported. Furthermore, we found some patterns of biological and environmental control over for the concentration of metals in shells.

Comment: In my opinion, at the moment, you have a combination of too many variables: CaCO3 polymorph, different stations environmental variables (incl. unknown chemical compositions of the seawater), size effect and the vital effect (calcification pathway) of the organisms. It becomes very difficult to disentangle different drivers of shell chemistry, which means you have to be more careful in your conclusions, or at least convince readers which variables are minor/neglectable. I think some variables, like the different sampling stations, can be convinced as being minor, by showing chemical variability or the hydrological situations between the stations. The authors often point out the strong seasonality in this region, section 2.1 and through the manuscript, e g. l. 481-483. Maybe it is possible to add a (supplementary) figure to section 2.1, if needed compiled from literature data, about the environmental variability in this area, to show differences in physio-chemical parameters. This way, readers, like myself, that are not familiar with the study area can have a good overview of the (yearly) environmental variability in this area.

Response: The manuscript will present the trace element concentrations in calcitic, aragonitic and bimineralic shells and the patterns governing bioaccumulation of metals in shells. To make the manuscript more accessible for readers, the discussion will be divided into the three parts focused, respectively, on the polymorphic form of calcium carbonate, on potential environmental factors (based on literature data), and on potential biological control based on metal variability in shell size classes. As conclusions, we will distinguish patterns of inter-species and inter-individual variations in the concentration of metals in studied shells, which are associated with biological and environmental control. There is literature data regarding the concentration of some studied metals (mainly in sediments) around the study area (such as Rainbow at al. 2000; Rainbow at al., 2004; Szefer at al. 2002) and we will include this into the manuscript.

Minor comments:

(Since I have a lot of major points for the discussion section, I give minimal textual changes, since I believe the manuscript, especially the discussion, will probably greatly change after revision.) Throughout the manuscript:

Comment: Change 'Mg/Ca ratio' to 'Mg/Ca'.

Response: This will be improved in the revised version of MS.

Comment: Check manuscript for (double) bracketing issues, for instance in l.207: '(Darwin, 1854) (Arthopoda, Maxillopoda)' should be e.g. '(Arthopoda, Maxillopoda; Darwin,

1854). Also lines 217, 228, etc. For l.131 and l. 515: reference should not have brackets.

Response: This will be improved in the revised version of MS.

Abstract:

Comment: The abstract as it reads a bit stiff. Please consider rewriting this section. For example, l. 28-29 on sample location can be merged with the first sentences, while line 29-30 is an explanation of the method, which should be either removed, or shortened, in my opinion.

Response: This section will be rewritten to improve the structure and to better introduce the reader to the content of the manuscript.

Comment: l. 26-27: 'The potential impact of environmental factors on the observed elemental concentrations in the studied shells is discussed': Is this really the case? Since there is no data on the environmental parameters presented, it is difficult to discuss the data in this framework.

Response: While we did not measure the concentrations of elements in the environment, the discussion about their impact on the composition of the shell is challenging, yet very valuable. While we do not have our own data on seawater concentrations, in the revised manuscript we will place more emphasis on the environmental characteristics based on literature data.

Introduction:

Comment: l.64-65: 'crystal layers are precipitated successively at regular periodicities,' is not true for all marine calcifyers, like Foraminifera. make it clear when you switch from all marine calcifyers to marine invertebrates.

Response: Thanks for spotting this, this will be corrected.

Comment: l.88-90: maybe add Stanley, 2008 , it is a nice overview paper. Stanley, S.

M. (2008). "Effects of global seawater chemistry on biomineralization: past, present, and future." Chemical reviews 108(11): 4483-4498.

Response: We will include this article. Thank you for suggestion.

Comment: l.144: remove . after shells

Response: This will be improved in the revised version of MS.

Method section:

Comment: l. 263: When were the samples taken?

Response: The samples were collected in May 2013 and June 2014. These information have been presented in Table 1, but we will also introduce them to the Chapter 2.3. Sample collection and preparation.

Comment: l. 295 What was the Ca concentrations in the sample solutions? 100ppm or varying?

Response: Calibration of the ICP-OES analysis was performed using solutions that were matrix-matched to the high calcium concentrations in the samples at a ratio of 49:1 calcium to magnesium. We will update this in the manuscript to make it clear.

Comment: l.297: What were the accuracy and precision of the measured elements?

Response: We will add more details to the method section: The accuracy and reproducibility of the analyses were checked using two calcium carbonate-rich certified reference materials (CRMs): JLs-1 Limestone and JDo-1 Dolomite (both from the Geological Survey of Japan) prepared by total digestion method (using hydrofluoric acid). The reference materials were diluted to match the concentrations of Ca in sample solutions. Ca, Mg and Sr concentrations were found to be within the uncertainty (1 standard deviation) of the reported values (Imai et al. 1996). Limits of quantification (LOQ) in solution for ICP-MS were generally determined as a concentration corresponding to ten times standard deviation of the signal obtained by analysing 5% HNO3 solution

(6–7 times) in each individual run. ICP-MS was run in helium (He) mode (5 ml min−1 He, 99.9995% purity) for lighter trace elements (V, Mn, Cu, Y and Cd) to minimize the molecular interferences from plasma and solution components and Ca from samples. The accuracy and reproducibility was checked by analyses of JLs-1 and JDo-1 before and after every batch of samples. The results obtained for all elements were within the uncertainty (2.5 SD) of the recommended values (Imai et al. 1996). Accuracy of Pb determination cannot be checked using these CRMs because of the large spread of reference values probably due to insufficient homogeneity of Pb distribution in these samples. Based on the analyses of CRMs and matrix-matched solutions, the maximum analytical error for the typical range of concentrations in the shells can be estimated (in relative percentage) as 1.5% for Ca, Mg and Sr; 3% for Ba; 20% for Cu and U; and 4–10% for all other elements. This is generally similar to what was reported in our previous publication (Piwoni-Piórewicz et al 2017).

Discussion section:

Comment: I would advise to divide the discussion session in smaller paragraphs to increase readability.

Response: The discussion will be divided into the three parts to make reading easier. Due to the large number of factors potentially controlling the metal concentrations in skeletons, the discussion will be first focused on the polymorphic form of calcium carbonate (with the context of the shell organic matter); then on potential environmental factors (based on literature data); and finally on a potential biological response based on tracking metal variability in shell size classes. The discussion will be focused on finding patterns of inter-species and inter-individual variations in the concentration of metals in studied shells.

Comment: I would like to see variables as life span, habitat depth and organic material in the shell (see above) included in the discussion.

Response: We will take these factors into account in the discussion section.

Comment: l. 372: You obtained specimens with the same polymorph from two contrasting temperatures: i.e. aragonite Cerastoderma glaucum (16.9 C), Limecola balthica (4.6 C), Mya arenaria (16.9 C), I would like to see a discussion on the (absence of) temperture effect Sr incorporation, which is currently lacking in the manuscript, while it is being discussed for Na and salinity.

Response: Two species: Cerastoderma glaucum and May arenaria were collected at 10 m depth from the environment affected by cyclic temperature variation. However, the rest of the species were gathered from 31 – 36 m depth where the yearly variation of water temperature was lower. This is an important factor that can affect the concentration of metals in shells and we agree that it should be discussed. In the revised version of the manuscript, this part will be added.

Comment: l. 478: ref? Are there any studies on this?

Response: As mentioned above, the literature data regarding the concentration of some studied metals around the study area (such as Rainbow at al. 2000; Rainbow at al., 2004) and sediment type will be included into the manuscript. This will allow us to compare the bioavailability of metals between stations in the context of their concentrations in shells.

Comment: l. 483: suggest to change 'animal' into 'organism'.

Response: This will be changed as suggested.

Comment: Fig 2 and 4: indicate which polymorph of CaCO3 is used, like Fig. 3.

Response: This will be indicated.

Comment: Fig. 3: where possible, please use the same scaling for the y-axis for comparability, e.g. y axis of Mg for aragonitic species.

Response: This will be improved.

References: Barker, S., Greaves, M., & Elderfield, H. (2003) A study of cleaning procedures used for foraminiferal Mg/Ca paleothermometry. Geochemistry, Geophysics, Geosystems, 4(9).

Bourget, E. In Barnacle Biology, Southward, A.J., Ed.; A.A., Balkema: Rotterdam, 1987, Rueda, J.L. & Smaal, A.C. (2004) Variation of the physiological energetics of the bivalve Spisula subtruncata (da Costa, 1778) within an annual cycle. Journal of Experimental Marine Biology and Ecology, 301: 141-157.

Holcomb, M., DeCarlo, T. M., Schoepf, V., Dissard, D., Tanaka, K., & McCulloch, M. (2015) Cleaning and pre-treatment procedures for biogenic and synthetic calcium carbonate powders for determination of elemental and boron isotopic compositions. Chemical Geology, 398, 11-21.

Imai, N., Terashima, S., Itoh, S. & Ando, A. (1996) Compilation of analytical data on nine GSJ geochemical reference samples, Sedimentary rock series, Geostand. Newsl., 20(2), 165–216.

Kim, G., Alleman, L.Y., & Church, T.M. (2004) Accumulation records of radionuclides and trace metals in two contrasting Delaware salt marshes. Marine Chemistry, 87: 87-96.

Lorens, R.B., & Bender, M.L. (1980) The impact of solution chemistry on Mytilus edulis calcite and aragonite. Geochimica et Cosmochimica Acta, 44: 1265-1278.

Loxton, J., Najorka, J., Humphreys-Williams, E., Kuklinski, P., Smith, A. M., Porter, J. S., & Spencer Jones, M. (2017) The forgotten variable: Impact of cleaning on the skeletal composition of a marine invertebrate, Chemical Geology, 474, 45-57.

Piwoni-Piórewicz, A., Kukliński, P., Strekopytov, S., Humphreys-Williams, E., Najorka, J., & Iglikowska, A. (2017) Size effect on the mineralogy and chemistry of Mytilus trossulus shells from the southern Baltic Sea: implications for environmental monitoring. Environmental Monitoring and Assessment, 189(4), 197.

Rainbow, P.S., Fialkowski, W., Sokolowski, A., Smith, B.D., Wolowicz, M. (2004) Geographical and seasonal variation of trace metal bioavailabilities in the Gulf of Gdansk, Baltic Sea using mussels (Mytilus trossulus) and barnacles (Balanus improvisus) as biomonitors. Marine Biology, 144, 271–286.

Rainbow, P.S., Wolowicz, M., Fialkowski, W., Smith, B.D., Sokolowski, A. (2000) Biomonitoring of trace metals in the Gulf of Gdansk , using mussels (Mytilus trossulus) and barnacles (Balanus improvisus). Water Research, 34, 1823–1829.

Strasser, M., Walensky, M., Reise, K. (1999) Juvenile-adult distribution of the bivalve Mya arenaria on intertidal flats in the Wadden Sea: why are there so few year classes? Helgol. Mar. Res. 33, 45.

Szefer, P., Frelek, K., Szefer, K., Lee, C.B., Kim, B.S., Warzocha, J., Zdrojewska, I., Ciesielski, T. (2002) Distribution and relationships of trace metals in soft tissue, byssus and shells of Mytilus edulis trossulus from the southern Baltic. Environmental Pollution, 120, 423–444.

Takesue, R.K., & van Geen, A. (2004) Mg/Ca, Sr/Ca, and stable isotopes in modern and Holocene Protothaca staminea shells from a northern California coastal upwelling region. Geochimica et Cosmochimica Acta, 68: 3845-3861.

Walls, R.A., Ragland, P.C., & Crisp, E.L. (1977) Experimental and natural early diagenetic mobility of Sr and Mg in biogenic carbonates. Geochimica et Cosmochimica Acta, 41: 1731-1737.

---

## Author Comment (AC2) · 15 Jan 2020

We are grateful for the review, that will help to improve our manuscript. We carefully read the comments and tried to answer all questions in a clear and concise manner.

Comment: Fundamental information regarding the measurements and concentration calculations is missing or unclear, and the analytical uncertainties and repeatability based on appropriate reference materials are also lacking. In addition, in my opinion,

the sample preparation and pre-cleaning procedures are questionable. The authors must prove the validity and explain their analytical procedures, and take into account analytical uncertainties when presenting or interpreting their data.

Response: The analytical details will be discussed in full in the revised version of MS; also see the specific comments below.

Comment: Furthermore, as detailed below, I have problems understanding how whole shell bulk measurements may be used to assess the role of ontogeny or even environmental variations. By using entire shells, the authors 'average' the composition of the growth lines precipitated during earlier and later stages of life, as well as the composition of growth lines built during different seasons or under different environmental conditions. Thus, I am not sure that by comparing bulk values from smaller vs. bigger (younger vs. older) individuals it is possible to determine whether environmental or ontogenetic controls drive the composition of the shell. Simply, the differences between the mean bulk values would depend on the elemental variability encompassed in the shell, which would depend on individuals' growth and environmental conditions experienced. The mean bulk values from older individuals integrate large intra-shell variabilities, while in younger individuals smaller intra-shell variabilities, but I think that with this design it is difficult to disentangle the underlying controls on the elemental composition of the carbonates. Moreover, the problem of using bulk also limits the interpretation of the data when it comes to the different polymorphs, and particularly this is the case for the bimineralic bivalve. Here, the authors may only conclude whether the composition of the mixture is different to other species building pure calcite vs. aragonite. However, it does not answer the question whether the composition of the calcitic part or aragonitic part within fundamentally differs and how much, which I think is the relevant question here. When discussing the composition of the bimeralic bivalves the authors could, at least, attempt to estimate the contribution from each polymorph to the mixture, and discuss the implications. Moreover, I am wondering why the entire shells were crushed? Surely, >100 mg is not required for the analyses, as concentration mea-

[Figure]

surements are typically done on <mg level. Why did the authors decide to measure the entire shell instead of e.g. a profile across the shell or different growth bands? Such approach, I believe, would be much better for defining an ontogenetic trend, and could also provide some insights into the intra-shell variability. The intra-shell variability, in particular, would be very meaningful to assess before any mean bulk values are used for interpretation of ontogenetic or environmental signals – i.e. how heterogenous are the shells, what is the driver, is it random or not, how big is the variation and what it reflects? I think it is really a shame this was not considered beforehand as a great amount of information from the shells is lost when measuring the whole shells rather than specific parts=-. Furthermore, I am not convinced that comparison of bulk large vs. small individuals, in this case, answers the question whether ontogenetic trend drives the elemental variability. For numerous calcifiers group, the partioning of elements between seawater and the carbonate is within a certain range band 'baseline' which is principally determined by their calcification mechanisms and mineralogy, and then this variability of the 'baseline' may be driven by environmental factors. In such case, simply by a probability, larger individuals would have lived longer vs. smaller individuals and thus likely witnessed during their life time more environmental fluctuations (e.g. temperature, nutrients, pH, O2, etc.). Thus, when using an average of an entire shell, it is reasonable to assume that the mean of the shell integrates larger intra-shell and therefore elemental variations in the older individuals in contrast to the younger, simply because they experienced more changes over their life. I believe that this is also quite apparent in Fig. 3. How may one, therefore, discriminate between ontogeny vs. environmental variability?

Response: In this study, individuals were collected in a wide range of sizes from each station, representing different ages and various periods of time, living under the influence of seasonal changes. The idea was to find any patterns related with the biological effect of organisms. This part of the discussion should include more detailed environmental dataset, which we will introduce (based on literature data; more details below) to draw more certain conclusions about the biological effect. Our strategy of investigating the whole shells was based on the resignation of small-scale analyzes at the expense of analyzing many individuals of different species. We presented data on the level of 12 metals in shells of mussels and barnacles from the Baltic populations, which has not been available until now. Thanks to this, we found patterns of biological and environmental control over for the concentration of metals in shells. Removal of organics without mobilisation of any trace elements associated with CaCO3 is not a task that is easy to achieve. There are good studies on this subject, e.g., Barker et al (2003), Holcomb et al (2015), see also Loxton et al (2017) for further discussion of this issue. However, we do not believe there is a single accepted protocol for bivalve shells that is tested and validated for a large range of trace elements. We, therefore, have opted for the analysis of the bulk composition instead of trying to analyse selectively the CaCO3 phases. We will make it more obvious and discuss further in the revised version of the manuscript. In the revised version of the manuscript we will emphasise that the variation observed could also be due to the presence of organic material within the carbonate structure. This contribution should be minor relative to the major influence of the carbonate shell, while bivalve and barnacle shells contain in general up to 5% organic matter (Bourget; 1987; Rueda and Smaal, 2004), yet some patterns were found eg. for Mg and Sr (Walls et al., 1977; Lorens and Bender, 1980; Takesue and van Geen, 2004).

Comment: In my opinion, the authors need take into account these problems, before any interpretations can be made. While a great deal of information is unfortunately lost by using average values, and I think the authors really have to reconsider the interpretations that can be made from this data and discuss their limits, I do acknowledge the authors' efforts for measuring numerous individuals, which I do not think is often done, and perhaps a point to that could be better taken advantage of. Just as a suggestion, maybe, this could be of use for defining the 'typical range' for each element for each species in the Gulf of Gdansk, which could be then compared to literature values from same / similar species in other parts of the world with very different settings. If possible, I think it would be interesting to see how the general elemental concentrations and

variability compares between regions or not, and could be of use when constraining environmental influences on the biomineral composition. In addition, I would also like to see a comparison between the different sampling sites within the Gulf of Gdansk. While on one hand it could be perhaps assumed that the differences between the sites are negligible, this is a very dynamic environment, and it might be that spatio-temporal variations account, at least partially, for some of the observed variabilities.

Response: These interpretations will be made clearer in the revised version of MS; many of the points we explain in the specific comments below. The suggestion about a 'typical range' for each element for each species in the Gulf of Gdansk, which could be then compared to literature values from same/similar species in other parts of the world with very different settings is very valuable and will be also discussed. We will establish a table will average values (and ranges) for each element in each species as a baseline value for the species in this region, that other researchers may use in the further studies.

Comment: One thing that has surprised me the most about this study is that, despite the careful organism sampling strategy, the authors did not consider collecting and measuring water samples. In my opinion, this should come first in this kind of studies, and something I was expecting to see, and thus a real shame it was not done, especially since the authors had the opportunity to do so (and elemental analyses on water samples are relatively more straightforward that on carbonates). Data on seawater chemistry is critical for the calculation of partitioning coefficients, which could ease the interpretation of the results from different sites (in the case that the chemistry at the different sites strongly varies). While it may be a tall task to ask for the measurements at this stage, the authors should, at least, compile the available information on local concentrations of elements in seawater (including additional physico-chemical characteristics), and estimate the partitioning coefficients for each element for the different species.

Response: Because we did not measure the concentration of elements in the environment, the discussion about its impact on the composition of the shell is challenging, yet very valuable. We should definitely place more emphasis on the environmental characteristics based on literature data. The sediment type, feeding strategy and environmental sources of metals are important factors affecting the concentration of metals in shells and should be discussed. In the revised version of the manuscript, this subject will be improved. We will add information about sediment type in study area in the context of metal bioavailability. In sandy sediments elemental concentrations are even several orders of magnitude lower than in silty sediments (Kim et al., 2004). There is literature data regarding the concentration of some studied metals (mainly in sediments) around the study area (such as Rainbow at al. 2000; Rainbow at al., 2004; Szefer at al. 2002) and we will include this into the manuscript.

Specific comments:

Comment: Line 1-2: I would suggest to reconsider the title language – 'composition' and 'com- posed', as well as 'different' twice in the same sentence, this is not orderly.

Response: Based on the reviews received and the changes that will take place in the manuscript, the title will be changed to: Patterns of metals concentration in invertebrates shells built of different CaCO3 polymorphs: a case study from the brackish Gulf of Gdansk (the Baltic Sea)

Comment: Line 32: 'Mg > Sr > Na' this needs a written definition first.

Response: This will be improved.

Comment: Line 195-197: Here, it would be particularly useful to provide concrete numbers on the local carbonate chemistry (other than $\Omega$). Ideally, this should have been measured upon the collection of the specimens from in situ water samples, however, if this is not available the authors could at least summarize the information from the literature. An overview table with the physico-chemical characteristics of the local waters (including temperature and salinity trends etc., carbonate chemistry as well as the

elemental composition), would be particularly useful.

Response: As mentioned above, available literature data about environment will be included in the revised MS.

Comment: Line 267: Why no water samples were collected?

Response: We agree that environmental research would be very useful in the discussion, yet unfortunately we do not have them. Therefore, we will use data available in the literature for the discussion of the results.

Comment: Line 282-286: I have difficulties following this protocol and serious doubts on its effectivity and validity. Previously, the authors state that the periostracum was first physically removed. This is good and indeed important as it constitutes a large amount of organic material, which is difficult to treat chemically without having an impact on the carbonate. However, organic rests might still be present on the inside of the shell for example from the mantle, and foremostly in the pore spaces. Thus, physical cleaning is insufficient, and at least at a powder stage it is a generally established routine to apply a cleaning protocol step, consisting of oxidation of organics by buffered hydrogen peroxide (Barker et al., 2003 G3 4, 8407). As far as I am aware, this protocol or close adaptations are commonly applied to a wide range of calcifiers from forams to corals, bivalves and even brachiopods. In this sentence the authors indeed mention the use of H2O2, but only after the dissolution of the sample, which logic I cannot follow. All in all, I do not think that this is the correct way to treat carbonates samples, and would strongly recommend to first demonstrate the validity of this protocol (if the authors insist on using it, or follow a more broadly used protocol such as that of Barker et al., 2003).

Response: We are well aware of the presence of the organic matter and are going to discuss it in more details in the revised MS. However, we do believe, as Inoue et al (2004) did that "plausible pre-treatment method [for the removal of organics] is yet to be established". With full appreciation of the importance of the protocols discussed and tested by Barker et al (2004) we believe that the benefits of any chemical treatment still

remain controversial, see, for example, discussion in Holcomb et al (2015) and Loxton et al (2017). We will aim to discuss the potential contribution of organic matter in detail in the revised MS.

Comment: Also, when it comes to ontogenetic trends, let's take for example bivalves and specifically Mytilus, as far as I am aware, broadly speaking their shell growth follows von Bertalanffy growth curve (see e.g. ig. 3; Steffani & Branch, 2003; Mar Ecol Prog Ser 246, 197-209), which is common for many calcifiers. This means that during the very early shell formation the carbonate precipitation is relatively faster, which for the incorporation of numerous elements translates into kinetic effects. It is thus the geochemical composition of the umbo and the first growth lines vs. the latter growth lines (the ones at the growth 'plateau') that form the greater part of the valve that is commonly at- tributed to being driven by ontogeny. Potentially, in the case of the very small and thus very young individuals, their geochemical composition may reflect one environmental condition e.g. certain season and one ontogenetic stage i.e. the one dominated by kinetic factors, but I am not sure this can be directly compared to older in- dividuals which mean elemental composition then reflects different ontogenetic stages (with potentially different contribution of each to the bulk), and broad range of seasons. Or am I missing something?

Response: As mentioned earlier, this part of the discussion will be based on a more accurate environmental background, which will strengthen the inference of potential biological control on metal concentration in shells. Despite the influence of many factors, some patterns of metals concentration were found and presented.

Comment: Line 294: What type of solutions? What do you mean by matrix-matched – one solution for each carbonate polymorph? Please provide more details.

Response: Calibration of the ICP-OES analysis was performed using solutions that were matrix-matched to the high calcium concentrations in the samples at a ratio of 49:1 calcium to magnesium.

Comment: Line 300: Why were the standards not treated the same way as samples? First, I do not think it is acceptable that the authors do not process the standards and the samples in the same way, and second, I do not think that the standards are representative and should be compared to these samples. The authors need to provide the measured absolute values (as well the relative standard deviation over the analysis period at least) of comparable biogenic standards such as JCp-1 or JCt-1, or similar internationally accepted alternatives.

Response: The authors appreciate that using biogenic standards would potentially be preferable; however, those standards were not available to the authors at the time of analysis. On the other hand, complete digestion eliminates potential uncertainty that might originate from potentially incomplete conversion of organic matter typically performed when digesting by HNO3 only (Inoue et al 2004). The use of reference limestone and dolomite for the control of the analysis (without considering the digestion step) is fully justified for the digestion protocol used (HF+HNO3 with evaporation and matrix modification to HNO3 solution of the same concentration). The digestion step includes HNO3+H2O2 mixture, which is perceived to be suitable for digesting CaCO3-based materials with low amount of non-refractory organic material. For comparison, a well-cited paper on the composition of JCp-1 or JCt-1 standards (Inoue et al 2004) employed a milder treatment of HNO3 only at room temperature. We agree that a full method validation employing homogenized samples of clams containing high level of organic matter (in addition to biogenic reference materials, which potentially do not cover the natural range in terms of organic matter content/reactivity) would be desirable, but this must be a subject of a separate study.

Comment: Also, regarding the methodology, I am wondering how were the obtained counts converted into concentrations; e.g. did the authors use a calibration line for this or standard-bracketing?

Response: Calibration was performed typically using 5 points covering the range of concentrations.

Comment: Did you normalise all measurements to a stable concentration of a selected element, e.g. Ca?

Response: We did not normalize measurements to Ca.

Comment: What was the precision of the individual analyses, and the long-term reproducibility? How many times was each sample measured? Line 305 'most trace elements' – which elements were measured in He mode and which not? The authors must provide these details with rigour. Line 307 'periodic analyses' do you mean the standards were not measured along with the samples in a sequence? I have serious doubts on these analytical protocols, and especially do not consider it a good practice to not include standards along with samples in a run.

Response: ICP-OES: The accuracy and reproducibility of the analyses were checked using two calcium carbonate-rich certified reference materials (CRMs): JLs-1 Limestone and JDo-1 Dolomite (both from the Geological Survey of Japan) prepared by total digestion method (using hydrofluoric acid). The reference materials were diluted to match the concentrations of Ca in sample solutions. Ca, Mg and Sr concentrations were found to be within the uncertainty (1 standard deviation) of the reported values (Imai et al. 1996). Limits of quantification (LOQ) in solution for ICP-MS were generally determined as a concentration corresponding to ten times standard deviation of the signal obtained by analysing 5% $HNO_3$ solution (6–7 times) in each individual run. ICP-MS was run in helium (He) mode (5 ml min$-1$ He, 99.9995% purity) for lighter trace elements (V, Mn, Cu, Y and Cd) to minimize the molecular interferences from plasma and solution components and Ca from samples. The accuracy and reproducibility was checked by analyses of JLs-1 and JDo-1 before and after every batch of samples. The results obtained for all elements were within the uncertainty (2.5 SD) of the recommended values. Accuracy of Pb determination cannot be checked using these CRMs because of the large spread of reference values probably due to insufficient homogeneity of Pb distribution in these samples. Based on the analyses of CRMs and matrix-matched solutions, the maximum analytical error for the typical

range of concentrations in the shells can be estimated (in relative percentage) as 1.5% for Ca, Mg and Sr; 3% for Ba; 20% for Cu and U; and 4–10%for all other elements.

Comment: Line 311: I would really welcome some visual representation for this – i.e. pictures of the different species, maybe with the different ontogenetic stages for each. It is really shame this is not provided; the authors study various interesting species, which offers an opportunity to include visually appealing picture figures, which is not used. Perhaps this is too much to ask, but given that the species build very different carbonate types and I assume microstructures, scanning electron microscope images could also be very relevant and interesting here.

Response: Unfortunately, we did not have the possibility to make scanning electron microscope images, but we will include macroscopic images presenting the studied species.

Comment: Line 321: Throughout the Results section the figures are referred to very sporadically only, and there are several instances that a value is given and a statement is made, however the figure is not referred to afterwards. Foremostly, all individual panels of the figures need sub-categories (e.g. a, b, c, etc. please check the Biogeosciences format style), and need to be mentioned where the individuals results are being discussed.

Response: This will be improved in the revised version of MS.

Comment: Line 322: I am not sure what the authors mean here, please rephrase.

Response: The species exhibited the highest concentration of Na, Sr and Mg and the lowest concentrations of U and Cd in shells. The levels of incorporated metals are similar between species, contrary to their bulk concentrations (Table 3, Fig. 2).

Comment: Line 327: I would say it is more appropriate to use $\mu$g/g rather than mg/kg.

Response: This will be improved in the revised version of MS.

Comment: Line 328: When concluding that some elements were 'generally present at higher concentration' or lower please also provide the concrete numbers in the text, here, but also in further parts of this section it is missing.

Response: This will be improved in the revised version of MS.

Comment: Line 334: What do you mean by 'lack of ontogenetic tren'?

Response: The lack of patterns related with the size classes

Comment: Line 371: The entire Discussion section needs major revision, and foremostly substantial reorganisation in order to make it more suitable to the readers and a wider audience. I am aware that dealing with many different variables like several elements, size classes, species and carbonate polymorphs is not easy, but the authors really need to find a better way for presenting their findings and extracting their 'main mes- sage points' to the audience. At the moment I find the Discussion very broad and, to me, it does not provide clear answers to the research questions. I am afraid that often problems are addressed that cannot be resolved by the present dataset. I would say that it is better if one or two key points are discussed in-depth rather than touching on the surface many (these may still be mentioned, but the in a more concise form, with focus on the key points). The structuring is also relevant for the other parts of the manuscript and especially the Results section. I would start with ensuring that were possible, the geochemical data is presented in a more systematic manner. The Discussion could benefit from being divided into different subsections, where different aspects are being discussed. The data quality and limitations need discussing, as well each of the different factors controlling the incorporation of the elements into the carbonate (preferably in different subsections), a comparison to other studies, and the implications of the presented findings (for e.g. biomineralisation, application as recorders of environmental conditions). At this stage, it is difficult for me to make a concrete suggestion on how to subdivide this, the authors need to see what works best when structuring the Discussion and the message they would like to convey. I would

also suggest to separate the Results section, perhaps by species could work well for this part.

Response: The manuscript will present the trace element concentrations in calcitic, aragonitic and bimineralic shells to assess the patterns governing bioaccumulation of metals in shells. To make the manuscript more accessible for readers, the revised version will more clearly present the trace element concentrations in calcitic, aragonitic and bimineralic shells and patterns governing bioaccumulation of trace elements in shells. The discussion will be divided into the three parts to make reading easier. Due to the large number of factors potentially controlling the metal concentrations in skeletons, the discussion will be first focused on the polymorphic form of calcium carbonate (with the context of the shell organic matter); then on potential environmental factors (based on literature data); and finally on a potential biological response based on tracking metal variability in shell size classes. The discussion will be focused on finding patterns of inter-species and inter-individual variations in the concentration of metals in studied shells. There is literature data regarding the concentration of some studied metals (mainly in sediments) around the study area (such as Rainbow at al. 2000; Rainbow at al., 2004; Szefer at al. 2002) and we will include this into the manuscript.

Comment: Line 372: There are numerous studies on Mg and Sr in carbonate, which uses and incorporation mechanisms, potential proxy-applications etc. need a better summary. Same for all other elements, the discussion of each element should be opened by the factors that control its incorporation into the carbonate. Also, as these are often not similar for calcite and aragonite, and especially since this study is focused on the incorporation of elements into different polymorphs, these two should be treated separately.

Response: This will be improved in the revised version of MS. We will put more attention on factors that control the incorporation of metals.

Comment: Line 375: The statistics should be provided in brackets. Also, please be

specific, how much?

Response: This will be improved in the revised version of MS.

Comment: Line 378: 'Mg was the dominant impurity', please rephrase, what do you mean?

Response: This will be improved in the revised version of MS.

Comment: Line 383: Please be specific, what species?

Response: This will be improved in the revised version of MS.

Comment: Line 397: What is the origin of the high Sr in barnacles?

Response: Unfortunately, so far, we have not been able to reach data on this issue. We will put in an effort to find the answer.

Comment: Line 405: The concentrations are sometimes given in mg/kg and sometimes in wt%, which is confusing. Please be consistent throughout the manuscript in figures, and this should be preferably $\mu$g/g.

Response: This will be improved in the revised version of MS.

Comment: Line 414: Please explain, what do you mean?

Response: The idea was that we found no patterns in metals concentration in shells at inter-individual and inter-species level. This will be improved in the revised version of MS.

Comment: Line 478: I wonder how would the data look if the metal concentrations are plotted as a function of the distance to the Vistula River mouth? Can you conclude that it is the contamination that controls the trace metal composition? A comparison to the species from non-contaminated water might help.

Response: This will be improved in the revised version of MS.

Comment: Line 481: Yes, and it is really necessary to add that the whole shells were measured. Therefore the mean values integrate these variations.

Response: This will be improved in the revised version of MS.

Comment: Line 486-489: Please rephrase. Also, of course, they varied but it is difficult to determine why.

Response: In the revised version of MS, the variability of elemental concentrations will be discussed with new details, and the discussion will have a new structure. The enrichment of environmental data will facilitate the inference of biological contribution to metals concentration in shells. We will not be able to recognize exactly what biological factor is responsible for variability, but we will present implications whether the inclusion of a given metal in the shell depends on the environment.

Comment: Line 497: 'chemical profiles' please rephrase, as far as I am aware no chemical profiles were made.

Response: 'Chemical profiles' will be replaced by 'obtained results'.

Comment: Line 479-509: This sections contains many redundant parts, and the discussion could be sharpened.

Response: This will be improved in the revised version of MS.

Comment: Line 510: In addition to relative increase or decrease in concentrations, also the variability in the elemental concentration for a size class should be considered (although I am not sure if the differences between size classes will be significant).

Response: This will be improved in the revised version of MS.

Comment: Line 520: Please be specific, which trace elements (please provide in brackets; similar cases can also be found in other parts of manuscript).

Response: This will be improved in the revised version of MS.

Comment: Line 527: Yes, but as mentioned I doubt this has anything to do with the size/age.

Response: This will be discussed with more details in the revised version of MS.

Comment: Figure 1: Please provide the full site names in the figure caption to abbreviations. What are the grey lines in the big panel (bathymetry?), please specify in caption as well.

Response: This will be improved in the revised version of MS. Yes, bathymetry.

Comment: Figure 2: This figure needs error bars. The analytical uncertainty should be shown here, as well as the variation of the mean i.e. the 2SD of the mean for each group and the respective n should be provided too. Also, what size classes were used for this? Is this the mean of a certain size class or the mean of all individuals, this needs definition in the caption. It may be more appropriate, too, instead of the mean of all individuals to depict the mean and the variation of each size class. I would also include information on the different polymorphs of each species. In general, I have no problems with the figures being black-and white only, but personally, I would try to improve the visual representation. In this case, maybe increasing the figure size to double and placing the legend within the top right corner could help separate a bit more out the different elements. Also, this is a detail, but to make it more intuitive, the grey filled symbols could be the aragonitic species, empty symbols the calcitic and half-filled for example bimineralic.

Response: These comments will be taken into account and the figure will be improved.

Comment: Figure 3: What is the x-axis? Please make the y-axis similar where possible, this is really difficult to read for me. Also, the information on the differences between size classes should be removed as at the moment there is too much information in this figure. The individual panels are missing sub-headings that should be also referred to in the manuscript text.

Response: This figure will be simplified and more easy to read.

Comment: Figure 4: Please appropriately label all panels as 'a,b,c, etc.' What do you mean by 'raw data as black dots'? (I see blue dots.) Please include polymorphs, analytical uncertainty, indicate the sizes for each category. Maybe better to put each species in a separate row. Why some size classes have values in between the size class number categories?

Response: These comments will be taken into account and the figure will be improved.

Comment: Figure 5: I find this figure difficult to follow, maybe there is a better way to illustrate the message? Should be 'dashed line' instead of 'broken line'. Why are some panels darker? Please specify in the caption.

Response: These comments will be taken into account and the figure will be improved.

References: Barker, S., Greaves, M., & Elderfield, H. (2003) A study of cleaning procedures used for foraminiferal Mg/Ca paleothermometry. Geochemistry, Geophysics, Geosystems, 4(9).

Bourget, E. In Barnacle Biology, Southward, A.J., Ed.; A.A., Balkema: Rotterdam, 1987.

Holcomb, M., DeCarlo, T. M., Schoepf, V., Dissard, D., Tanaka, K., & McCulloch, M. (2015) Cleaning and pre-treatment procedures for biogenic and synthetic calcium carbonate powders for determination of elemental and boron isotopic compositions. Chemical Geology, 398, 11-21.

Imai, N., Terashima, S., Itoh, S. & Ando, A. (1996) Compilation of analytical data on nine GSJ geochemical reference samples, Sedimentary rock series, Geostand. Newsl., 20(2), 165–216.

Inoue, M., Nohara, M., Okai, T., Suzuki, A., & Kawahata, H. (2004) Concentrations of Trace Elements in Carbonate Reference Materials Coral JCp‐1 and Giant Clam

JCt‐1 by Inductively Coupled Plasma‐Mass Spectrometry. Geostandards and Geoanalytical Research, 28: 411-416.

Kim, G., Alleman, L.Y., & Church, T.M. (2004) Accumulation records of radionuclides and trace metals in two contrasting Delaware salt marshes. Marine Chemistry 87: 87-96.

Lorens, R.B., & Bender, M.L. (1980) The impact of solution chemistry on Mytilus edulis calcite and aragonite. Geochimica et Cosmochimica Acta 44: 1265-1278.

Loxton, J., Najorka, J., Humphreys-Williams, E., Kuklinski, P., Smith, A. M., Porter, J. S., & Spencer Jones, M. (2017) The forgotten variable: Impact of cleaning on the skeletal composition of a marine invertebrate. Chemical Geology, 474, 45-57.

Piwoni-Piórewicz, A., Kukliński, P., Strekopytov, S., Humphreys-Williams, E., Najorka, J., & Iglikowska, A. (2017) Size effect on the mineralogy and chemistry of Mytilus trossulus shells from the southern Baltic Sea: implications for environmental monitoring. Environmental Monitoring and Assessment, 189(4), 197.

Rainbow, P.S., Fialkowski, W., Sokolowski, A., Smith, B.D., Wolowicz, M. (2004) Geographical and seasonal variation of trace metal bioavailabilities in the Gulf of Gdansk, Baltic Sea using mussels (Mytilus trossulus) and barnacles (Balanus improvisus) as biomonitors. Marine Biology 144, 271–286.

Rainbow, P.S., Wolowicz, M., Fialkowski, W., Smith, B.D., Sokolowski, A. (2000) Biomonitoring of trace metals in the Gulf of Gdansk , using mussels (Mytilus trossulus) and barnacles (Balanus improvisus). Water Research 34, 1823–1829.

Rueda, J.L. & Smaal, A.C. (2004) Variation of the physiological energetics of the bivalve Spisula subtruncata (da Costa, 1778) within an annual cycle. Journal of Experimental Marine Biology and Ecology 301: 141-157.

Strasser, M., Walensky, M., Reise, K. (1999) Juvenile-adult distribution of the bivalve Mya arenaria on intertidal flats in the Wadden Sea: why are there so few year classes?

Helgol. Mar. Res. 33, 45.

Szefer, P., Frelek, K., Szefer, K., Lee, C.B., Kim, B.S., Warzocha, J., Zdrojewska, I., Ciesielski, T. (2002) Distribution and relationships of trace metals in soft tissue, byssus and shells of Mytilus edulis trossulus from the southern Baltic. Environmental Pollution 120, 423–444.

Takesue, R.K., & van Geen, A. (2004) Mg/Ca, Sr/Ca, and stable isotopes in modern and Holocene Protothaca staminea shells from a northern California coastal upwelling region. Geochimica et Cosmochimica Acta 68: 3845-3861.

Walls, R.A., Ragland, P.C., & Crisp, E.L. (1977) Experimental and natural early diagenetic mobility of Sr and Mg in biogenic carbonates. Geochimica et Cosmochimica Acta 41: 1731-1737.

---

## Author Response (AR1)

We are grateful for the opportunity to make the corrections based on the detailed reviews received. All comments were very valuable to us. Considering the number and importance of both reviewer's comments, we have made extensive changes and thus prepared a new version of our manuscript. We have not highlighted changes in the text but tried to respond to the reviewers' comments below.

**Response to Reviewer #1 Inge van Dijk**

Major comments:

*I was surprised to read about the cleaning methods used in this study. When investigate chemical composition of foraminiferal shells, we use a much more intense cleaning method with oxidation and reduction steps (especially when specimens are collected in the field), to remove any organics as well as diagenetic coating. I wonder if e.g. only mechanical removal of any organisms present on the shell is enough to remove any (organic or chemical) trace completely, and if coatings are present on the shell after the described cleaning protocol. Are there studies comparing different cleaning techniques?*

**Response:** Concerning any surface contamination, the difference between the foraminifera and mollusk shells is the specific surface area, which is low in case of relatively large shells. The argument against chemical cleaning protocols was based on the observed preferential leaching of Mg (and, therefore, potentially many trace elements) out of carbonate skeletons during chemical cleaning (Loxton et al., 2017; Mannella et al., 2020). We have added a paragraph in a discussion section (lines 383 – 401) to draw the attention to the fact that in this study the whole shells were analyzed, along with organic matter.

*Furthermore, the amount of organic inside the CaCO₃ might have a huge effect on the elemental composition of the shell, e.g. on Na, and the contribution of organics might differ between species and CaCO₃ polymorphs, as also acknowledge by the authors (in l.54: 'species-specific organic matrix'). With the method described in this study, the organic matrix will also be analyzed. I would like to see this issue discussed in the revised version of the manuscript. Has there been any study on the chemical composition of the organics in different species of bivalves, albeit on the compounds of the matrix or by microscale-analysis of the shell with e.g. nanoSIMS on cross-sections? This should be at least mentioned in the discussion as a potential reason for the offset between species, if not discussed in full detail.*

**Response:** Removal of organics without mobilisation of any trace elements associated with CaCO₃ is not a task that is easy to achieve. There are good studies on this subject, e.g., (Barker et al, 2003; Holcomb et al., 2015; Loxton et al., 2017; Mannella et al., 2020). However, we do not believe there is a single accepted protocol for bivalve shells, that is tested and validated for a large range of trace elements. We, therefore, have opted for the analysis of the bulk composition instead of trying to isolate and analyse the CaCO₃ phases.

In the revised version of the manuscript, we emphasise that the contribution of the organic matter to the elemental variation is likely to be relatively minor considering that bivalve and barnacle shells contain, in general, less than 5% organic matter (Bourget, 1987; Marin and Luquet, 2004; Rueda and Smaal, 2004)

*The authors are comparing small and big adults and use the obtained data to look at ontogenetic effects. However, the authors also state this is a very variable environment, with big seasonal (and maybe yearly) changes. Overprinted on the size effect, there is a time effect: the bigger specimens have recorded events that the smaller specimens did not experience. I'm missing the longevity of the different species in the discussion of the results. For example: The size effect observed for A. improvisus (life span = 1 year) might simply be a seasonal signal in food supply (and thus maybe growth rate) or*

*physio- or chemical parameters like seawater temperature. If the samples are taken in end of summer (sample date not mentioned in the manuscript, should be added), it would explain why the Mg of the larger specimens is lower: lower temperatures lead to lower incorporation of Mg and these larger specimens likely experienced the winter period, while the smaller specimens maybe spawned in spring. As for the species with longer lifespans (of 10-12 years), could the decrease in element incorporation with size (thus, for older specimens) be due to increased heavy metal output of the Vistula river over the last years? Is there any (historical) data on this?*

**Response:** In the revised version of the manuscript the discussion has been extended about the size effect on observed patterns of elemental levels with emphasis on environmental influence on it. As a result, patterns related with the biological effect of organisms have been more clearly justified. Unfortunately, we have been unable to present any data on the recent heavy metal output of the Vistula river, but we have added the paragraph 4.3, which contains environmental data describing the Gulf of Gdańsk in the context of elemental concentrations in shells.

*The authors see some difference between size classes, they conclude that in general smaller specimens have increased trace metal incorporation. Can this also be due to absorption of elements or diagenetic precipitation on the outside of the shell, compared to the more pristine CaCO3 below the surface, leading to a surface/volume effect? E.g. larger specimens have thicker shells, and thus lower surface over area ratio?*

**Response:** This potential issue has been mentioned in the discussion (lines 454 – 456).

*I would also like to see habitat depth included in the discussion. The authors are assuming the shell chemistry of the different organisms tested are all reflecting either food or ambient seawater chemistry, but some organisms are living on hard substrates, while other live a few cm in the sediment, and the most extreme one (Mya arenaria) can live 20-30 cm in the sediment. The latter species would have a totally different "ambient seawater conditions", as it is exposed to interstitial water that is very likely to have totally different chemical signature then the overlying water. It would probably be in contact with e.g. much higher Mn concentrations. This is not reflected in the shell chemistry, so maybe this species does not take up elements from the seawater, but more from food intake? For your purpose it would be best to have also some kind of idea of the (evolution) of trace metal concentrations in seawater over time. Is there any chemical data on the seawater available from this area? E.g. about the metal concentrations close to the Vistula river (l. 478)? Is there data showing that station GN is indeed increased in heavy metals compared to the other stations? These other stations are located in a bay area, making it possible the residence time of the water is higher, and there might be an actual increase in the metal concentration here.*

**Response:** The sediment type, feeding strategy and environmental sources of metals are important factors affecting the concentration of metals in shells and were discussed in the revised version of manuscript. We have added information about sediment type in the study area and have put more emphasis on discussing feeding strategy in the context of metal bioavailability.

*In retrospect, for the main goal of the study, it maybe would have been better to not analyze full shells, but make small aliquots/subsamples by e.g. drilling the shell. Is there any data (in literature) on small-scale variation in the shells of (some of these) species?*

**Response:** Our strategy of investigating the whole shells has been a deliberate choice in order to achieve better detection limits (comparing to e.g. LA-ICP-MS), and to analyze many individuals of different species instead of focusing on a thorough analysis of single individuals. We have presented data on the concentration of 12 metals in shells of mussels and barnacles from the Baltic populations (in total over 2,300 measurements), which have not been previously studied.

*In my opinion, at the moment, you have a combination of too many variables: CaCO3 polymorph, different stations environmental variables (incl. unknown chemical compositions of the seawater), size effect and the vital effect (calcification pathway) of the organisms. It becomes very difficult to disentangle different drivers of shell chemistry, which means you have to be more careful in your conclusions, or at least convince readers which variables are minor/neglectable. I think some variables, like the different sampling stations, can be convinced as being minor, by showing chemical variability or the hydrological situations between the stations.*

*The authors often point out the strong seasonality in this region, section 2.1 and through the manuscript, e.g. l. 481-483. Maybe it is possible to add a (supplementary) figure to section 2.1, if needed compiled from literature data, about the environmental variability in this area, to show differences in physio-chemical parameters. This way, readers, like myself, that are not familiar with the study area can have a good overview of the (yearly) environmental variability in this area.*

**Response:** To make the manuscript more accessible for readers, the aims of this study have been clarified (lines 119 – 128) and the discussion has been divided into the three parts: CaCO$_3$ polymorph type and elemental concentrations; Size classes and potential biological impact on elemental concentrations; Environmental factors and elemental concentrations. As conclusions, we have distinguished patterns of inter-species and inter-individual variations in the concentration of elements in studied shells, which are associated with biological and environmental control over the biomineralization process.

Minor comments:

(Since I have a lot of major points for the discussion section, I give minimal textual changes, since I believe the manuscript, especially the discussion, will probably greatly change after revision.)

Throughout the manuscript:

*-Change 'Mg/Ca ratio' to 'Mg/Ca'.*
**Response:** Modified where sensible.

*-Check manuscript for (double) bracketing issues, for instance in l.207: '(Darwin, 1854) (Arthopoda, Maxillopoda) ' should be e.g. '(Arthopoda, Maxillopoda; Darwin, 1854). Also lines 217, 228, etc. For l.131 and l. 515: reference should not have brackets.*
**Response:** Corrected where applicable.

Abstract:
*-The abstract as it reads a bit stiff. Please consider rewriting this section. For example, l. 28-29 on sample location can be merged with the first sentences, while line 29-30 is an explanation of the method, which should be either removed, or shortened, in my opinion.*
**Response:** The abstract has been rewritten.

*-l. 26-27: 'The potential impact of environmental factors on the observed elemental concentrations in the studied shells is discussed': Is this really the case? Since there is no data on the environmental parameters presented, it is difficult to discuss the data in this framework.*
**Response:** Although we have not measured the concentrations of elements in the environment, we believe the discussion about their impact on the composition of the shell is challenging, yet very valuable. In the newer version of the manuscript

more information based on the literature data about the environmental characteristics were added, and we based the discussion on it.

Introduction:

-l.64-65: 'crystal layers are precipitated successively at regular periodicities,' is not true for all marine calcifyers, like Foraminifera. make it clear when you switch from all marine calcifyers to marine invertebrates.

**Response:** This has been improved.

-l.88-90: maybe add Stanley, 2008, it is a nice overview paper. Stanley, S. M. (2008). "Effects of global seawater chemistry on biomineralization: past, present, and future." Chemical reviews 108(11): 4483-4498.

**Response:** This has been included.

-l.144: remove . after shells

**Response:** This has been improved.

Method section:

-l. 263: When were the samples taken?

**Response:** The samples were collected in May 2013 and June 2014. This information has been presented in Table 1.

-l. 295 What was the Ca concentrations in the sample solutions? 100ppm or varying?

**Response:** Further details have been added to the revised manuscript.

-l.297: What were the accuracy and precision of the measured elements?

**Response:** We have added such details to the method section (lines 256 – 260).

Discussion section:

-I would advise to divide the discussion session in smaller paragraphs to increase readability.

**Response:** The discussion has been divided into the three parts to make reading easier.

-I would like to see variables as life span, habitat depth and organic material in the shell (see above) included in the discussion.

**Response:** This has been added and existing parts improved.

-l. 372: You obtained specimens with the same polymorph from two contrasting temperatures: i.e. aragonite Cerastoderma glaucum (16.9∘C), Limecola balthica (4.6∘C), Mya arenaria (16.9∘C), I would like to see a discussion on the (absence of) temperture effect Sr incorporation, which is currently lacking in the manuscript, while it is being discussed for Na and salinity.

**Response:** The environmental variables have been discussed in more details in the revised version of the manuscript.

-l. 478: ref? Are there any studies on this?

**Response:** The literature data regarding the concentration of some studied metals around the study area (such as Rainbow at al. 2000; Rainbow at al., 2004) and sediment type has been included into the manuscript.

*-l. 483: suggest to change 'animal 'into ,organism'.*

**Response:** This has been changed as suggested.

*-Fig 2 and 4: indicate which polymorph of CaCO3 is used, like Fig. 3.*

**Response:** Figures and tables have been improved to make them easier to read.

*-Fig. 3: where possible, please use the same scaling for the y-axis for comparability, e.g. y axis of Mg for aragonitic species.*

**Response:** Figures and tables have been improved to make them easier to read.

**Response to Reviewer #2 Anonymous Referee**

*Fundamental information regarding the measurements and concentration calculations is missing or unclear, and the analytical uncertainties and repeatability based on appropriate reference materials are also lacking. In addition, in my opinion, the sample preparation and pre-cleaning procedures are questionable. The authors must prove the validity and explain their analytical procedures, and take into account analytical uncertainties when presenting or interpreting their data.*

**Response:** The analytical details have been described in more detail in the revised version of manuscript.

*Furthermore, as detailed below, I have problems understanding how whole shell bulk measurements may be used to assess the role of ontogeny or even environmental variations. By using entire shells, the authors 'average 'the composition of the growth lines precipated during earlier and later stages of life, as well as the composition of growth lines built during different seasons or under different environmental conditions. Thus, I am not sure that by comparing bulk values from smaller vs. bigger (younger vs. older) individuals it is possible to determine whether environmental or ontogenetic controls drive the composition of the shell. Simply, the differences between the mean bulk values would depend on the elemental variability encompassed in the shell, which would depend on individuals 'growth and environmental conditions experienced. The mean bulk values from older individuals integrate large intra-shell variabilities, while in younger individuals smaller intra-shell variabilities, but I think that with this design it is difficult to disentangle the underlying controls on the elemental composition of the carbonates.*

*Moreover, the problem of using bulk also limits the interpretation of the data when it comes to the different polymorphs, and particularly this is the case for the bimineralic bivalve. Here, the authors may only conclude whether the composition of the mixture is different to other species building pure calcite vs. aragonite. However, it does not answer the question whether the composition of the calcitic part or aragonitic part within fundamentally differs and how much, which I think is the relevant question here. When discussing the composition of the bimeralic bivalves the authors could, at least, attempt to estimate the contribution from each polymorph to the mixture, and discuss the implications.*

*Moreover, I am wondering why the entire shells were crushed? Surely, >100 mg is not required for the analyses, as concentration measurements are typically done on <mg level. Why did the authors decide to measure the entire shell instead of e.g. a profile across the shell or different growth bands? Such approach, I believe, would be much better for defining an ontogenetic trend, and could also provide some insights into the intra-shell variability. The intra-shell variability, in particular, would be very meaningful to assess before any mean bulk values are used for interpretation of ontogenetic or environmental signals – i.e. how heterogenous are the shells, what is the driver, is it random or not, how*

*big is the variation and what it reflects? I think it is really a shame this was not considered beforehand as a great amount of information from the shells is lost when measuring the whole shells rather than specific parts. Furthermore, I am not convinced that comparison of bulk large vs. small individuals, in this case, answers the question whether ontogenetic trend drives the elemental variability. For numerous calcifiers group, the partioning of elements between seawater and the carbonate is within a certain range band 'baseline' which is principally determined by their calcification mechanisms and mineralogy, and then this variability of the 'baseline' may be driven by environmental factors. In such case, simply by a probability, larger individuals would have lived longer vs. smaller individuals and thus likely witnessed during their life time more environmental fluctuations (e.g. temperature, nutrients, pH, O2, etc.). Thus, when using an average of an entire shell, it is reasonable to assume that the mean of the shell integrates larger intra-shell and therefore elemental variations in the older individuals in contrast to the younger, simply because they experienced more changes over their life. I believe that this is also quite apparent in Fig. 3. How may one, therefore, discriminate between ontogeny vs. environmental variability?*

**Response:** Our strategy of investigating the whole shells has been a deliberate choice in order to achieve better detection limits (e.g. LA-ICP-MS), and to analyze many individuals of different species instead of focusing on a thorough analysis of single individuals. We have presented data on the concentration of 12 metals in shells of mussels and barnacles from the Baltic populations (in total over 2,300 measurements), which have not been previously studied. The manuscript has been significantly modified in accordance with the reviewers' suggestions and now provides a better discussion of the major points raised above.. Furthermore, the discussion now includes more detailed environmental dataset (based on literature data) to draw more robust conclusions about the biological effect.

As mentioned in the responses to Reviewer #1: Removal of organics without mobilisation of any trace elements associated with $CaCO_3$ is not a task that is easy to achieve. There are good studies on this subject, e.g., Barker et al (2003), Holcomb et al (2015), Loxton et al (2017). However, we do not believe there is a single accepted protocol for bivalve shells that is tested and validated for a large range of trace elements. We, therefore, have opted for the analysis of the bulk composition instead of trying to isolate and analyse the $CaCO_3$ phases. In the revised version of the manuscript we have emphasised that the variation observed could also be due to the presence of organic material within the carbonate structure.

*In my opinion, the authors need take into account these problems, before any interpretations can be made. While a great deal of information is unfortunately lost by using average values, and I think the authors really have to reconsider the interpretations that can be made from this data and discuss their limits, I do acknowledge the authors' efforts for measuring numerous individuals, which I do not think is often done, and perhaps a point to that could be better taken advantage of. Just as a suggestion, maybe, this could be of use for defining the 'typical range' for each element for each species in the Gulf of Gdansk, which could be then compared to literature values from same / similar species in other parts of the world with very different settings. If possible, I think it would be interesting to see how the general elemental concentrations and variability compares between regions or not, and could be of use when constraining environmental influences on the biomineral composition. In addition, I would also like to see a comparison between the different sampling sites within the Gulf of Gdansk. While on one hand it could be perhaps assumed that the differences between the sites are negligible, this is a very dynamic environment, and it might be that spatio-temporal variations account, at least partially, for some of the observed variabilities.*

**Response:** These issues have been discussed more clearly in the revised version of manuscript.

*One thing that has surprised me the most about this study is that, despite the careful organism sampling strategy, the authors did not consider collecting and measuring water samples. In my opinion, this should come first in this kind of*

*studies, and something I was expecting to see, and thus a real shame it was not done, especially since the authors had the opportunity to do so (and elemental analyses on water samples are relatively more straightforward that on carbonates). Data on seawater chemistry is critical for the calculation of partitioning coefficients, which could ease the interpretation of the results from different sites (in the case that the chemistry at the different sites strongly varies). While it may be a tall task to ask for the measurements at this stage, the authors should, at least, compile the available information on local concentrations of elements in seawater (including additional physico-chemical characteristics), and estimate the partitioning coefficients for each element for the different species.*

**Response:** It is hard to not agree with the reviewer on above aspect. And because we have not measured the concentration of elements in the environment, the discussion about its impact on the composition of the shell is challenging, yet very valuable. In the newer version of the manuscript we have put more emphasis on the environmental characteristics based on the literature data. This also include sediment type, feeding strategy and environmental sources of metals which are all important factors affecting the concentration of metals in shells. All these issues were discussed in more details in the new manuscript.

Specific comments:
*Line 1-2: I would suggest to reconsider the title language ' –composition 'and 'com- posed', as well as 'different 'twice in the same sentence, this is not orderly.*

**Response:** Based on the reviews received and the modification that we have introduced in the manuscript, the title has been changed to: The patterns of elemental concentration (Ca, Na, Sr, Mg, Mn, Ba, Cu, Pb, V, Y, U and Cd) in shells of invertebrates representing different $CaCO_3$ polymorphs: a case study from the brackish Gulf of Gdańsk (the Baltic Sea).

*Line 32: 'Mg > Sr > Na 'this needs a written definition first.*

**Response:** This was improved.

*Line 195-197: Here, it would be particularly useful to provide concrete numbers on the local carbonate chemistry (other than Ω). Ideally, this should have been measured upon the collection of the specimens from in situ water samples, however, if this is not available the authors could at least summarize the information from the literature. An overview table with the physico-chemical characteristics of the local waters (including temperature and salinity trends etc., carbonate chemistry as well as the elemental composition), would be particularly useful.*

**Response:** As mentioned above, available literature data about environmental factors has been included in the revised manuscript.

*Line 267: Why no water samples were collected?*

**Response:** We agree that environmental research would be very useful in the discussion, yet unfortunately we do not have them. Therefore, we have used the data available in the literature for the discussion of our results.

*Line 282-286: I have difficulties following this protocol and serious doubts on its effectivity and validity. Previously, the authors state that the periostracum was first physically removed. This is good and indeed important as it constitutes a large amount of organic material, which is difficult to treat chemically without having an impact on the carbonate. However, organic rests might still be present on the inside of the shell for example from the mantle, and foremostly in the pore spaces. Thus, physical cleaning is insufficient, and at least at a powder stage it is a generally established routine to apply a cleaning protocol step, consisting of oxidation of organics by buffered hydrogen peroxide (Barker et al., 2003 G3*

*4, 8407). As far as I am aware, this protocol or close adaptations are commonly applied to a wide range of calcifiers from forams to corals, bivalves and even brachiopods. In this sentence the authors indeed mention the use of H2O2, but only after the dissolution of the sample, which logic I cannot follow. All in all, I do not think that this is the correct way to treat carbonates samples, and would strongly recommend to first demonstrate the validity of this protocol (if the authors insist on using it, or follow a more broadly used protocol such as that of Barker et al., 2003).*

**Response:** We have discussed these issues in more detail in the revised manuscript. However, we do suggest, as Inoue et al (2004) did that "plausible pre-treatment method [for the removal of organics] is yet to be established". With full appreciation of the importance of the protocols discussed and tested by Barker et al (2004) we believe that the benefits of any chemical treatment still remain controversial, see, for example, discussion in Holcomb et al (2015) and Loxton et al (2017). Mannella et al. (2020) showed that the suitability of chemical pre-treatments for organic matter removal from carbonate matrices should be evaluated on a case-by-case basis and, in case of relatively low organic content, should be avoided. We have discussed the potential contribution of organic matter in the revised version of manuscript.

*Also, when it comes to ontogenetic trends, let's take for example bivalves and specifically Mytilus, as far as I am aware, broadly speaking their shell growth follows von Bertalanffy growth curve (see e.g. ig. 3; Steffani & Branch, 2003; Mar Ecol Prog Ser 246, 197-209), which is common for many calcifiers. This means that during the very early shell formation the carbonate precipitation is relatively faster, which for the incorporation of numerous elements translates into kinetic effects. It is thus the geochemical composition of the umbo and the first growth lines vs. the latter growth lines (the ones at the growth 'plateau') that form the greater part of the valve that is commonly at- tributed to being driven by ontogeny. Potentially, in the case of the very small and thus very young individuals, their geochemical composition may reflect one environmental condition e.g. certain season and one ontogenetic stage i.e. the one dominated by kinetic factors, but I am not sure this can be directly compared to older individuals which mean elemental composition then reflects different ontogenetic stages (with potentially different contribution of each to the bulk), and broad range of seasons. Or am I missing something?*

**Response:** As mentioned earlier, this part of the discussion has been elaborated based on a more detailed environmental background. This, in our opinion, strengthens the inference of potential biological control on the elemental concentration in shells. Despite the influence of many factors, we have observed statistically significant patterns of metals concentration in shells.

*Line 294: What type of solutions? What do you mean by matrix-matched – one solution for each carbonate polymorph? Please provide more details.*

**Response:** The method has been described in more detail in the revised manuscript.

*Line 300: Why were the standards not treated the same way as samples? First, I do not think it is acceptable that the authors do not process the standards and the samples in the same way, and second, I do not think that the standards are representative and should be compared to these samples. The authors need to provide the measured absolute values (as well the relative standard deviation over the analysis period at least) of comparable biogenic standards such as JCp-1 or JCt-1, or similar internationally accepted alternatives.*

**Response:** The authors appreciate that using biogenic standards would potentially be preferable; however, those standards were not available to the authors at the time of analysis. On the other hand, complete digestion eliminates potential uncertainty that might originate from potentially incomplete conversion of organic matter typically performed when digesting by $HNO_3$ only (Inoue et al 2004). The use of reference limestone and dolomite for the control of the analysis

(without considering the digestion step) is fully justified for the digestion protocol used (HF+HNO₃ with evaporation and matrix modification to HNO₃ solution of the same concentration). The digestion step includes $HNO_3+H_2O_2$ mixture, which is perceived to be suitable for digesting $CaCO_3$-based materials with low amount of non-refractory organic material. For comparison, a well-cited paper on the composition of JCp-1 or JCt-1 standards (Inoue et al 2004) employed a milder treatment of $HNO_3$ only at room temperature. We agree that a full method validation employing homogenized samples of clams containing high level of organic matter (in addition to biogenic reference materials, which potentially do not cover the natural range in terms of organic matter content / reactivity) would be desirable, but this must be a subject of a separate study.

*Also, regarding the methodology, I am wondering how were the obtained counts converted into concentrations; e.g. did the authors use a calibration line for this or standard-bracketing?*
**Response:** Calibration has been performed typically using 5 points covering the range of concentrations.

*Did you normalise all measurements to a stable concentration of a selected element, e.g. Ca?*
**Response:** We have not normalized measurements to Ca, but we have discussed the variability of Ca concentrations.

*What was the precision of the individual analyses, and the long-term reproducibility? How many times was each sample measured? Line 305 'most trace elements – 'which elements were measured in He mode and which not? The authors must provide these details with rigour. Line 307 'periodic analyses 'do you mean the standards were not measured along with the samples in a sequence? I have serious doubts on these analytical protocols, and especially do not consider it a good practice to not include standards along with samples in a run.*
**Response:** We have updated the Material and Methods section, in the revised manuscript, to include the following details: ICP-OES: The accuracy and reproducibility of the analyses were checked using two calcium carbonate-rich certified reference materials (CRMs): JLs-1 Limestone and JDo-1 Dolomite (both from the Geological Survey of Japan) prepared by total digestion method (using hydrofluoric acid). The reference materials were diluted to match the concentrations of Ca in sample solutions. Ca, Mg and Sr concentrations were found to be within the uncertainty (1 standard deviation) of the reported values (Imai et al. 1996). Limits of quantification (LOQ) in solution for ICPMS were generally determined as a concentration corresponding to ten times standard deviation of the signal obtained by analysing 5% HNO3 solution (6–7 times) in each individual run. ICP-MS was run in helium (He) mode (5 ml min−1 He, 99.9995% purity) for lighter trace elements (V, Mn, Cu, Y and Cd) to minimize the molecular interferences from plasma and solution components and Ca from samples. The accuracy and reproducibility were checked by analyses of JLs-1 and JDo-1 before and after every batch of samples. The results obtained for all elements (Table 1) were within the uncertainty (2.5 SD) of the recommended values (Imai et al. 1996). Accuracy of Pb determination cannot be checked using these CRMs because of the large spread of reference values probably due to insufficient homogeneity of Pb distribution in these samples. Based on the analyses of CRMs and matrix-matched solutions, the maximum analytical error for the typical range of concentrations in the shells can be estimated (in relative percentage) as 1.5% for Ca, Mg and Sr; 3% for Ba; 20% for Cu and U; and 4–10%for all other elements.

*Line 311: I would really welcome some visual representation for this – i.e. pictures of the different species, maybe with the different ontogenetic stages for each. It is really shame this is not provided; the authors study various interesting species, which offers an opportunity to include visually appealing picture figures, which is not used. Perhaps this is too*

*much to ask, but given that the species build very different carbonate types and I assume microstructures, scanning electron microscope images could also be very relevant and interesting here.*

**Response:** Unfortunately, we have not had the opportunity to make scanning electron microscope images, but we include macroscopic images presenting the studied species (Figure 1).

*Line 321: Throughout the Results section the figures are referred to very sporadically only, and there are several instances that a value is given and a statement is made, however the figure is not referred to afterwards. Foremostly, all individual panels of the figures need sub-categories (e.g. a, b, c, etc. please check the Biogeosciences format style), and need to be mentioned where the individuals results are being discussed.*

**Response:** This has been improved in the revised version of manuscript.

*Line 322: I am not sure what the authors mean here, please rephrase.*

**Response:** This has been improved in the revised version of manuscript.

*Line 327: I would say it is more appropriate to use µg/g rather than mg/kg.*

**Response:** This has been improved in the revised version of manuscript.

*Line 328: When concluding that some elements were 'generally present at higher concentration' or lower please also provide the concrete numbers in the text, here, but also in further parts of this section it is missing.*

**Response:** This has been improved in the revised version of manuscript.

*Line 334: What do you mean by 'lack of ontogenetic trend'?*

**Response:** This has been improved in the revised version of manuscript to be clearer.

*Line 371: The entire Discussion section needs major revision, and foremostly substantial reorganisation in order to make it more suitable to the readers and a wider audience. I am aware that dealing with many different variables like several elements, size classes, species and carbonate polymorphs is not easy, but the authors really need to find a better way for presenting their findings and extracting their 'main mes- sage points' to the audience. At the moment I find the Discussion very broad and, to me, it does not provide clear answers to the research questions. I am afraid that often problems are addressed that cannot be resolved by the present dataset. I would say that it is better if one or two key points are discussed in-depth rather than touching on the surface many (these may still be mentioned, but the in a more concise form, with focus on the key points). The structuring is also relevant for the other parts of the manuscript and especially the Results section. I would start with ensuring that were possible, the geochemical data is presented in a more systematic manner. The Discussion could benefit from being divided into different subsections, where different aspects are being discussed. The data quality and limitations need discussing, as well each of the different factors controlling the incorporation of the elements into the carbonate (preferably in different subsections), a comparison to other studies, and the implications of the presented findings (for e.g. biomineralisation, application as recorders of environmental conditions). At this stage, it is difficult for me to make a concrete suggestion on how to subdivide this, the authors need to see what works best when structuring the Discussion and the message they would like to convey. I would also suggest to separate the Results section, perhaps by species could work well for this part.*

**Response:** The manuscript has been reorganized to be more accessible for readers. The aims of this study have been clarified (lines 119 – 128). The discussion has been focused on finding patterns of inter-species and inter-individual

variations in the concentration of elements in studied shells, and was divided into the three parts: CaCO₃ polymorph type and elemental concentrations; Size classes and potential biological impact on elemental concentrations; Environmental factors and elemental concentrations.

*Line 372: There are numerous studies on Mg and Sr in carbonate, which uses and incorporation mechanisms, potential proxy-applications etc. need a better summary. Same for all other elements, the discussion of each element should be opened by the factors that control its incorporation into the carbonate. Also, as these are often not similar for calcite and aragonite, and especially since this study is focused on the incorporation of elements into different polymorphs, these two should be treated separately.*

**Response:** This has been improved in the revised version of the manuscript. We put more attention on the factors that control the incorporation of elements.

*Line 375: The statistics should be provided in brackets. Also, please be specific, how much?*

**Response:** This has been improved in the revised version.

*Line 378: 'Mg was the dominant impurity', please rephrase, what do you mean?*

**Response:** This has been improved in the revised version.

*Line 383: Please be specific, what species?*

**Response:** This has been improved in the revised version.

*Line 397: What is the origin of the high Sr in barnacles?*

**Response:** This issue has been improved and discussed in the revised version of the manuscript.

*Line 405: The concentrations are sometimes given in mg/kg and sometimes in wt%, which is confusing. Please be consistent throughout the manuscript in figures, and this should be preferably µg/g.*

**Response:** This has been improved in the revised version.

*Line 414: Please explain, what do you mean?*

**Response:** This has been improved in the revised version.

*Line 478: I wonder how would the data look if the metal concentrations are plotted as a function of the distance to the Vistula River mouth? Can you conclude that it is the contamination that controls the trace metal composition? A comparison to the species from non-contaminated water might help.*

**Response:** In the revised version of the manuscript we have included more information about the environmental background. Unfortunately, we do not have the data that could be used to create a mentioned figure, but we have given more attention to this in the discussion. We also have included the data in Table 4, that show the concentration of elements in the shells of the studied organisms from regions other than the Gulf of Gdańsk.

*Line 481: Yes, and it is really necessary to add that the whole shells were measured. Therefore, the mean values integrate these variations.*

**Response:** This has been improved in the revised version of the manuscript.

Line 486-489: Please rephrase. Also, of course, they varied but it is difficult to determine why.

**Response:** In the revised version of the manuscript, the variability of elemental concentrations has been discussed with the new details, and the discussion has a new structure. The enrichment of environmental data facilitates the inference of the biological contribution to elemental concentration in shells. We were not able to define exactly which factor is responsible for the variability, but we present suggestions whether the incorporation of a given metal in the shell is more influenced by biological or environmental factors.

*Line 497: 'chemical profiles' please rephrase, as far as I am aware no chemical profiles were made.*
**Response:** This has been improved in the revised version of the manuscript.

*Line 479-509: These sections contains many redundant parts, and the discussion could be sharpened.*
**Response:** This has been improved in the revised version of the manuscript.

*Line 510: In addition to relative increase or decrease in concentrations, also the variability in the elemental concentration for a size class should be considered (although I am not sure if the differences between size classes will be significant).*
**Response:** This was improved in the revised version of the manuscript.

*Line 520: Please be specific, which trace elements (please provide in brackets; similar cases can also be found in other parts of manuscript).*
**Response:** This has been improved in the revised version of the manuscript.

*Line 527: Yes, but as mentioned I doubt this has anything to do with the size/age.*
**Response:** This has been improved in the revised version of the manuscript.

*Figure 1: Please provide the full site names in the figure caption to abbreviations. What are the grey lines in the big panel (bathymetry?), please specify in caption as well.*
**Response:** This has been improved in the revised version of the manuscript. Yet in case of the sampling site names, they are given symbol by us therefore they have no full names.

*Figure 2: This figure needs error bars. The analytical uncertainty should be shown here, as well as the variation of the mean i.e. the 2SD of the mean for each group and the respective n should be provided too. Also, what size classes were used for this? Is this the mean of a certain size class or the mean of all individuals, this needs definition in the caption. It may be more appropriate, too, instead of the mean of all individuals to depict the mean and the variation of each size class. I would also include information on the different polymorphs of each species. In general, I have no problems with the figures being black-and white only, but personally, I would try to improve the visual representation. In this case, maybe increasing the figure size to double and placing the legend within the top right corner could help separate a bit more out the different elements. Also, this is a detail, but to make it more intuitive, the grey filled symbols could be the aragonitic species, empty symbols the calcitic and half-filled for example bimineralic.*
**Response:** These comments have been taken into account and the figures have been improved.

*Figure 3: What is the x-axis? Please make the y-axis similar where possible, this is really difficult to read for me. Also, the information on the differences between size classes should be removed as at the moment there is too much information in this figure. The individual panels are missing sub-headings that should be also referred to in the manuscript text.*
**Response:** This figure has been simplified to be easier to read.

*Figure 4: Please appropriately label all panels as 'a,b,c, etc.' What do you mean by 'raw data as black dots'? (I see blue dots.) Please include polymorphs, analytical uncertainty, indicate the sizes for each category. Maybe better to put each species in a separate row. Why some size classes have values in between the size class number categories?*
**Response:** These comments have been taken into account and the figures have been improved.

*Figure 5: I find this figure difficult to follow, maybe there is a better way to illustrate the message? Should be 'dashed line' instead of 'broken line'. Why are some panels darker? Please specify in the caption.*
**Response:** These comments have been taken into account and the figures have been improved.

**References used in the above responses to the reviewers' comments:**

Barker, S., Greaves, M., and Elderfield, H.: A study of cleaning procedures used for foraminiferal Mg/Ca paleothermometry, Geochem. Geophys. Geosyst., 4, 8407, https://doi.org/10.1029/2003GC000559, 2003.

Bourget, E.: Barnacle shells: composition, structure and growth, in: Barnacle biology, edited by: Southward, A.J., A.A.Balkema, Rotterdam, The Netherlands, 267–285, https://doi.org/10.1201/9781315138053-14 1987.

Holcomb, M., DeCarlo, T.M., Schoepf, V., Dissard, D., Tanaka, K., and McCulloch, M.: Cleaning and pre-treatment procedures for biogenic and synthetic calcium carbonate powders for determination of elemental and boron isotopic compositions, Chem. Geol., 398, 11–21, https://doi.org/10.1016/j.chemgeo.2015.01.019, 2015.

Inoue, M., Nohara, M., Okai, T., Suzuki, A., and Kawahata, H.: Concentrations of trace elements in carbonate reference materials Coral JCp-1 and Giant Clam JCt-1 by inductively coupled plasma-mass spectrometry. Geostand. Geoanal. Res. 28, 411-416, https://doi.org/10.1111/j.1751-908X.2004.tb00759.x, 2004.

Loxton, J., Najorka, J., Humphreys-Williams, E., Kuklinski, P., Smith, A. M., Porter, J. S., Spencer Jones, M.: The forgotten variable: Impact of cleaning on the skeletal composition of a marine invertebrate, Chem. Geol., 474, 45–57, https://doi.org/10.1016/j.chemgeo.2017.10.022, 2017.

Mannella, G., Zanchetta, G., Regattieri, E., Perchiazzi, N., Drysdale, R. N., Giaccio, B., Leng, M. J., and Wagneret, B.: Effects of organic removal techniques prior to carbonate stable isotope analysis of lacustrine marls: A case study from palaeo-lake Fucino (central Italy). Rapid Commun. Mass Spectrom., 34, e8623, https://doi.org/10.1002/rcm.8623, 2020.

Marin, F. and Luquet, G.: Molluscan shell proteins, Compt. Rend. Palevol, 3, 469–492, https://doi.org/10.1016/j.crpv.2004.07.009, 2004.

Rainbow, P. S., Wolowicz, M., Fialkowski, W., Smith, B. D., and Sokolowski, A.: Biomonitoring of trace metals in the Gulf of Gdansk , using mussels (*Mytilus trossulus*) and barnacles (*Balanus improvisus*), Water Res., 34, 1823–1829, https://doi.org/10.1016/S0043-1354(99)00345-0, 2000.

Rainbow, P. S., Fialkowski, W., Sokolowski, A., Smith, B. D., and Wolowicz, M.: Geographical and seasonal variation of trace metal bioavailabilities in the Gulf of Gdansk, Baltic Sea using mussels (*Mytilus trossulus*) and barnacles (*Balanus improvisus*) as biomonitors, Mar. Biol., 144, 271–286, https://doi.org/10.1007/s00227-003-1197-2, 2004.

Rueda, J. L. and Smaal, A. C.: Variation of the physiological energetics of the bivalve *Spisula subtruncata* (da Costa, 1778) within an annual cycle, J. Exp. Mar. Bio. Ecol., 301, 141–157, https://doi.org/10.1016/j.jembe.2003.09.018, 2004.

---

## Referee Report (RR1)

I would like to thank Piwoni-Piórewicz and co-authors for providing a new and updated version of the manuscript. However, for future reference, please indicate line numbers with all your changes in your response to reviewers comments, and provide a manuscript with tracked changes / or highlighted changes in text. This way, the reviewer can quickly see the changes and improvements made by the author (also changes based on comments from another reviewer/editor), just a response like: "These issues have been discussed more clearly in the revised version of manuscript." is not sufficient in my opinion.

Some specific comments on the response to the author document:

- In the 'Author response' they do state that there is no protocol they could have used ('we do not believe there is a single accepted protocol for bivalve shells, that is tested and validated for a large range of trace elements'), however, if the whole aim of your study is to compare such a wide range of elements in different shells, maybe some method development should have taken place before. Even (relatively) simple comparison of shell fragments of a single shell with different cleaning treatment (versus no cleaning) applied, could have make their dataset more robust. I highly suggest that for future work cleaning methods as well as the impact of organics should be investigated!

-Authors: In the revised version of the manuscript, we emphasise that the contribution of the organic matter to the elemental variation is likely to be relatively minor considering that bivalve and barnacle shells contain, in general, less than 5% organic matter (Bourget, 1987; Marin and Luquet, 2004; Rueda and Smaal, 2004)

- Reviewer: 5% in weight? Is there any idea about the concentration of certain elements in this organic matter? Even if it is 5% of the total weight, if the concentration of e.g. Na is very high, this still leads to a biased E/Ca for the CaCO3. Also you refer to Takesue et al., 2008, who found e.g. 19% organic matter.

-After comments by both me and the other reviewer, I still do not see any discussion on the fact that lager (and thus older) shells have also experienced more time, and more environmental fluctuations. Can the authors comment on this directly?

Minor comments:

(General comment) The manuscript is incredibly dense in information. Of course it is great to have numerous references and explanations for observed patterns, but also makes it sometimes difficult to absorb the main take home messages. Be careful for repetition, for instance, please check overlap between paragraph 2.2 and 4.3.

Line 17: please add clearly you work on bulk material

Line 27: "Moreover, the elemental concentrations tend to be lower in larger than in smaller shells." This is confusing, suggesting to change to something like "lower in the large size class compared to the smaller size class

Line 41-43: changes in font size

Line 44: consider changing metals to trace metals or trace elements.

Line 45: consider rephrasing, you mean the pathways of trace metals into the CaCO3? What do you mean with multistage process? Do the organisms form there shell through meta-stable phases of CaCO3?

Line 124: remove 'it'

Line 124-125: I disagree. You demonstrate the study area is highly fluctuating, so the older specimens will have experienced different environmental conditions. Change e.g. to '.. we can try to disentangle...'

Line 135: renumber your figures, this should be fig. 1, since it is the first figure you reference). Check full manuscript

Line 169: So this is then Fig. 2.

Paragraph 2.3 As requested before, please add the sample time(month) in the manuscript, not only in the table.

Line 259: Is this 20% error propagated into the SD/values reported in Table 3?

Line 221: change to 'based on the  size'

Line 235-238: Samples were split and measured by both ICP-OES and ICP-MS?

Line 254: What was the precision (%) compared to the reference values?

Line 272-274: "However, the concentrations of given metals were different between shells of Cerastoderma glaucum, Mya arenaria, Limecola balthica, Mytilus trossulus and Amphibalanus improvisus, showing high variability (Table 3, Fig. 3)." This sentence is confusion, please clarify the meaning. The variability was within a species, or between species?

Lin 367: change 'contained'

Line 364: in inorganic precipitated carbonate, precipitation rate is inversely correlated to Mn incorporation (i.e. higher precipitation equals less Mn, Lorens, 1981). Why is the opposite suggested for organic carbonates?

Line 368: change to 'is in some degree regulating'

Line 382: trace metal, not trace metals

Line 394-395: Please look into these sentences and your references. Takesue et al., 2008 showed actually the organic content was 19% for their studied species, and say aragnotic shells have 'large component of non-lattice-bound Mg and Mn'. This is actually suggesting the opposite: at least Mg and Mn might be impacted by % organic matter.

Line 394: separate ref for Sr? if not, remove

Line 399-401. I totally agree, and believe this should also be mentioned in the abstract. (e.g. line 28: add composition and contribution of organic material)

Discussion: Please also include organic material as a factor contributing to difference in (at least) Mg and Mn.

Line 410-4111: I do not follow this logic. The larger shells (adults) still experienced a full year of environmental fluctuations

Line 571 factors instead of features? Also, precipitation rate is a biological control? I would leave it at growth rate.

581-583: It does not connect to the previous sentence, maybe rephrase or remove? This will never be possible from field samples, since it is impossible to know the biological impacts (growth, feed rate etc.).

Table 2. Typo for size class I of Mytilus (should be 6-15)?

Figure 3. Consider also adding a symbols/column with values from other studies on the same species, maybe as a supplementary figure, if figure 2 becomes to information dense. I see the authors added Table 4, but I would strongly advise to change this to a graphical presentation.

---

## Referee Report (RR2)

Review of revised manuscript version bg-2019-367

**"The patterns of elemental concentration (Ca, Na, Sr, Mg, Mn, Ba, Cu, Pb, V, Y, U and Cd) in shells of invertebrates representing different CaCO$_3$ polymorphs: a case study from the brackish Gulf of Gdańsk (the Baltic Sea)"** by Anna Piwoni-Piórewicz, Stanislav Strekopytov, Emma Humphreys-Williams, and Piotr Kukliński

I would like to acknowledge the authors' comprehensive revision. It is a shame that certain changes could not be implemented, but I understand these would require further sampling and measurements that at this stage may no longer be feasible. The authors could, however, keep them in mind for their future work. Overall, I enjoyed reading the new manuscript version, that has been significantly improved and will be of valuable contribution to the field. I only have few minor, mostly editorial comments.

Line 18: 'shells are discussed'

Line 26: '…Cd) are more variable'

Line 30: do you refer to your different sites? The should be 'sampling sites within the Gulf', if this refers to the comparison to literature data then this should be specified.

Line 36: ',mainly polymers,' comma missing

Line 41: different font sizes, please correct

Line 49: I would suggest to move the brackets after chemical factors, i.e. as follows '…chemical factors (e.g. metal…; Blackmore and Wang, 2002)'

Line 52: I do not think this is correct, they are important, but I think dissolved C in the ocean is the largest C reservoir?

Line 117: 'carbonate skeleton'

Line 135: this is the figure to appear so should be Figure 1

Line 156: delete 'and' i.e. 'that is, lower Ca2+'

Line 158-159: The equation for omega is incorrect, should be: $\Omega = \frac{[Ca^{2+}] \times [CO_3^{2-}]}{K_{sp}^*}$, where K*sp is the solubility product calcite or aragonite (depending on the in situ conditions). The citation is not necessary as this is basic thermodynamics, but I see Kawahata et al., 2019 present a nice overview so could be put as 'e.g. Kawahata et al., 2019'.

Line 214-215: There is something wrong with this sentence, it is said that 'Three species were found at MA…' but in the brackets there is only one given, and then for following stations M2 and MW it is again one each.

Line 234: Subheading space formatting – above and below not same as for the other headings

Line 234: 'uncertainitity (2.5 SD)' please be more specific, 2.5 of what?

Line 259: '4-10 % for all other elements' I would suggest to provide the details for each, unless it is always assumed that the uncertainty is not better than 10%, but then please mention it

Line 260: 'et al. 2017'

Line 275: Should not be '316,000'? Please check the comma positions.

Line 276: Should not be '363,000'?

Line 283: 'Both Sr and Mg were…'

Line 296: 'Shells' which ones you refer to?

Line 300: Two spaces missing, should be 'H ='

Line 310: 'Shells' which ones?

Line 370: better 'increased incorporation'?

Line 372: comma missing after 'in our study'

Line 376: 'while larger Pb radii are favoured in aragonite structure'?

Line 379: 'determine their concentrations'

Line 385: improvement of the data'

Line 405: 'life spans'

Line 409: 'drive its biogeochemical cycles'

Line 411: commas missing ',to some extent,'

Line 413-416: Please rephrase this sentence (seems convulted)

Line 429: 'if a significant'

Line 431: ' while size class effects were less pronounced in clams'? Also 'the varied elements' please rephrase

Line 451: 'activity of an organism'

Line 488: Mg/Ca in many marine calcifiers has been used a temperature proxy, and known to be positively correlated with temperature. At the same time, Mg incorporation into the lattice is strongly dependent on growth rate, with more Mg incorporated under decreased precipitation rates. In addition, and in contrast to some other elements like Sr for example, Mg is thought to be transported by different pathways, and is strongly physiologically relevant. Growth increments are also lined with very high Mg contents, potentially linked to

increased organics. Some organisms, like low-Mg brachiopods for example, actively discriminate against Mg. In summary, there are many known controls over Mg, but from the data in Figure 3 it is clear that in the studied samples it is the mineralogy (and especially on such larger orders).

Line 530: 'in a shallow zone'

Line 544: 'most likely caused'

Line 545: 'as the trace elements'?

Line 6002: 'Mytilus' should be in italics

Table 1: First row in the table 'in situ' should be in italics

Table 3: Is the SD the standard deviation between all specimens of a species, please specify. Also please include sample size n along with the SDs and SEs.

Table 4: I think there is something wrong with the formatting, some cells have hyphens that appear out of place, please check.

Supplement: 'Table S1' (not Tabke S1)

A couple of suggestions for the authors to consider that could be of aid to their future work:

I assume that the difference in the dissolution procedures between standards and samples should not significantly affect the final results of the study, however, for the future I urge the authors to treat samples and standards exactly the same. They should also consider finding more appropriate standards for biologic samples. A potential easy solution would be the making of an in-house standard by homogenising several individuals of a species, and cross-checking the different dissolution procedures on it. If an accepted international standard is not available for routine measurements, the authors could at least request small aliquots from the community or colleagues, cross check their in-house standard against it, and include it to the routine of running the limestone and dolomite standards. This would make their data and analytical procures way more convincing from the start and overcome some of the misunderstandings.

For the future, I would like to advise the authors to also collect and consider measuring water samples alongside the calcifiers. Such information would be very valuable and of great interest to the field. Widely accepted protocols for water sampling for trace element analyses are available from GEOTRACES.

---

## Author Response (AR2)

*We would like to thank both Reviewers for the detailed and valuable comments which allowed us to improve this manuscript. Implementation of reviewers' suggestions added a lot of quality to the manuscript.*

**Review 1 of revised manuscript version bg-2019-367**

Some specific comments on the response to the author document:

5 • In the 'Author response' they do state that there is no protocol they could have used ('we do not believe there is a single accepted protocol for bivalve shells, that is tested and validated for a large range of trace elements'), however, if the whole aim of your study is to compare such a wide range of elements in different shells, maybe some method development should have taken place before. Even (relatively) simple comparison of shell fragments of a single shell with different cleaning treatment (versus no cleaning) applied, could have make
10 their dataset more robust. I highly suggest that for future work cleaning methods as well as the impact of organics should be investigated!

*We did not have an opportunity to implement this approach at this stage, but we agree that organic matter is an important factor in biogeochemical research and we will make every effort to improve the methodology in our future research.*

15 Authors: In the revised version of the manuscript, we emphasise that the contribution of the organic matter to the elemental variation is likely to be relatively minor considering that bivalve and barnacle shells contain, in general, less than 5% organic matter (Bourget, 1987; Marin and Luquet, 2004; Rueda and Smaal, 2004)

• Reviewer: 5% in weight? Is there any idea about the concentration of certain elements in this organic matter? Even if it is 5% of the total weight, if the concentration of e.g. Na is very high, this still leads to a biased E/Ca
20 for the CaCO3. Also, you refer to Takesue et al., 2008, who found e.g. 19% organic matter.

*We agree that even a small amount of the organic content of a shell can affect its chemical composition. According to literature data, the organic content of the studied organisms ranges up to about 5% in weight (e.g. Wolowicz and Goulletquer, 1999). Unfortunately, we are not able to determine what proportion of the examined elements comes from the organic matter and this is our task for the future research. In the presented manuscript, we noted*
25 *that the entire skeletons were analyzed and we tried to highlight it in the discussion of the results. One of the references we refer to is Takesue at al. (2008) and in fact the authors studied shells there by up to 19% of organic matter. But the trends of metal accumulation were similar to many other researchers, which was widely discussed in this publication, therefore we decided include this work in our manuscript.*

• After comments by both me and the other reviewer, I still do not see any discussion on the fact that lager (and
30 thus older) shells have also experienced more time, and more environmental fluctuations. Can the authors comment on this directly?

*Please look at the fragment in Lines 398 – 412, where we discussed this issue.*

Minor comments:

(General comment) The manuscript is incredibly dense in information. Of course, it is great to have numerous references and explanations for observed patterns, but also makes it sometimes difficult to absorb the main take home messages. Be careful for repetition, for instance, please check overlap between paragraph 2.2 and 4.3. *In the reviewed version of the manuscript, we tried to limit the overlapping information, for example on paragraph 2.2.*

- Line 17: please add clearly you work on bulk material. *This was added*
- Line 27: "Moreover, the elemental concentrations tend to be lower in larger than in smaller shells." This is confusing, suggesting to change to something like "lower in the large size class compared to the smaller size class. *Improved.*
- Line 41-43: changes in font size. *Corrected.*
- Line 44: consider changing metals to trace metals or trace elements. *Corrected.*
- Line 45: consider rephrasing, you mean the pathways of trace metals into the $CaCO_3$? What do you mean with multistage process? Do the organisms form their shell through meta-stable phases of $CaCO_3$? *We meant that elements taken from the environment are transported to the skeleton through the body, so biological processes can affect their final concentration in the skeletons. The sentence was worded ambiguously, so we corrected it but chose to put it elsewhere, in Lines 94 – 96.*
- Line 124: remove 'it'. *Corrected.*
- Line 124-125: I disagree. You demonstrate the study area is highly fluctuating, so the older specimens will have experienced different environmental conditions. Change e.g. to '.. we can try to disentangle...'. *Corrected.*
- Line 135: renumber your figures, this should be fig. 1, since it is the first figure you reference). Check full manuscript. Line 169: So this is then Fig. 2. *Corrected.*
- Paragraph 2.3 As requested before, please add the sample time(month) in the manuscript, not only in the table. *Corrected.*
- Line 259: Is this 20% error propagated into the SD/values reported in Table 3? *No, the SDs in Table 3 reflect the spread of concentrations between individual specimens, not the analytical error. In most, if not all cases, the spread is much higher than the estimate of the analytical error anyway.*
- Line 221: change to 'based on the size'. *Corrected.*
- Line 235-238: Samples were split and measured by both ICP-OES and ICP-MS? *Both methods of analysis are solution-based. From an aliquot of homogenized powder, a single solution was prepared, which was then split and diluted as necessary for both the ICP-OES and ICP-MS analyses.*
- Line 254: What was the precision (%) compared to the reference values? *The estimated analytical uncertainty based on the reference values is given in Lines 255 – 258: "Based on the analyses of CRMs and matrix-matched solutions, the maximum analytical error for the typical range of concentrations in the shells can be estimated (in relative percentage) as 1.5% for Ca, Mg and Sr; 3% for Ba; 20% for Cu and U; and 10% for all other elements." If by "precision" one understands repeatability, it is similar or smaller than the uncertainty of the reference values, see Table 1 in Piwoni-Piorewicz et al. (2017). The measurements in this work were done using the same technique, equipment and analysts and gave very similar results for the standards we decided not to include the tabulated data for the standards in this work, but refer the reader to our previous work: Lines 258 – 259: "More details on method validation were reported previously (Piwoni-Piórewicz et al 2017)."*

- Line 272-274: "However, the concentrations of given metals were different between shells of Cerastoderma glaucum, Mya arenaria, Limecola balthica, Mytilus trossulus and Amphibalanus improvisus, showing high variability (Table 3, Fig. 3)." This sentence is confusion, please clarify the meaning. The variability was within a species, or between species? *Corrected.*
- Lin 367: change 'contained'. *Changed into "produced".*
- Line 364: in inorganic precipitated carbonate, precipitation rate is inversely correlated to Mn incorporation (i.e. higher precipitation equals less Mn, Lorens, 1981). Why is the opposite suggested for organic carbonates? *The inorganic calcite precipitation experiments have shown a negative relationship between precipitation rate and the Mn partition coefficient (e.g. Lorens, 1981). However, the relationship between the growth rate of organism and shell Mn/Ca was revealed to be positive. This indicates that Mn concentration in shells is not directly controlled by a kinetic calcium carbonate precipitation rate, but must reflect other processes, likely physiological in origin (e.g. Carré et al., 2006; Freiras et al., 2016). We discussed this issue in lines 445 – 448.*
- Line 368: change to 'is in some degree regulating' Line 382: trace metal, not trace metals. *Corrected.*
- Line 394-395: Please look into these sentences and your references. Takesue et al., 2008 showed actually the organic content was 19% for their studied species, and say aragnotic shells have 'large component of non-lattice-bound Mg and Mn'. This is actually suggesting the opposite: at least Mg and Mn might be impacted by % organic matter. Line 394: separate ref for Sr? if not, remove. *These sentences were improved and the role of organic matrix in the total concentration of metals in CaCO₃ skeletons was highlighted.*
- Line 399-401. I totally agree, and believe this should also be mentioned in the abstract. (e.g. line 28: add composition and contribution of organic material). *Improved.*
- Discussion: Please also include organic material as a factor contributing to difference in (at least) Mg and Mn. *This was included in Lines 394 – 398.*
- Line 410-4111: I do not follow this logic. The larger shells (adults) still experienced a full year of environmental fluctuations. *Yes, but we believe that the long-lived shells additionally experienced environmental variability between years, and consequently elemental variability in shells of barnacles with lifespan of ~1 year might by lower.*
- Line 571 factors instead of features? Also, precipitation rate is a biological control? I would leave it at growth rate. *Corrected.*
- 581-583: It does not connect to the previous sentence, maybe rephrase or remove? This will never be possible from field samples, since it is impossible to know the biological impacts (growth, feed rate etc.). *The sentence was removed.*
- Table 2. Typo for size class I of Mytilus (should be 6-15)? *Corrected.*
- Figure 3. Consider also adding a symbols/column with values from other studies on the same species, maybe as a supplementary figure, if Figure 2 becomes to information dense. I see the authors added Table 4, but I would strongly advise to change this to a graphical presentation. *Such a figure would indeed be too information-dense. We still believe that the data in Table 4 are better in a tabular form so that other researchers can use it for reference, but we agree that this table can go to the Supplement if needed.*

**Review 2 of revised manuscript version bg-2019-367**

- Line 18: 'shells are discussed' Line 26: '...Cd) are more variable'. *Corrected.*
- Line 30: do you refer to your different sites? This should be 'sampling sites within the Gulf', if this refers to the comparison to literature data then this should be specified. *Corrected.*
- Line 36: ',mainly polymers,' comma missing Line 41: different font sizes, please correct. *Corrected.*
- Line 49: I would suggest to move the brackets after chemical factors, i.e. as follows '...chemical factors (e.g. metal...; Blackmore and Wang, 2002)'. *Corrected.*
- Line 52: I do not think this is correct, they are important, but I think dissolved C in the ocean is the largest C reservoir? *We agree that this is an imprecise statement and this was corrected.*
- Line 117: 'carbonate skeleton'. *Amended.*
  Line 135: this is the figure to appear so should be Figure 1. *Corrected.*
- Line 156: delete 'and' i.e. 'that is, lower Ca2+'. *Corrected.*
- Line 158-159: The equation for omega is incorrect, K*sp is the solubility product calcite or aragonite (depending on the in situ conditions). The citation is not necessary as this is basic thermodynamics, but I see Kawahata et al., 2019 present a nice overview so could be put as 'e.g. Kawahata et al., 2019'. *Corrected.*
- Line 214-215: There is something wrong with this sentence, it is said that 'Three species were found at MA...' but in the brackets there is only one given, and then for following stations M2 and MW it is again one each. *Corrected.*
- Line 234: Subheading space formatting – above and below not same as for the other headings. *Corrected.*
- Line 234: 'uncertainty (2.5 SD)' please be more specific, 2.5 of what? *The uncertainty of the recommended values (Imai, 1996).*
- Line 259: '4-10 % for all other elements' I would suggest to provide the details for each, unless it is always assumed that the uncertainity is not better than 10%, but then please mention it. *Changed to 10%.*
- Line 260: 'et al. 2017'. *Corrected.*
- Line 275: Should not be '316,000'? Please check the comma positions. *Corrected.*
- Line 276: Should not be '363,000'? *Yes, this was corrected.*
- Line 283: 'Both Sr and Mg were...' *Corrected.*
- Line 296: 'Shells' which ones you refer to? *Corrected.* Line 300: Two spaces missing, should be 'H ='. *Corrected.* Line 310: 'Shells' which ones? *Improved.*
- Line 370: better 'increased incorporation'? *Corrected.*
- Line 372: comma missing after 'in our study'. *Corrected.* Line 376: 'while larger Pb radii are favoured in aragonite structure'? *Corrected.*
- Line 379: 'determine their concentrations'. *Corrected.*
- Line 385: improvement of the data'. *Corrected.*
- Line 405: 'life spans'. *Corrected.*
- Line 409: 'drive its biogeochemical cycles'. *Corrected.*
- Line 411: commas missing ', to some extent,'. *Corrected.* Line 413-416: Please rephrase this sentence (seems convulted). *Improved.*
- Line 429: 'if a significant'. *Corrected.*

150 • Line 431: ' while size class effects were less pronounced in clams'? Also 'the varied elements' please rephrase. *Improved.*

• Line 451: 'activity of an organism'. *Corrected.*

• Line 488: Mg/Ca in many marine calcifiers has been used a temperature proxy, and known to be positively correlated with temperature. At the same time, Mg incorporation into the lattice is strongly dependent on
155 growth rate, with more Mg incorporated under decreased precipitation rates. In addition, and in contrast to some other elements like Sr for example, Mg is thought to be transported by different pathways, and is strongly physiologically relevant. Growth increments are also lined with very high Mg contents, potentially linked to increased organics. Some organisms, like low-Mg brachiopods for example, actively discriminate against Mg. In summary, there are many known controls over Mg, but from the data in Figure 3 it is clear that in the studied
160 samples it is the mineralogy (and especially on such larger orders). *We totally agree with this and all was included in the different sections of our discussion.*

• Line 530: 'in a shallow zone'. *Corrected.*

• Line 544: 'most likely caused'. *Corrected.* Line 545: 'as the trace elements'? *Corrected.* Line 6002: 'Mytilus' should be in italics. *Corrected.* Table 1: First row in the table 'in situ' should be in italics. *Corrected.*
165 • Table 3: Is the SD the standard deviation between all specimens of a species, please specify. Also please include sample size n along with the SDs and SEs. *Yes, this is the SD between all specimens. This is now explained in the table. The sample size is also given in the table.*

• Table 4: I think there is something wrong with the formatting, some cells have hyphens that appear out of place, please check. *Improved.*
170 • Supplement: 'Table S1' (not Tabke S1). *Corrected.*

A couple of suggestions for the authors to consider that could be of aid to their future work:

I assume that the difference in the dissolution procedures between standards and samples should not significantly affect the final results of the study, however, for the future I urge the authors to treat samples and standards exactly the same. They should also consider finding more appropriate standards for biologic samples. A potential easy
175 solution would be the making of an in-house standard by homogenising several individuals of a species, and cross-checking the different dissolution procedures on it. If an accepted international standard is not available for routine measurements, the authors could at least request small aliquots from the community or colleagues, cross check their in-house standard against it, and include it to the routine of running the limestone and dolomite standards. This would make their data and analytical procures way more convincing from the start and overcome some of the
180 misunderstandings.

For the future, I would like to advise the authors to also collect and consider measuring water samples alongside the calcifiers. Such information would be very valuable and of great interest to the field. Widely accepted protocols for water sampling for trace element analyses are available from GEOTRACES.

*We are very grateful for your comments, we will definitely use these suggestions in our future work.*

185 *Line 1084: We changed the photos 2, 4 and 5 in Figure 2 as the new ones, in our opinion, represent species better.*

References:

[revised manuscript text omitted]

— usunięto: B

— usunięto: higher growth rate or

— usunięto: shell organic matter

— usunięto: in

— usunięto: At one station (GN, Fig. 2), we observed the highest concentration of Mn and Ba in the shells, and this pattern might be determined by the environmental parameters at that location. Yet, shells of both species with the highest concentration of Mn (*A. improvisus, M. trossulus*) contained calcite, therefore we should not rule out that the polymorph type of CaCO$_3$ is regulating to some degree the level of shell Mn in low salinity environment.¶

— usunięto: great incorporation

— usunięto: , in aragonite structure

— usunięto: metals

— usunięto: , Fig. 2

— usunięto: s

— usunięto: metal

— usunięto: metals

— usunięto: , Fig. 2

— usunięto: metals

— usunięto: metal

— usunięto: n

[revised manuscript text omitted]

— usunięto: N
— usunięto: ±1
— usunięto:

| Element (El) | Sample size n | Mean conc. [$\mu g\ g^{-1}$] | SD | Weight ratio El/Ca [arbitrary] | Molar ratio El/Ca [mmol/mol] |
|---|---|---|---|---|---|
| Ca | 40 | 363,000 | 18,159 | 1 | 1 |
| Na | 40 | 3,080 | 207 | 0.0085 | 14.8 |
| Sr | 40 | 2,160 | 280 | 0.0060 | 2.72 |
| Mg | 40 | 150 | 25 | 0.0004 | 0.68 |
| Ba | 40 | 2.60 | 1.6 | 0.00716 | 2.1 |
| Mn | 40 | 1.40 | 1 | 0.00386 | 2.8 |
| Cu | 40 | 0.031 | 0.024 | 0.00009 | 0.053 |
| Y | 40 | 0.020 | 0.018 | 0.00006 | 0.025 |
| V | 40 | 0.016 | 0.014 | 0.00004 | 0.035 |
| Pb | 40 | 0.015 | 0.014 | 0.00004 | 0.008 |
| U | 40 | 0.006 | 0.005 | 0.00002 | 0.003 |
| Cd | 37 | 0.002 | 0.003 | 0.00001 | 0.002 |

— usunięto: N
— usunięto: ±1
— usunięto:

| Element (El) | Sample size n | Mean conc. [$\mu g\ g^{-1}$] | SD | Weight ratio El/Ca [arbitrary] | Molar ratio El/Ca [mmol/mol] |
|---|---|---|---|---|---|
| Ca | 40 | 344,000 | 28,000 | 1 | 1 |
| Na | 21 | 2,010 | 310 | 0.006 | 10.2 |
| Sr | 41 | 1,170 | 100 | 0.003 | 1.6 |
| Mg | 40 | 1,060 | 170 | 0.003 | 5.1 |
| Mn | 35 | 54 | 15 | 0.157 | 114.16 |
| Ba | 38 | 17.0 | 6.7 | 0.049 | 14.40 |
| Cu | 40 | 14.0 | 10 | 0.041 | 25.44 |
| Pb | 32 | 1.00 | 1.4 | 0.003 | 0.56 |
| V | 40 | 0.78 | 0.52 | 0.002 | 1.78 |
| Y | 40 | 0.70 | 1.2 | 0.002 | 0.91 |
| Cd | 40 | 0.09 | 0.11 | 0.0003 | 0.09 |
| U | 40 | 0.05 | 0.033 | 0.0002 | 0.03 |

**Table 4.** Elemental concentrations in shells from different regions based on this study and literature data.

| Element concentration in shell | Ca | Na | Sr | Mg | Mn | Ba | Cu | Pb | Cd | U |
|---|---|---|---|---|---|---|---|---|---|---|
| | | | [$\mu g\ g^{-1}$] | | | | | [$\mu g\ g^{-1}$] | | |
| *Amphibalanus improvisus* [a] | 316000 | 2850 | 2250 | 3940 | 625 | 73 | 3.90 | 1.34 | 0.11 | 0.06 |
| *Balanus sp.* [o] | | 1600 – 5000 | | | 80 – 3800 | | | | | |
| *Balanus sp.* [z] | | | | | 39 – 313 | | | <18.2 | 0 – 1.48 | |
| *Balanus balanoides* [d] | | | | | 77 | | | | | |

Sformatowana tabela
Sformatowano: Do lewej
Sformatowano: Do lewej
Sformatowano: Do lewej
Sformatowano: Do lewej

[revised manuscript text omitted]

— **usunięto:** metals

1575

---

## Author Response (AR3)

*Dear Editor Lennart de Nooijer,*

*Thank you very much for reading the revised version of our manuscript and for your valuable comments. We have made every effort to respond to your reviews.*

**Abstract.**

Please remove the first two sentences of the second paragraph (this is already apparent from the numerous papers published on biogenic carbonates) and combine the two paragraphs into one. *Removed.*

**Introduction.**

line 34. Now it sounds as if these organisms actively seek for e.g. Sr and Mg to produce their shells. It may well be that these ions are incorporated 'by accident' and that only Ca and DIC are actively collected. Please rephrase. *Corrected.*

line 39. That they can be found in the same specimen is only true for some taxa. *Corrected.*

line 41-42. This is incorrect. The 'bulk elemental composition' (a rather vague term, btw) is determined by the biomineralization mechanism of the organism. The presence of organic compounds may also influence the (trace) element composition, but I am not sure what 'crystal lattice' does here. The crystal structure is set by the polymorph precipitated, I guess. And it also determines how much of which element can be incorporated (i.e. aragonite has a much lower maximum Mg content than calcite). I think this whole sentence can be omitted. Otherwise please rephrase. *This fragment has been removed.*

line 63-65. Since the introduction is rather long, I suggest to delete the first two sentences of this paragraph. *Corrected as suggested.*

line 84-85. Please delete this sentence. *Deleted.*

line 96: "...pathway to the skeleton or shell....", please delete "of the body". *Corrected.*

line 99. Doesn't have to be completely independent: the environment (e.g. temperature) can affect the transport of e.g. Mg towards the fluid from which the CaCO3 is precipitated. *The sentence has been modified to be more specific.*

line 113-119. This whole paragraph is a bit repetitious. Consider removing it altogether. *Removed.*

**Methods.**

line 136-137. "The north-western part of the gulf is separated by..." "... and in the west and south...". *Corrected.*

line 151-153. Remove here: the same is already stated in the first paragraph. *Removed*

**Results.**

lines 274 and further. Please turn "316,000 +/- 26,000 µg/g" into "316 +/- 26 mg/g". Also, for other high (> 1 mg) concentrations. *The units have been converted in the Results and Discussion sections, and in Figure 4, yet not in Figure 3 and Tables to keep the results consistent.*

line 296. "...variability in four trace...". *Corrected.*

**Discussion.**

line 325. "The average Ca concentration...". *Corrected.*

line 380-396. It may also be that ions are present in microscopic fluid inclusions. This will need to be mentioned here, as well as the effect it may have on the assumed lattice-bound element concentrations. *This issue was mentioned in lines 379 – 384.*

line 412. "lead". *Corrected.*

[revised manuscript text omitted]

Polski

| | | |
|---|---|---|
| **Strona 8: [4] — sformatowano** | **Anna Piwoni-Piórewicz** | **02.11.2020 18:28:00** |

Polski

| | | |
|---|---|---|
| **Strona 8: [5] — usunięto** | **Anna Piwoni-Piórewicz** | **26.10.2020 21:30:00** |
| **Strona 8: [5] — usunięto** | **Anna Piwoni-Piórewicz** | **26.10.2020 21:30:00** |
| **Strona 8: [5] — usunięto** | **Anna Piwoni-Piórewicz** | **26.10.2020 21:30:00** |
| **Strona 8: [5] — usunięto** | **Anna Piwoni-Piórewicz** | **26.10.2020 21:30:00** |
| **Strona 8: [5] — usunięto** | **Anna Piwoni-Piórewicz** | **26.10.2020 21:30:00** |
| **Strona 8: [5] — usunięto** | **Anna Piwoni-Piórewicz** | **26.10.2020 21:30:00** |
| **Strona 8: [5] — usunięto** | **Anna Piwoni-Piórewicz** | **26.10.2020 21:30:00** |
| **Strona 8: [5] — usunięto** | **Anna Piwoni-Piórewicz** | **26.10.2020 21:30:00** |
| **Strona 8: [5] — usunięto** | **Anna Piwoni-Piórewicz** | **26.10.2020 21:30:00** |
| **Strona 8: [5] — usunięto** | **Anna Piwoni-Piórewicz** | **26.10.2020 21:30:00** |
| **Strona 8: [5] — usunięto** | **Anna Piwoni-Piórewicz** | **26.10.2020 21:30:00** |
| **Strona 8: [5] — usunięto** | **Anna Piwoni-Piórewicz** | **26.10.2020 21:30:00** |

| Strona 8: [5] — usunięto | Anna Piwoni-Piórewicz | 26.10.2020 21:30:00 |
|---|---|---|
| Strona 8: [5] — usunięto | Anna Piwoni-Piórewicz | 26.10.2020 21:30:00 |
| Strona 8: [5] — usunięto | Anna Piwoni-Piórewicz | 26.10.2020 21:30:00 |
| Strona 8: [5] — usunięto | Anna Piwoni-Piórewicz | 26.10.2020 21:30:00 |
| Strona 8: [5] — usunięto | Anna Piwoni-Piórewicz | 26.10.2020 21:30:00 |
| Strona 8: [5] — usunięto | Anna Piwoni-Piórewicz | 26.10.2020 21:30:00 |
| Strona 8: [5] — usunięto | Anna Piwoni-Piórewicz | 26.10.2020 21:30:00 |
| Strona 8: [5] — usunięto | Anna Piwoni-Piórewicz | 26.10.2020 21:30:00 |
| Strona 8: [5] — usunięto | Anna Piwoni-Piórewicz | 26.10.2020 21:30:00 |
| Strona 8: [5] — usunięto | Anna Piwoni-Piórewicz | 26.10.2020 21:30:00 |
| Strona 8: [5] — usunięto | Anna Piwoni-Piórewicz | 26.10.2020 21:30:00 |
| Strona 8: [5] — usunięto | Anna Piwoni-Piórewicz | 26.10.2020 21:30:00 |
| Strona 8: [5] — usunięto | Anna Piwoni-Piórewicz | 26.10.2020 21:30:00 |